# Stochastic Extragradient with Flip-Flop Shuffling & Anchoring: Provable Improvements

**Jiseok Chae**
Department of Mathematical Sciences
KAIST
Daejeon, Republic of Korea
jsch@kaist.ac.kr

**Chulhee Yun**
Kim Jaechul Graduate School of AI
KAIST
Seoul, Republic of Korea
chulhee.yun@kaist.ac.kr

**Donghwan Kim**
Department of Mathematical Sciences
KAIST
Daejeon, Republic of Korea
donghwankim@kaist.ac.kr

## Abstract

In minimax optimization, the extragradient (EG) method has been extensively studied because it outperforms the gradient descent-ascent method in convex-concave (C-C) problems. Yet, stochastic EG (SEG) has seen limited success in C-C problems, especially for unconstrained cases. Motivated by the recent progress of shuffling-based stochastic methods, we investigate the convergence of *shuffling-based SEG* in unconstrained *finite-sum* minimax problems, in search of convergent shuffling-based SEG. Our analysis reveals that both random reshuffling and the recently proposed flip-flop shuffling alone can suffer divergence in C-C problems. However, with an additional simple trick called anchoring, we develop the *SEG with flip-flop anchoring* (SEG-FFA) method which successfully converges in C-C problems. We also show upper and lower bounds in the strongly-convex-strongly-concave setting, demonstrating that SEG-FFA has a provably faster convergence rate compared to other shuffling-based methods.

## 1 Introduction

Minimax problems with a finite-sum structure, which are optimization problems of the form

$$\min_{\boldsymbol{x}} \max_{\boldsymbol{y}} f(\boldsymbol{x}, \boldsymbol{y}) := \frac{1}{n} \sum_{i=1}^{n} f_i(\boldsymbol{x}, \boldsymbol{y}), \tag{1}$$

can be found in many interesting applications, such as generative adversarial networks [19], refining diffusion models [28], adversarial training [37], optimal transport based generative models [48], multi-agent reinforcement learning [53], and so on. Deterministic methods for minimax problems, such as *gradient descent-ascent* (GDA) [3] and *extragradient* (EG) [29], have been extensively studied in the literature. It is though known that, unlike *gradient descent* (GD) for minimization problems, GDA may diverge even when $f$ is convex on $\boldsymbol{x}$ and concave on $\boldsymbol{y}$. On the other hand, EG employs a two-step update procedure, named *extrapolation* and *update* steps (see Section 2 for details), which allows it to find an optimum under this *convex-concave* setting [29, 51], and moreover, attains a convergence rate faster than GDA [4] when $f$ is strongly convex on $\boldsymbol{x}$ and strongly concave on $\boldsymbol{y}$.

In contrast, attempts to construct stochastic variants of these algorithms have not been so fruitful. When $f$ is convex-concave, *stochastic gradient descent-ascent* (SGDA) clearly may diverge, just as

38th Conference on Neural Information Processing Systems (NeurIPS 2024).

in the deterministic GDA. To make matters worse, *stochastic extragradient* (SEG) methods have also had limited success on unconstrained convex-concave problems. As we elaborate in Section 2 in more detail, existing versions of SEG and their analyses have limitations that hinder its application to general unconstrained finite-sum convex-concave problems, requiring additional assumptions such as bounded domain, increasing batch size, convex-concavity of each component $f_i$, uniformly bounded gradient variance, and/or absence of convergence rates.

In the context of finite-sum optimization, most of the theoretical studies on stochastic methods have long been based on the *with-replacement* sampling scheme, where an index $i(t)$ is independently and uniformly sampled among $\{1, \ldots, n\}$ at each iteration $t$. Such a sampling scheme is relatively easy to theoretically analyze, because the sampled $f_{i(t)}$ is an unbiased estimator of the full objective function $f$. In practice, however, inspired by the empirical observations of faster convergence in finite-sum minimization [8, 47], the *without-replacement* sampling schemes have been the de facto standard. Among them, the most popular is the *random reshuffling* (RR) scheme, where in every *epoch* consisting of $n$ iterations, the indices are chosen exactly once in a randomly shuffled order.

This gap between theory and practice in minimization problems is being closed by the recent breakthroughs in stochastic gradient descent (SGD), namely that SGD with RR leads to a provably faster convergence compared to with-replacement SGD when the number of epochs is large enough [1, 35, 39, 41, 55, 56]. This has motivated further studies on finding other shuffling-based sampling schemes that can improve upon RR, resulting in the discoveries such as the *flip-flop* scheme [46] and *gradient balancing* (GraB) [11, 32]. The flip-flop scheme is a particularly simple yet interesting modification of RR with improved rates in *quadratic* problems: a random permutation is used twice in a single epoch (i.e., two passes over $n$ components in an epoch), but the order is reversed in the second pass.

The aforesaid progress in minimization also triggered the study of stochastic minimax methods with shuffling. Similar to minimization problems, SGDA with RR indeed converges faster than the with-replacement SGDA, under assumptions such as strongly-convex-strongly-concave objectives [15] or $f$ satisfying the Polyak-Łojasiewicz condition [13]. Despite the superiority of EG over GDA, the *SEG with shuffling* has not been shown to have a solid theoretical advantage over the SGDA with shuffling yet. This motivated us to study the following question:

> ***Can shuffling schemes provide convergence guarantees for SEG, improved upon SGDA with shuffling, in unconstrained finite-sum (strongly-)convex-(strongly-)concave settings?***

There are two types of SEG: *same-sample* SEG, where a sample chosen is used both for the extrapolation step and the update step, and *independent-sample* SEG, where two independently chosen samples are used in each step. We will particularly focus on the same-sample SEG because it combines more naturally with shuffling-based schemes than independent-sample SEG. Therefore, to be more specific, we are interested in developing shuffling-based variants of *same-sample* SEG in unconstrained finite-sum minimax problems with minimal modifications to the algorithm. We show that **(a)** in convex-concave settings, our new method reaches an optimum with a guarantee on the rate of convergence, overcoming the limitations of existing results; **(b)** in strongly-convex-strongly-concave settings, the method converges faster than other SGDA/SEG variants.

## 1.1 Our Contributions

In this paper, we study various same-sample SEG algorithms under different shuffling schemes, and propose the *stochastic extragradient with flip-flop anchoring* (SEG-FFA) method, which is SEG amended with the techniques of *flip-flop* shuffling scheme and *anchoring*. Here, by *anchoring* we refer to a step of taking a convex combination between the initial and final iterates of an epoch, resembling the celebrated *Krasnosel'skiĭ-Mann iteration* [30, 33] as we discuss in Section 5. With such minimal modifications to SEG, we show that SEG-FFA achieves provably improved convergence guarantees. More precisely, our contributions can be listed as follows (see Table 1 for a summary). For clarity, we use SEG-US to refer to with-replacement SEG, where US stands for uniform sampling.

- We first study the same-sample versions of SEG-US, SEG with RR (SEG-RR), and SEG with flip-flop (SEG-FF). We show that they all can diverge when $f$ is convex-concave,[1]

---

[1]This does not contradict the result in [25], which shows that the *independent-sample* SEG with carefully designed step sizes rule converges to optima for convex-concave settings, albeit without a convergence rate.

Table 1: Summary of upper/lower convergence rate bounds of *same-sample* SEG for unconstrained finite-sum minimax problems, without requiring increasing batch size, convex-concavity of each component, and uniformly bounded gradient variance. Pseudocode of algorithms can be found in Appendix A. We only display terms that become dominant for sufficiently large $T$ and $K$. To compare the with-replacement versions (-US) against shuffling-based versions, one can substitute $T = nK$. The optimality measure used for SC-SC problems is $\mathbb{E}[\|\hat{z} - z^*\|^2]$ for the last iterate $\hat{z}$. For C-C problems, we consider $\min_{t=0,\dots,T} \mathbb{E}[\|Fz_t\|^2]$ for with-replacement methods and $\min_{k=0,\dots,K} \mathbb{E}[\|Fz_0^k\|^2]$ for shuffling-based methods.

| | STRONGLY-CONVEX-STRONGLY-CONCAVE | | CONVEX-CONCAVE | |
| METHOD | UPPER BOUND | LOWER BOUND | UPPER BOUND | LOWER BOUND |
| --- | --- | --- | --- | --- |
| SGDA-US | $\mathcal{O}(\frac{1}{T})$ [31] | $\Omega(\frac{1}{T})$ [13] | N/A | $\Omega(1)$ (AS GDA) |
| SEG-US | $\mathcal{O}(\frac{1}{T})$ [20] | $\Omega(\frac{1}{T})$ [6] | N/A[†‡] | $\Omega(1)$ (THM. 4.1) |
| SGDA-RR | $\tilde{\mathcal{O}}(\frac{1}{nK^2})$ [15] | $\Omega(\frac{1}{nK^3})$ (THM. 5.6) | N/A | $\Omega(1)$ (AS GDA) |
| SEG-RR | $\tilde{\mathcal{O}}(\frac{1}{nK^2})$ [18] | $\Omega(\frac{1}{nK^3})$ (THM. 5.6) | $\mathcal{O}(\frac{1}{(nK)^{1/3}})$? [18][§] | $\Omega(1)$ (THM. 4.1) |
| SEG-FF | $\tilde{\mathcal{O}}(\frac{1}{nK^2})$ (THM. F.5) | – | N/A | $\Omega(1)$ (THM. 4.1) |
| SEG-FFA | $\tilde{\mathcal{O}}(\frac{1}{nK^4})$ (THM. 5.5) | – | $\tilde{\mathcal{O}}(\frac{1}{K^{1/3}})$ (THM. 5.4) | – |

[†] [17, 20] show upper bounds for SEG-US, but they require increasing batch sizes as well as other assumptions (see Appendix B.1).
[‡] [25] shows that *independent-sample* SEG-US converges for stepsizes $\alpha_t$, $\beta_t$ decaying at different rates, but gives no conv. rate.
[§] Unfortunately, the proof of this convergence bound in this recent AISTATS 2024 paper seems to be incorrect: see Appendix B.4.

by constructing an explicit counterexample (Theorem 4.1). This shows that shuffling alone *cannot* fix the divergence issue of SEG-US.

- We next investigate the underlying cause for the nonconvergence of SEG-US, SEG-RR, and SEG-FF. In particular, we identify that either they fail to match the update equation of the reference method EG beyond *first-order* Taylor expansion terms, or attempting to match both the *first-* and *second-order* Taylor expansion terms results in divergence (Proposition 5.2).

- By adopting a simple technique of *anchoring* on top of flip-flop shuffling, we devise our algorithm SEG-FFA, whose epoch-wise update deterministically matches EG up to second-order Taylor expansion terms (Proposition 5.3). We prove that SEG-FFA enjoys improved convergence guarantees, as anticipated by our design principle. Most importantly, we show that SEG-FFA achieves a convergence rate of $\tilde{\mathcal{O}}(1/K^{1/3})$ when $f$ is convex-concave, where $K$ denotes the number of epochs. This is in stark contrast to other baseline algorithms that diverge under this setting (see the last column of Table 1).

- Moreover, we show that when $f$ is strongly-convex-strongly-concave, SEG-FFA achieves a convergence rate of $\tilde{\mathcal{O}}(1/nK^4)$ (Theorem 5.5). In addition, by proving $\Omega(1/nK^3)$ lower bounds for the convergence rates of SGDA-RR and SEG-RR under the same setting (Theorem 5.6), we show that SEG-FFA has a *provable advantage* over these baseline algorithms.

## 2   Related Works

**Extragradient and EG+**   Extragradient (EG) method [29] is a widely used minimax optimization method, well-known for resolving the nonconvergence issue of GDA on convex-concave problems. In this paper, we also consider EG+ [17], which is a generalization of EG. The update rule of EG+ is defined, for stepsizes $\{\eta_{1,k}\}_{k \geq 0}$ and $\{\eta_{2,k}\}_{k \geq 0}$, as

$$\begin{cases} u^k \leftarrow x^k - \eta_{1,k} \nabla_x f(x^k, y^k) \\ v^k \leftarrow y^k + \eta_{1,k} \nabla_y f(x^k, y^k) \end{cases}, \qquad \begin{cases} x^{k+1} \leftarrow x^k - \eta_{2,k} \nabla_x f(u^k, v^k) \\ y^{k+1} \leftarrow y^k + \eta_{2,k} \nabla_y f(u^k, v^k) \end{cases}. \qquad (2)$$

The first step is called the *extrapolation step*, and the second step is called the *update step*. If $f$ is convex-concave, Diakonikolas et al. [17] show that EG+ reaches an optimum when $\eta_{1,k} \geq \eta_{2,k}$. In particular, when $\eta_{1,k} = \eta_{2,k}$, we recover the standard EG by Korpelevich [29].

**Stochastic Variants of Extragradient** In (2), if the *stochastic estimators* of $\nabla_{\boldsymbol{x}} f$ and $\nabla_{\boldsymbol{y}} f$ are used instead of the gradients themselves, we get the standard SEG. If an estimator chosen is used for both the extrapolation and the update steps, we get the *same-sample* SEG, which we focus on in this paper; see Appendix A for the pseudocode.

While EG improves upon GDA, unfortunately, SEG has not been able to show a clear advantage over SGDA. On one hand, analyses of SEG on strongly-convex-strongly-concave problems have shown some success; see, *e.g.,* [18, 20]. Yet, on the other hand, for general unconstrained convex-concave problems, to the best of our knowledge, the existing stochastic variants of EG and their analyses face several limitations.[23] Assumptions commonly imposed in the existing literature include: **(i)** the domain is bounded, either explicitly or implicitly [27, 36], **(ii)** one must increase the batch size to achieve convergence [9, 17, 20],[4] and **(iii)** each component $f_i$ is convex-concave [20, 36], and **(iv)** the components have uniformly bounded gradient variance [9, 17, 42]. For further details, see Appendix B.1 and Table 2 therein. Notably, Hsieh et al. [25] prove convergence of the *independent-sample* SEG without these four restrictions, but the result lacks an explicit convergence rate.

Our proposed SEG-FFA overcomes all the aforementioned limitations, and reaches an optimum with an explicit rate in unconstrained convex-concave problems, under relatively mild conditions. The readers may also refer to [7] for a comprehensive overview on this topic.

Meanwhile, under the finite-sum setting, variance reduction schemes have also been considered, achieving some promising results [2, 10]. Yet, although theoretically appealing, variance reduction is less widely used in practice due to their curiously inferior performance in training neural networks [16]. On top of this practical issue, variance reduction techniques share the aforementioned limitation (ii), as accessing full gradients can be viewed as increasing the batch size. In contrast, our main goal in this paper is to study how a carefully chosen sampling scheme, with minimal modifications to the algorithm, can improve the convergence of SEG without the need for increased batch size; therefore, we believe that our work is not directly comparable to variance reduction-based EG.

**Taylor Expansion Matching and Convergence Guarantees** It has been repeatedly reported that the convergence of an optimization method is deeply related to the degree to which the Taylor expansion (with respect to the step size) of its update equation matches with that of an already known convergent method. For example, Mokhtari et al. [38] observed that the advantage of EG over GDA comes from the Taylor expansion of update equations of EG matching that of the *proximal point* (PP) method [34] up to second-order terms, whereas GDA matches PP only up to first-order terms.

The advantages of the shuffling scheme over the with-replacement sampling can be explained in a similar way. One key property of shuffling-based methods is that, while the individual estimators are biased as they are dependent to other estimators within the same epoch, the overall stochastic error across the epoch decreases dramatically compared to using $n$ independent unbiased estimators. For instance, in SGD with RR [1] and in SGDA with RR [15], the overall progress made within each epoch exactly matches their deterministic counterparts up to the first-order, leaving an error as small as $\mathcal{O}(\eta^2)$, where $\eta$ is the stepsize. Rajput et al. [46] observed that, when each component functions are convex quadratics, then using flip-flop on SGD can reduce the error further to $\mathcal{O}(\eta^3)$, resulting in an even faster convergence. As we further elaborate in Section 5, the motivation behind our design principle of SEG-FFA is also based on this line of observations.

## 3 Notations and Problem Settings

Let $[n] \subset \mathbb{Z}$ denote the set $\{1, \ldots, n\}$. The set of all permutations on $[n]$ will be denoted by $\mathcal{S}_n$. For the finite-sum minimax problem (1), we denote the *saddle gradient* operators by

$$\boldsymbol{F}(\cdot) := \begin{bmatrix} \nabla_{\boldsymbol{x}} f(\cdot) \\ -\nabla_{\boldsymbol{y}} f(\cdot) \end{bmatrix}, \quad \boldsymbol{F}_i(\cdot) := \begin{bmatrix} \nabla_{\boldsymbol{x}} f_i(\cdot) \\ -\nabla_{\boldsymbol{y}} f_i(\cdot) \end{bmatrix}, \quad i \in [n].$$

---

[2]Most of these results are carried out assuming access to a stochastic oracle of $f$, which indeed subsumes the finite-sum setting as a special case. However, it seems unlikely that these limitations of the existing studies will be easily resolved by simply narrowing the focus down to the finite-sum setting; see Appendix B.3.

[3]Recently, Emmanouilidis et al. [18] claimed the convergence of SEG-RR in the convex-concave setting. Unfortunately, however, there seems to be a flaw in their proof. We defer a discussion on this to Appendix B.4.

[4]In fact, for the methods studied in [17, 20] it is possible to show that increasing the batch size is *strictly necessary and unavoidable* for convergence; see Appendix H.2.

The derivative of an operator will be denoted with a prefix $D$. For example, the derivative of $\boldsymbol{F}$ is denoted by $D\boldsymbol{F}$. Often a single vector will be used to denote the minimization and the maximization variable at once. For instance, for $\boldsymbol{z} \in \mathbb{R}^{d_1+d_2}$ which is a concatenation of $\boldsymbol{x} \in \mathbb{R}^{d_1}$ and $\boldsymbol{y} \in \mathbb{R}^{d_2}$, we simply write $\boldsymbol{F}\boldsymbol{z}$ to denote $\boldsymbol{F}(\boldsymbol{x}, \boldsymbol{y})$.

It is well known that, if $f$ is $\mu$-strongly convex on $\boldsymbol{x}$ and $\mu$-strongly concave on $\boldsymbol{y}$ for some $\mu > 0$ (respectively, $\mu = 0$), then its saddle gradient $\boldsymbol{F}$ is $\mu$-strongly monotone (respectively, monotone), in the following sense. For a proof of this standard fact, see, *e.g.*, [22].

**Assumption 3.1** (Monotonicity & Strong Monotonicity). For $\mu > 0$, we say that an operator $\boldsymbol{F}$ is $\mu$-strongly monotone if, for any $\boldsymbol{z}, \boldsymbol{w} \in \mathbb{R}^{d_1+d_2}$, it holds that

$$\langle \boldsymbol{F}\boldsymbol{z} - \boldsymbol{F}\boldsymbol{w}, \boldsymbol{z} - \boldsymbol{w} \rangle \geq \mu \|\boldsymbol{z} - \boldsymbol{w}\|^2. \tag{3}$$

If (3) holds for $\mu = 0$, then we say that $\boldsymbol{F}$ is monotone.

Thus, from now on, we will use the term *strongly monotone* (respectively, *monotone*) problems rather than strongly-convex-strongly-concave (respectively, convex-concave) problems. Notice that we only assume that the full saddle gradient $\boldsymbol{F}$ is (strongly) monotone, not the individual $\boldsymbol{F}_i$'s.

In addition, we remark that our convergence analysis under the monotonicity of $\boldsymbol{F}$, Theorem 5.4, in fact requires only a relaxed version of monotonicity, known as *star*-monotonicity. This condition imposes the inequality (3) with $\mu = 0$, but only when $\boldsymbol{w} = \boldsymbol{z}^*$, where $\boldsymbol{z}^*$ is a point such that $\boldsymbol{F}\boldsymbol{z}^* = \boldsymbol{0}$. This relaxation allows for a certain degree of nonconvex-nonconcavity in $f$. For a more detailed discussion on the star-monotonicity condition, see Appendix G.1.

Other three underlying assumptions we make on the problem (1) can be listed as follows.

**Assumption 3.2** (Existence of an Optimal Solution). An optimal solution of the problem (1), which is a point we denote by $\boldsymbol{z}^* = (\boldsymbol{x}^*, \boldsymbol{y}^*)$ that satisfies

$$f(\boldsymbol{x}^*, \boldsymbol{y}) \leq f(\boldsymbol{x}^*, \boldsymbol{y}^*) \leq f(\boldsymbol{x}, \boldsymbol{y}^*)$$

for any $\boldsymbol{x} \in \mathbb{R}^{d_1}$ and $\boldsymbol{y} \in \mathbb{R}^{d_2}$, exists in $\mathbb{R}^{d_1+d_2}$.

Because the problem is unconstrained and $f$ is convex-concave, a point $\boldsymbol{z}^*$ is an optimum if and only if $\boldsymbol{F}\boldsymbol{z}^* = \boldsymbol{0}$. For strongly monotone problems, Assumption 3.2 is not explicitly required, as it is guaranteed *a priori* [5, Proposition 22.11]. For monotone problems, we explicitly impose Assumption 3.2 in order to exclude pathological problems such as $f(x, y) = x - y$.

**Assumption 3.3** (Smoothness). Each $f_i$ is $L$-smooth, and each $\boldsymbol{F}_i$ is $M$-smooth. That is, for any $\boldsymbol{z}, \boldsymbol{w} \in \mathbb{R}^{d_1+d_2}$,

   (i) $\|\boldsymbol{F}_i\boldsymbol{z} - \boldsymbol{F}_i\boldsymbol{w}\| \leq L \|\boldsymbol{z} - \boldsymbol{w}\|$,

   (ii) $\|D\boldsymbol{F}_i\boldsymbol{z} - D\boldsymbol{F}_i\boldsymbol{w}\| \leq M \|\boldsymbol{z} - \boldsymbol{w}\|$.

It is worth mentioning that the gradient operator $\boldsymbol{F}_i$ arising from a quadratic function $f_i$ is $M$-smooth with $M = 0$. Notice also that, by the finite-sum structure $\boldsymbol{F} = \frac{1}{n} \sum_{i=1}^{n} \boldsymbol{F}_i$, it is clear that Assumption 3.3 implies $f$ being $L$-smooth and $\boldsymbol{F}$ being $M$-smooth.

The $L$-smoothness assumption on the objective functions is standard in the optimization literature, while the $M$-smoothness assumption on the saddle gradients may look less standard. This smoothness assumption on the saddle gradient, in other words the Lipschitz Hessian condition, for analyzing SEG-FFA stems from the analysis of the flip-flop sampling scheme [46]. In particular, this is needed for bounding the high-order error terms between the (deterministic) EG and SEG-FFA in Section 5.1. The existing analysis of flip-flop sampling [46] is limited to quadratic functions that trivially have 0-Lipschitz Hessians ($M = 0$), so our analysis is a step forward.

**Assumption 3.4** (Component Variance). There exist constants $\rho \geq 0$ and $\sigma \geq 0$ such that

$$\frac{1}{n} \sum_{i=1}^{n} \|\boldsymbol{F}_i\boldsymbol{z} - \boldsymbol{F}\boldsymbol{z}\|^2 \leq (\rho \|\boldsymbol{F}\boldsymbol{z}\| + \sigma)^2 \qquad \forall \boldsymbol{z}. \tag{4}$$

For strongly monotone problems, Assumption 3.4 is not explicitly required, because it can be obtained as a consequence of the preceding assumptions: see Lemma C.9. Nevertheless, for convenience, we will keep the notations $\rho$ and $\sigma$ as in (4) for the strongly monotone setting as well.

In many existing works on stochastic optimization methods for minimax problems, Assumption 3.4 with $\rho = 0$ is imposed. This *uniform* bound on the variance simplifies the convergence analyses, but it is also fairly restrictive especially in the unconstrained settings. Already for bilinear finite-sum minimax problems $f(\boldsymbol{x}, \boldsymbol{y}) = \frac{1}{n} \sum_{i=1}^{n} \boldsymbol{x}^\top \boldsymbol{B}_i \boldsymbol{y}$, one can easily check that setting $\rho = 0$ forces the matrices $\boldsymbol{B}_i$ to be exactly equal to each other. For machine learning applications, it has been also reported that the assumption with $\rho = 0$ often fails to hold [7]. Therefore, allowing the variance to grow with the gradient $\boldsymbol{F} \boldsymbol{z}$ makes the assumption much more realistic.

The Lipschitz Hessian condition and the component variance assumption for monotone problems may still look rather strong. We leave the study on how one can relax such assumptions to prove *upper* bounds for convergence rates as an interesting future direction. On the other hand, while our lower bound results in Theorems 4.1 and 5.6 are derived under those strong assumptions, they still serve as lower bound results also for larger function classes that do not have those assumptions. In other words, the value of those results are not limited because of those assumptions being imposed.

## 4 Shuffling Alone Is Not Enough

Under the settings we have discussed, we study the SEG with shuffling-based sampling schemes. First we describe the precise methods of our consideration, namely the SEG-RR and SEG-FF.

For $k \geq 0$, in the beginning of an epoch, a random permutation $\tau_k$ is sampled from a uniform distribution over $\mathcal{S}_n$. Then, for $n$ iterations, we use each of the component functions once, in the order determined by $\tau_k$. That is, for $i = 0, 1, \ldots, n-1$ we do

$$
\begin{aligned}
\boldsymbol{w}_i^k &\leftarrow \boldsymbol{z}_i^k - \alpha_k \boldsymbol{F}_{\tau_k(i+1)} \boldsymbol{z}_i^k, \\
\boldsymbol{z}_{i+1}^k &\leftarrow \boldsymbol{z}_i^k - \beta_k \boldsymbol{F}_{\tau_k(i+1)} \boldsymbol{w}_i^k,
\end{aligned}
\tag{5}
$$

for some stepsizes $\alpha_k$ and $\beta_k$. In case of SEG-RR, the epoch is completed here, and we set $\boldsymbol{z}_0^{k+1} \leftarrow \boldsymbol{z}_n^k$ as the initial point for the next epoch.

In case of SEG-FF, we additionally perform $n$ more iterations in the epoch, as proposed in Rajput et al. [46]. In these additional iterations, the component functions are each used once more, but in the reverse order. That is, for $i = n, n+1, \ldots, 2n-1$, we do

$$
\begin{aligned}
\boldsymbol{w}_i^k &\leftarrow \boldsymbol{z}_i^k - \alpha_k \boldsymbol{F}_{\tau_k(2n-i)} \boldsymbol{z}_i^k, \\
\boldsymbol{z}_{i+1}^k &\leftarrow \boldsymbol{z}_i^k - \beta_k \boldsymbol{F}_{\tau_k(2n-i)} \boldsymbol{w}_i^k.
\end{aligned}
\tag{6}
$$

Then we set $\boldsymbol{z}_0^{k+1} \leftarrow \boldsymbol{z}_{2n}^k$ as the initial point for the next epoch. The full pseudocode of these methods can be found in Appendix A.

When $\boldsymbol{F}$ is strongly monotone, it is possible to show that both SEG-RR and SEG-FF indeed provide speed-up over SEG-US. The well-known rate of SEG-US under strong monotonicity of $\boldsymbol{F}$ is $\Theta(1/T)$, where $T$ is the total number of iterations [6, 20]. Translating this rate to our shuffling-based setting, where there are $\Theta(n)$ iterations per epoch, this rate amounts to $\Theta(1/nK)$. Recently, Emmanouilidis et al. [18] have shown that SEG-RR, under the same setting as ours, attains a convergence rate of $\tilde{\mathcal{O}}(1/nK^2)$, on par with the rate of SGDA-RR [15]. In Appendix F, we also show that SEG-FF attains a similar rate of convergence.

However, it turns out that the benefit of shuffling does not extend further beyond the strongly monotone setting. In fact, when $\boldsymbol{F}$ is merely monotone, then in the worst case, SEG-RR and SEG-FF suffers from nonconvergence, just as in the case of SEG-US.

**Theorem 4.1.** *For $n = 2$, there exists a minimax problem with $f(x, y) = \frac{1}{2} \sum_{i=1}^{2} f_i(x, y)$ having a monotone $\boldsymbol{F}$, consisting of $L$-smooth quadratic $f_i$'s satisfying Assumption 3.4 with $(\rho, \sigma) = (1, 0)$, such that SEG-US, SEG-RR and SEG-FF diverge in expectation for any positive stepsizes.*

We provide the explicit counterexample and the proof of divergence in Appendix H.1. Note that Theorem 4.1 and its proof in Appendix H.1 imply that $\min_{t=0,\ldots,T} \mathbb{E}[\|\boldsymbol{F}\boldsymbol{z}_t\|^2] = \Omega(1)$ for SEG-US and $\min_{k=0,\ldots,K} \mathbb{E}[\|\boldsymbol{F}\boldsymbol{z}_0^k\|^2] = \Omega(1)$ for SEG-RR and SEG-FF, as summarized in Table 1.

# 5 SEG-FFA: SEG with Flip-Flop Anchoring

In this section, we investigate the underlying cause for nonconvergence of SEG-RR and SEG-FF from the perspective of how accurately they match the convergent EG or PP methods in terms of the Taylor expansions of updates. We then propose adding a simple *anchoring* step at the end of each epoch of SEG-FF. It turns out that adding the *anchoring* step, which is a step of taking a convex combination of an iterate with a previously computed iterate, reduces the stochastic noise and leads to a method with improved convergence properties.

## 5.1 Design Principle: Second-Order Matching

As observed by [38], the key feature of EG behind its superior convergence properties compared to GDA is its update rule closely resembling PP, while the "error" of GDA as an approximation of PP is so large that it hinders convergence. The difference between the updates of EG and PP, in the Taylor expansion, is as small as $\mathcal{O}(\eta^3)$ per iteration, where $\eta$ is the stepsize. On the other hand, GDA and PP show a difference of $\mathcal{O}(\eta^2)$, and this greater "error" explains why GDA diverges while EG and PP converge. Of course, EG and PP are not the only two algorithms that converge in the monotone setting; let us recall the update rule of EG+ method [17], and Taylor-expand it as the following:

$$
\begin{aligned}
z^+ &:= z - \eta_2 F(z - \eta_1 Fz) \\
&= z - \eta_2 Fz + \eta_1 \eta_2 DF(z) Fz + O(\eta_1^2 \eta_2).
\end{aligned}
\tag{7}
$$

EG+ is known to converge for unconstrained monotone problems if $\eta_1 \geq \eta_2$. When $\eta_1 = \eta_2$, it recovers EG and matches PP up to second-order terms.

Based on these observations, we now state our key principle for designing a convergent version of SEG: *second-order matching*. We would like to choose proper stepsizes, sampling scheme, and anchoring scheme so that our without-replacement SEG can *deterministically* match the update equation of a convergent algorithm (EG/PP or EG+) up to the $\mathcal{O}(\eta^2)$ terms (i.e., *second-order* terms in the Taylor expansion), thereby satisfying a small $\mathcal{O}(\eta^3)$ approximation error. We show that **(a)** this *second-order matching* can be achieved with *flip-flop anchoring*, but not solely by permutation-based sampling such as RR and flip-flop (without anchoring), and **(b)** second-order matching indeed grants convergence for monotone problems. In particular, we demonstrate that

1. SEG-RR suffers a poor approximation error of $\mathcal{O}(\eta^2)$ as an approximation of EG/EG+.

2. SEG-FF can match EG+ up to second-order terms, but it results in a choice of stepsizes ($\eta_2 = 2\eta_1$) that make EG+ diverge (Proposition 5.2).

3. SEG-FFA, the method we propose, matches EG up to second-order terms to get an error of $\mathcal{O}(\eta^3)$ (Proposition 5.3), achieving convergence in monotone problems (Theorem 5.4).

To this end, let us consider a general form of SEG that incorporates any arbitrary sampling scheme. More precisely, in the $k$-th "epoch" consisted of $N$ iterations, the components are chosen in the order of $T_0^k, T_1^k, \cdots, T_{N-1}^k$, where $T_i^k \in \{F_1, \ldots, F_n\}$ for each $i$. For our purpose, we assume that $N$ is some multiple of $n$ (e.g., $N = n$ for SEG-RR, $N = 2n$ for SEG-FF). Then, given $\alpha$ and $\beta$ we perform SEG updates, for $i = 0, 1, \ldots, N-1$,

$$
\begin{aligned}
w_i^k &\leftarrow z_i^k - \alpha T_i^k z_i^k, \\
z_{i+1}^k &\leftarrow z_i^k - \beta T_i^k w_i^k.
\end{aligned}
\tag{8}
$$

### 5.1.1 Necessity of Flip-Flop Sampling

The general method in (8) that sets the initial point for the next epoch as $z_0^{k+1} \leftarrow z_N^k$ satisfies the following property.

**Proposition 5.1.** *Suppose that Assumption 3.3 holds. For some $\epsilon_N^k = o\big((\alpha + \beta)^2\big)$, it holds that*

$$
z_0^{k+1} = z_0^k - \beta \sum_{j=0}^{N-1} T_j^k z_0^k + \alpha\beta \sum_{j=0}^{N-1} DT_j^k(z_0^k) T_j^k z_0^k + \beta^2 \sum_{i<j} DT_j^k(z_0^k) T_i^k z_0^k + \epsilon_N^k.
\tag{9}
$$

See Appendix D.1 for the proof. To make (7) and (9) match up to the second-order, both the equations

$$\frac{\eta_2}{n} \sum_{j=1}^{n} \boldsymbol{F}_i \boldsymbol{z}_0^k = \beta \sum_{j=0}^{N-1} \boldsymbol{T}_j^k \boldsymbol{z}_0^k \qquad \text{and} \tag{10}$$

$$\frac{\eta_1 \eta_2}{n^2} \Big( \sum_{j=1}^{n} D\boldsymbol{F}_j(\boldsymbol{z}_0^k)\boldsymbol{F}_j \boldsymbol{z}_0^k + \sum_{i \neq j} D\boldsymbol{F}_j(\boldsymbol{z}_0^k)\boldsymbol{F}_i \boldsymbol{z}_0^k \Big) = \alpha\beta \sum_{j=0}^{N-1} D\boldsymbol{T}_j^k(\boldsymbol{z}_0^k)\boldsymbol{T}_j^k \boldsymbol{z}_0^k + \beta^2 \sum_{i<j} D\boldsymbol{T}_j^k(\boldsymbol{z}_0^k)\boldsymbol{T}_i^k \boldsymbol{z}_0^k \tag{11}$$

must hold. Clearly, without-replacement sampling will make (10) hold. However, it is easy to check that random reshuffling falls short of making (11) hold. This is because, if RR is used, then $\boldsymbol{T}_0^k, \boldsymbol{T}_1^k, \dots, \boldsymbol{T}_{n-1}^k$ is nothing but a reordering of $\boldsymbol{F}_1, \dots, \boldsymbol{F}_n$ into $\boldsymbol{F}_{\tau(1)}, \dots, \boldsymbol{F}_{\tau(n)}$, so the RHS of (11) can only contain terms $D\boldsymbol{F}_{\tau(j)}(\boldsymbol{z}_0^k)\boldsymbol{F}_{\tau(i)}\boldsymbol{z}_0^k$ with $i \leq j$. This observation motivates the use of flip-flop sampling, because choosing $\boldsymbol{T}_i^k = \boldsymbol{T}_{2n-1-i}^k$ lets all the required terms $D\boldsymbol{F}_j(\boldsymbol{z}_0^k)\boldsymbol{F}_i \boldsymbol{z}_0^k$ to appear in the RHS of (11).

### 5.1.2 Designing SEG-FFA

Flip-flop does resolve the aforesaid issue, but still another complication remains for plain SEG-FF.

**Proposition 5.2.** *Suppose we use flip-flop sampling (without anchoring). In order to make* (10) *and* (11) *hold, we must choose* $\beta = \eta_1/n$ *and* $\alpha = \beta/2$. *However, this leads to* $\eta_2 = 2\eta_1$, *which is the set of parameters that fails to make EG+ converge.*

For the proof, see Appendix D.2. This shows that a modification is necessary to develop a stochastic method that achieves second-order matching to *convergent* EG/EG+ methods.

We thus propose to add an *anchoring* step:

$$\boldsymbol{z}_0^{k+1} \leftarrow \tfrac{1}{2}\left(\boldsymbol{z}_N^k + \boldsymbol{z}_0^k\right), \tag{12}$$

after finishing the $N$ updates (8), instead of $\boldsymbol{z}_0^{k+1} \leftarrow \boldsymbol{z}_N^k$. This is our *Stochastic ExtraGradient with Flip-Flop Anchoring* (SEG-FFA) method, named after the design of combining the flip-flop sampling scheme and the anchoring step. We note that this idea of taking a convex combination has originally appeared in the Krasnosel'skiĭ-Mann iteration [30, 33], and also under the name of *Lookahead* methods [12, 43]. This slightly differs from the more widely used *Halpern iteration* [23] based anchoring (*cf.* [54]), which would have used the initial point $\boldsymbol{z}_0^0$ instead of $\boldsymbol{z}_0^k$ in (12).

This anchoring step changes (9) accordingly, and essentially amounts to dividing the right-hand sides of (10) and (11) each by 2 (see Appendix D for the detailed derivations). We show that choosing $\alpha = \beta/2$ in fact leads to the second-order matching to EG, i.e., EG+ with $\eta_1 = \eta_2$.

**Proposition 5.3.** *Suppose that Assumptions 3.3 and 3.4 hold. Then, for* $\beta_k = \eta$ *and* $\alpha_k = \beta_k/2$, *SEG-FFA becomes an approximation of EG with error at most* $\mathcal{O}(\eta^3)$. *In other words, we achieve*

$$\left\| \boldsymbol{z}_0^k - \eta n \boldsymbol{F}(\boldsymbol{z}_0^k - \eta n \boldsymbol{F}\boldsymbol{z}_0^k) - \boldsymbol{z}_0^{k+1} \right\| = \mathcal{O}(\eta^3).$$

In other words, adding the anchoring step allows us to get a method that well approximates the convergent EG with an error as small as $\mathcal{O}(\eta^3)$. For a more in-depth discussion, see Appendix E.

### 5.2 Convergence Analysis of SEG-FFA

As a result of the second-order matching, we obtain SEG-FFA, a stochastic method that has an error of $\mathcal{O}(\eta^3)$ as an approximation of EG. Achieving this order of magnitude for the approximation error turns out to be the key to the exact convergence to an optimum under the monotone setting.

**Theorem 5.4.** *Suppose that* $\boldsymbol{F}$ *is (star-)monotone, Assumptions 3.2, 3.3, and 3.4 hold, and we are running SEG-FFA. Then, for any* $K \geq 1$, *by choosing the stepsizes sufficiently small and decaying as* $\beta_k = \mathcal{O}(1/k^{1/3}\log k)$ *and* $\alpha_k = \beta_k/2$, *the iterates generated by SEG-FFA achieves the bound*

$$\min_{k=0,1,\dots,K} \mathbb{E}\left\|\boldsymbol{F}\boldsymbol{z}_0^k\right\|^2 = \mathcal{O}\left(\frac{(\log K)^2}{K^{1/3}}\right).$$

For the full statement of the theorem and its proof, see Appendix G. We note that, although Theorem 5.4, and also Theorem 5.5 below, are stated specifically for SEG-FFA, our analyses show that both theorems can be applied to any method that achieves the second-order matching in terms of Proposition 5.3.

The reduced error also shows a gain in the rate of convergence under the strongly monotone setting. This aligns with the intuition that error hinders convergence, hence having a smaller error is beneficial.

**Theorem 5.5.** *Suppose that $F$ is $\mu$-strongly monotone with $\mu > 0$ and Assumption 3.3 holds. Then, there exists a choice of $\eta > 0$ such that, when SEG-FFA is run for $K$ epochs with constant stepsizes $\beta_k = \eta$ and $\alpha_k = \eta/2$, for some constant $\omega$ independent of $\eta$, the iterates generated by SEG-FFA achieves the bound*

$$\mathbb{E}\left\|\boldsymbol{z}_0^K - \boldsymbol{z}^*\right\|^2 \leq \exp\left(-\frac{1}{2}\mu\omega nK\right)\left\|\boldsymbol{z}_0^0 - \boldsymbol{z}^*\right\|^2 + \mathcal{O}\left(\frac{\left(\log(n^{1/4}K)\right)^4}{nK^4}\right).$$

Theorem 5.5 actually stems from a unified analysis that encompasses all the shuffling-based SEG methods introduced in this paper, including SEG-RR and SEG-FF. See Appendix F for the details.

Notice the exponent 4 of the number of epochs $K$ in the convergence rate, which is twice as large as the exponent 2 of SGDA-RR and SEG-RR. In fact, this gain in the rate of convergence turns out to be fundamental. As we show in the following theorem, the theoretical lower bounds of convergence for SGDA-RR and SEG-RR with constant stepsize are both $\Omega(1/nK^3)$. This exhibits that there is a *provable gap* between those methods and SEG-FFA, which attains $\tilde{\mathcal{O}}(1/nK^4)$.

**Theorem 5.6.** *Suppose $n \geq 2$. For both SGDA-RR with constant stepsize $\alpha_k = \alpha > 0$ and SEG-RR with constant stepsize $\alpha_k = \alpha > 0$, $\beta_k = \beta > 0$, there exists a $\mu$-strongly monotone minimax problem $f(\boldsymbol{z}) = \frac{1}{n}\sum_{i=1}^n f_i(\boldsymbol{z})$ with $\mu > 0$ such that regardless of stepsizes, we have*

$$\mathbb{E}\left[\left\|\boldsymbol{z}_0^K - \boldsymbol{z}^*\right\|^2\right] = \begin{cases} \Omega\left(\frac{\sigma^2}{L\mu nK}\right) & \text{if } K \leq L/\mu, \\ \Omega\left(\frac{L\sigma^2}{\mu^3 nK^3}\right) & \text{if } K > L/\mu. \end{cases}$$

*Proof.* The full statement and the proof are presented in Appendix H.3. □

## 6 Experiments

We consider randomly generated quadratic problems of the form

$$\min_{\boldsymbol{x}\in\mathbb{R}^{d_x}}\max_{\boldsymbol{y}\in\mathbb{R}^{d_y}}\frac{1}{n}\sum_{i=1}^n\begin{bmatrix}\boldsymbol{x}\\\boldsymbol{y}\end{bmatrix}^\top\begin{bmatrix}\boldsymbol{A}_i & \boldsymbol{B}_i\\\boldsymbol{B}_i^\top & -\boldsymbol{C}_i\end{bmatrix}\begin{bmatrix}\boldsymbol{x}\\\boldsymbol{y}\end{bmatrix} - \boldsymbol{t}_i^\top\begin{bmatrix}\boldsymbol{x}\\\boldsymbol{y}\end{bmatrix}. \tag{13}$$

In particular, we sample the random components so that the full objective is either monotone or strongly monotone, respectively, while each of the components may be nonmonotone. For the exact descriptions on how we constructed the problems, see Appendix I.1.

**Monotone Case** We ran the experiment on 5 random instances of (13) with the stepsizes scheduled as $\eta_k = \eta_0/(1+k/10)^{0.34}$ where $\eta_0 = \min\{0.01, \frac{1}{L}\}$ for SEG-FFA, and $\alpha_k = \beta_k = \eta_k$ for SEG-US, SEG-RR, and SEG-FF. The exponent $0.34$ is to ensure a sufficient decay rate required by Theorem 5.4, and the convergence of SEG-FFA under such a stepsize scheduling is validated in Remark G.5. The value of $\eta_0$ is, however, a heuristically determined small number. The results of the geometric mean over the 5 runs are plotted in Figure 1. As expected by our theory, SEG-FFA successfully shows convergence, while all of SEG-FF, SEG-RR, and SEG-US diverge in the long run.

**Strongly monotone case** Along with the variants of SEG, we also compare the performances of SGDA-RR and SGDA-US. We ran the experiment on 5 random instances of (13) with stepsizes $\eta_k = 0.001$, and the results are plotted in Figure 1. Additional results obtained from using other stepsizes can be found in Appendix I.4. We again observe an agreement between the empirical results and our theory; SEG-FFA eventually finds the point with the smallest gradient norm among the methods that are considered.

Further additional experiments and ablation studies we have conducted can be found in Appendix I.

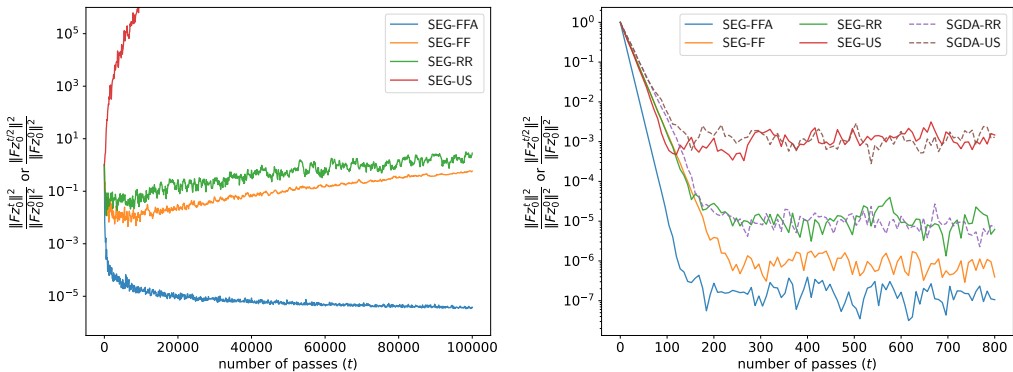

Figure 1: Experimental results on the (left) monotone and (right) strongly monotone examples, comparing the variants of SEG. For a fair comparison, we take the number of passes over the full dataset as the abscissae. In other words, we plot $\|\boldsymbol{F z}_0^{t/2}\|^2/\|\boldsymbol{F z}_0^0\|^2$ for SEG-FFA and SEG-FF, as they pass through the whole dataset twice every epoch, and $\|\boldsymbol{F z}_0^{t}\|^2/\|\boldsymbol{F z}_0^0\|^2$ for the other methods, as they pass once every epoch.

## 7   Conclusion

We proposed SEG-FFA, a new stochastic variant of EG that uses flip-flop sampling and anchoring. While being a minimal modification from the vanilla SEG, SEG-FFA attains the crucial "second-order matching" property to the deterministic EG, leading to a two-fold improved convergence. On one hand, SEG-FFA reaches an optimum in the monotone setting, unlike many baseline methods such as SEG-US, SEG-RR, and SEG-FF that diverge. Moreover, in the strongly monotone setting, SEG-FFA shows a faster convergence with a provable gap from the other methods.

An interesting future direction would be to extend our work to more general nonconvex-nonconcave problems, further exploring the potentials of the second-order matching technique. It would also be appealing to further study whether it is possible to devise a new method that achieves second-order (or higher) matching without the anchoring step, potentially enhancing our understanding of the effectiveness of the matching technique.

## Acknowledgments and Disclosure of Funding

This work was supported in part by the National Research Foundation of Korea (NRF) grant funded by the Korea government (MSIT) (No. RS-2019-NR040050). JC and DK acknowledge support from the NRF grant (No. RS-2022-NR071715) funded by the Korea government (MSIT), and the Samsung Science & Technology Foundation grant (No. SSTF-BA2101-02). CY acknowledges support from the NRF grant (No. RS-2023-00211352) funded by the Korea government (MSIT).

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

# Contents

# A    Pseudocode of the Algorithms

We present the pseudocode of the algorithms we consider in this paper in Algorithms 2, 3 and 4, with the pseudocode of the with-replacement stochastic methods in Algorithm 1.

---

**Algorithm 1** SEG-US / SGDA-US

---

**Input:** The number of components $n$; stepsize sequences $\{\alpha_t\}_{t\geq 0}$ and $\{\beta_t\}_{t\geq 0}$
**Initialize:** $\boldsymbol{z}_0 \in \mathbb{R}^{d_1+d_2}$
**for** $t = 0, 1, \ldots$ **do**
    sample $i(t)$ uniformly from $\{1, \ldots, n\}$
    **if** SGDA-US **then**
        $\boldsymbol{z}_{t+1} \leftarrow \boldsymbol{z}_t - \alpha_t \boldsymbol{F}_{i(t)} \boldsymbol{z}_t$
    **else if** SEG-US **then**
        $\boldsymbol{w}_t \leftarrow \boldsymbol{z}_t - \alpha_t \boldsymbol{F}_{i(t)} \boldsymbol{z}_t$
        $\boldsymbol{z}_{t+1} \leftarrow \boldsymbol{z}_t - \beta_t \boldsymbol{F}_{i(t)} \boldsymbol{w}_t$
    **end if**
**end for**

---

**Algorithm 2** SEG-RR / SGDA-RR

---

**Input:** The number of components $n$; stepsize sequences $\{\alpha_k\}_{k\geq 0}$ and $\{\beta_k\}_{k\geq 0}$
**Initialize:** $\boldsymbol{z}_0^0 \in \mathbb{R}^{d_1+d_2}$
**for** $k = 0, 1, \ldots$ **do**
    sample $\tau_k$ uniformly from $\mathcal{S}_n$
    **for** $i = 0$ **to** $n - 1$ **do**
        **if** SGDA-RR **then**
            $\boldsymbol{z}_{i+1}^k \leftarrow \boldsymbol{z}_i^k - \alpha_k \boldsymbol{F}_{\tau_k(i+1)} \boldsymbol{z}_i^k$
        **else if** SEG-RR **then**
            $\boldsymbol{w}_i^k \leftarrow \boldsymbol{z}_i^k - \alpha_k \boldsymbol{F}_{\tau_k(i+1)} \boldsymbol{z}_i^k$
            $\boldsymbol{z}_{i+1}^k \leftarrow \boldsymbol{z}_i^k - \beta_k \boldsymbol{F}_{\tau_k(i+1)} \boldsymbol{w}_i^k$
        **end if**
    **end for**
    $\boldsymbol{z}_0^{k+1} \leftarrow \boldsymbol{z}_n^k$
**end for**

---

**Algorithm 3** SEG-FF

---

**Input:** The number of components $n$; stepsize sequences $\{\alpha_k\}_{k\geq 0}$ and $\{\beta_k\}_{k\geq 0}$
**Initialize:** $\boldsymbol{z}_0^0 \in \mathbb{R}^{d_1+d_2}$
**for** $k = 0, 1, \ldots$ **do**
    sample $\tau_k$ uniformly from $\mathcal{S}_n$
    **for** $i = 0$ **to** $n - 1$ **do**
        $\boldsymbol{w}_i^k \leftarrow \boldsymbol{z}_i^k - \alpha_k \boldsymbol{F}_{\tau_k(i+1)} \boldsymbol{z}_i^k$
        $\boldsymbol{z}_{i+1}^k \leftarrow \boldsymbol{z}_i^k - \beta_k \boldsymbol{F}_{\tau_k(i+1)} \boldsymbol{w}_i^k$
    **end for**
    **for** $i = n$ **to** $2n - 1$ **do**
        $\boldsymbol{w}_i^k \leftarrow \boldsymbol{z}_i^k - \alpha_k \boldsymbol{F}_{\tau_k(2n-i)} \boldsymbol{z}_i^k$
        $\boldsymbol{z}_{i+1}^k \leftarrow \boldsymbol{z}_i^k - \beta_k \boldsymbol{F}_{\tau_k(2n-i)} \boldsymbol{w}_i^k$
    **end for**
    $\boldsymbol{z}_0^{k+1} \leftarrow \boldsymbol{z}_{2n}^k$
**end for**

---

---
**Algorithm 4** SEG-FFA
---
**Input:** The number of components $n$; stepsize sequences $\{\eta_k\}_{k \geq 0}$
**Initialize:** $\boldsymbol{z}_0^0 \in \mathbb{R}^{d_1 + d_2}$
**for** $k = 0, 1, \ldots$ **do**
    sample $\tau_k$ uniformly from $\mathcal{S}_n$
    **for** $i = 0$ **to** $n - 1$ **do**
        $\boldsymbol{w}_i^k \leftarrow \boldsymbol{z}_i^k - \frac{\eta_k}{2} \boldsymbol{F}_{\tau_k(i+1)} \boldsymbol{z}_i^k$
        $\boldsymbol{z}_{i+1}^k \leftarrow \boldsymbol{z}_i^k - \eta_k \boldsymbol{F}_{\tau_k(i+1)} \boldsymbol{w}_i^k$
    **end for**
    **for** $i = n$ **to** $2n - 1$ **do**
        $\boldsymbol{w}_i^k \leftarrow \boldsymbol{z}_i^k - \frac{\eta_k}{2} \boldsymbol{F}_{\tau_k(2n-i)} \boldsymbol{z}_i^k$
        $\boldsymbol{z}_{i+1}^k \leftarrow \boldsymbol{z}_i^k - \eta_k \boldsymbol{F}_{\tau_k(2n-i)} \boldsymbol{w}_i^k$
    **end for**
    $\boldsymbol{z}_0^{k+1} \leftarrow \frac{\boldsymbol{z}_0^k + \boldsymbol{z}_{2n}^k}{2}$
**end for**
---

# B  Further Details and Discussions on the Related Works

## B.1  A Summary of the Limitations of the Existing Works in the Monotone Setting

In Table 2, we have summarized the settings considered in each of the previous works on stochastic variants of EG discussed in Section 2, and compare them with our settings. Please note that we focus on the monotone $\boldsymbol{F}$ setting in the table. Entries that are worth further discussions are marked, with the corresponding explanations below.

Table 2: Comparison of the underlying settings between ours and the existing works.

|  | same sample? | required batch size | bounded domain? | uniform gradient variance? | monotone components? |
|---|---|---|---|---|---|
| Ours | ✓ | constantly 1 | ✗ | ✗ | ✗ |
| Cai et al. [9]* | N/A | increasing | ✗ | ✓ | ✗ |
| Choudhury et al. [14]* | N/A | increasing | ✗ | ✗ | ✗ |
| Diakonikolas et al. [17] | ✗ | increasing‡ | ✗ | ✓ | ✗ |
| Gorbunov et al. [20] | ✓ | increasing‡ | ✗ | ✗ | (star-)✓† |
|  | ✗ | increasing‡ | ✗ | ✓ | ✗ |
| Hsieh et al. [25]§ | ✗ | constant | ✗ | ✗ | ✗ |
| Juditsky et al. [27] | ✗ | constant | ✓ | ✓ | ✗ |
| Mishchenko et al. [36] | ✓ | constant | ✓¶ | ✗¶ | ✓ |
| Pethick et al. [42] | ✗ | constant | ✗ | ✓ | ✗ |

**(∗)**  The methods proposed in these works are not stochastic variants of EG in a strict sense. The method introduced by Cai et al. [9] is rather a hybrid of EG and the Halpern iteration [23], while the method by Choudhury et al. [14] is a stochastic version of the so-called *optimistic gradient* method [44]. Hence, determining whether these methods fall into the category of same-sample methods or not is unnecessary. Nonetheless, as these works focus on solving a similar problem to ours, we include them as references.

**(†)**  Under the assumptions that Gorbunov et al. [20] make in their paper, one can show that each of the components must necessarily be (star-)monotone when the full $\boldsymbol{F}$ is (star-)monotone. For further explanations on why this is the case, see the following Appendix B.2.

**(‡)**  Yet, to be precise, what Gorbunov et al. [20] have shown in the monotone case is that SEG-US can find an optimal solution if we *increase* the batch size each iteration. If the batch size is fixed as a constant, then they were only able to show that the iterates will be *bounded* in the (star-)monotone setting. In particular, they did not provide a guarantee that the iterates will be necessarily convergent.

In fact, as we demonstrate with an explicit counterexample in Appendix H.2, if we do not increase the batch size each iteration, then it is possible to show that SEG-US in the worst case will *never*

converge to an optimal point. This nonconvergence result in fact holds for any SEG-US whose extrapolation and update stepsizes differ by a constant factor. Hence, it not only applies to [20], but also to [17].

(§) Hsieh et al. [25] show that *independent-sample* SEG-US converges for stepsizes $\alpha_t, \beta_t$ decaying at certain different rates, but gives no convergence rates.

(¶) Mishchenko et al. [36] assume a uniformly bounded gradient variance in the strongly monotone case. In the monotone case, the bound they derived depends on the supremum of the gradient variance over the domain that is under consideration. Hence, in the monotone case, either the domain has to be (implicitly) bounded, or the uniform gradient variance assumption should be imposed.

## B.2   On the Assumptions Made by Gorbunov et al. [20]

We would like to first clarify that in [20], the requirement to increase the batch size is utilized only in the monotone setting: see, *e.g.*, Corollary E.4 therein.

Gorbunov et al. [20] use a generalized notion of $\mu$-strong monotonicity, namely the $\mu$-*quasi strong monotonicity*, which requires the operator $\boldsymbol{F}$ to satisfy

$$\langle \boldsymbol{F}z, z - z^* \rangle \geq \mu \| z - z^* \|^2 . \tag{14}$$

In the notion of $\mu$-quasi strong monotonicity they also allow $\mu \leq 0$. In particular, if (14) holds with $\mu = 0$, then $\boldsymbol{F}$ is called a *star-monotone* operator. In Appendix G.1 we further discuss on star-monotone operators.

Meanwhile, let us further elaborate on why in the (star-)monotone setting, the assumptions made by the authors of [20] lead to each component being star-monotone. In their work the authors require, as equation (10) therein, that

$$\frac{1}{n} \sum_{i:\mu_i \geq 0} \mu_i + \frac{4}{n} \sum_{i:\mu_i < 0} \mu_i \geq 0. \tag{15}$$

Observe that this amounts to

$$\mu := \frac{1}{n} \sum_{i=1}^{n} \mu_i \geq -\frac{3}{n} \sum_{i:\mu_i < 0} \mu_i = \frac{3}{n} \sum_{i:\mu_i < 0} |\mu_i| . \tag{16}$$

However, if any of $\mu_i$ is strictly negative, then the rightmost sum in (16) becomes strictly positive, hence cannot be less than or equal to $\mu$ if $\mu = 0$. Therefore, the only possible case is when the rightmost sum is an empty sum. In other words, (15) can hold with $\mu = 0$ only when $\mu_i \geq 0$ for all $i$, so that each $\boldsymbol{F}_i$ is star-monotone. We would like to remind the readers that our analyses, on the other hand, do not have any restrictions on the individual components.

## B.3   Finite Sum Structure vs. General Stochastic Setting

The works mentioned in Section 2 usually assume that we have access to a stochastic oracle that returns a stochastic estimator of $\boldsymbol{F}$. Indeed, having a finite sum structure is a special case of having a stochastic oracle, as each $\boldsymbol{F}_i$ can be seen as an estimator of $\boldsymbol{F}$. One might then ask whether assuming the finite sum structure can help the works mentioned in Section 2 overcome the mentioned limitations. We strongly believe that this is not the case. Recall Theorem 4.1, where we have constructed an explicit counterexample that SEG-US, SEG-RR, and SEG-FF all diverge. Because the set of problems with a finite sum structure is a subset of the set of problems with a stochastic oracle, the (counter-)example in Theorem 4.1 also works as an example that displays the nonconvergence of SEG-US, SEG-RR, and SEG-FF in the general stochastic setting. That is, a variant of SEG that only modifies the stepsizes and/or the sampling scheme into a without-replacement based one will suffer from nonconvergence, due to the counterexample in Theorem 4.1. It is also true that there are some methods that cannot exactly be classified as one of SEG-US, SEG-RR, or SEG-FF, but this counterexample demonstrates that, unless explicitly proven otherwise, there is not a good reason to believe that the existing convergence analyses will be easily extended beyond the assumptions they are each based on.

### B.4 On the Claimed Convergence of SEG-RR in the Monotone Setting by Emmanouilidis et al. [18]

Recently, a paper focusing on the study of SEG-RR [18] has been published. As we have briefly introduced in Table 1 with a discussion in Section 4, the authors have established a convergence rate $\tilde{\mathcal{O}}(1/nK^2)$ of SEG-RR in the strongly monotone setting, using an independent analysis of ours.

On the other hand, the authors of [18] furthermore claim that SEG-RR is capable of finding an optimum in the monotone setting, which is seemingly contradictory to our analyses. We assert that this is not the case, as their proof, at least in their AISTATS 2024 version, seems to have a flaw.

In establishing equation (85) in [18], the authors claim that the inequality

$$\frac{1}{K} \sum_{k=0}^{K} \frac{1}{G^k} \, \mathbb{E}\left[\|\boldsymbol{F}(\boldsymbol{z}_0^k)\|\right] \geq \mathbb{E}\left[\left\|\boldsymbol{F}\left(\frac{1}{K} \sum_{k=0}^{K} \frac{1}{G^k} \boldsymbol{z}_0^k\right)\right\|^2\right]$$

holds by Jensen's inequality, where $G \geq 6$ is a fixed constant. However, Jensen's inequality cannot be applied here, because not only $\|\boldsymbol{F}(\cdot)\|^2$ is possibly nonconvex, but also the weights multiplied to the iterates, namely $1/KG^k$, do not sum up to 1. Hence, the "averaged" iterate is not in the form of a convex combination. So, even if $\|\boldsymbol{F}(\cdot)\|^2$ was convex, if we were to properly apply Jensen's inequality, at least the averaged iterate should be multiplied by $\frac{1}{\sum_{k=0}^{K} 1/G^k}$ instead of $\frac{1}{K}$. Yet then, the sum $\sum_{k=0}^{K} \frac{1}{G^k} \leq \frac{G}{G-1}$ is bounded above by a constant independent of $K$, and the right hand side of the equation right above (85) in [18] shall no longer be divided by $K$. Therefore, their claimed convergence is unobtainable.

We would also like to remark that the linear decay rate of $1/G^k$ can make the series $\sum_{k=0}^{\infty} \frac{1}{G^k} \mathbb{E}[\|\boldsymbol{F}(\boldsymbol{z}_0^k)\|^2]$ convergent even when $\mathbb{E}[\|\boldsymbol{F}(\boldsymbol{z}_0^k)\|^2]$ grows exponentially as $k \to \infty$, as long as its rate of exponential growth is less than $G$. In particular, once their (85) is corrected, there is no contradiction with our divergence result in Theorem 4.1.

## C  Useful Lemmata

**Lemma C.1** (Polarization identity). *For any two vectors* $\boldsymbol{a}$ *and* $\boldsymbol{b}$, *it holds that*

$$2 \langle \boldsymbol{a}, \boldsymbol{b} \rangle = \|\boldsymbol{a}\|^2 + \|\boldsymbol{b}\|^2 - \|\boldsymbol{a} - \boldsymbol{b}\|^2$$
$$= \|\boldsymbol{a} + \boldsymbol{b}\|^2 - \|\boldsymbol{a}\|^2 - \|\boldsymbol{b}\|^2 .$$

*Proof.* The identities are immediate from $\|\boldsymbol{a} \pm \boldsymbol{b}\|^2 = \|\boldsymbol{a}\|^2 \pm 2 \langle \boldsymbol{a}, \boldsymbol{b} \rangle + \|\boldsymbol{b}\|^2$. $\qquad \square$

**Lemma C.2** (Weighted AM-GM inequality). *For any* $\gamma > 0$ *and two vectors* $\boldsymbol{a}$ *and* $\boldsymbol{b}$ *in* $\mathbb{R}^d$,

$$2 |\langle \boldsymbol{a}, \boldsymbol{b} \rangle| \leq \gamma \|\boldsymbol{a}\|^2 + \frac{1}{\gamma} \|\boldsymbol{b}\|^2 .$$

*Proof.* Notice that

$$2 |\langle \boldsymbol{a}, \boldsymbol{b} \rangle| \leq 2 \left(|a_1 b_1| + \cdots + |a_d b_d|\right)$$
$$\leq \left(\gamma a_1^2 + \frac{b_1^2}{\gamma}\right) + \cdots + \left(\gamma a_d^2 + \frac{b_d^2}{\gamma}\right) = \gamma \|\boldsymbol{a}\|^2 + \frac{1}{\gamma} \|\boldsymbol{b}\|^2 . \qquad \square$$

**Lemma C.3** (Young's inequality). *For any* $\gamma > 0$ *and two vectors* $\boldsymbol{a}$ *and* $\boldsymbol{b}$,

$$\|\boldsymbol{a} + \boldsymbol{b}\|^2 \leq (1 + \gamma) \|\boldsymbol{a}\|^2 + \left(1 + \frac{1}{\gamma}\right) \|\boldsymbol{b}\|^2 . \tag{17}$$

*In particular, as a special case where* $\gamma = 1$, *it holds that*

$$\|\boldsymbol{a} + \boldsymbol{b}\|^2 \leq 2 \|\boldsymbol{a}\|^2 + 2 \|\boldsymbol{b}\|^2 . \tag{18}$$

*Proof.* The left hand side of (17) is $\|\boldsymbol{a}\|^2 + 2 \langle \boldsymbol{a}, \boldsymbol{b} \rangle + \|\boldsymbol{b}\|^2$. Applying Lemma C.2 then suffices. $\quad \square$

**Lemma C.4.** *For any two vectors $\boldsymbol{a}$ and $\boldsymbol{b}$, it holds that*

$$\|\boldsymbol{a} - \boldsymbol{b}\|^2 \geq \frac{1}{2} \|\boldsymbol{a}\|^2 - \|\boldsymbol{b}\|^2 .$$

*Proof.* From (18) it follows that

$$\|\boldsymbol{a}\|^2 = \|(\boldsymbol{a} - \boldsymbol{b}) + \boldsymbol{b}\|^2 \leq 2 \|\boldsymbol{a} - \boldsymbol{b}\|^2 + 2 \|\boldsymbol{b}\|^2 .$$

Simply rearranging the terms gives us the result. □

**Lemma C.5** (Generalized Young's inequality). *For any nonnegative scalars $p_1, \ldots, p_n$ such that $p_1 + \cdots + p_n = 1$ and vectors $\boldsymbol{a}_1, \ldots, \boldsymbol{a}_n$, it holds that*

$$\|p_1 \boldsymbol{a}_1 + \cdots + p_n \boldsymbol{a}_n\|^2 \leq p_1 \|\boldsymbol{a}_1\|^2 + \cdots + p_n \|\boldsymbol{a}_n\|^2 .$$

*In particular, setting $p_1 = \cdots = p_n = \frac{1}{n}$ and multiplying both sides by $n^2$ yields*

$$\|\boldsymbol{a}_1 + \cdots + \boldsymbol{a}_n\|^2 \leq n \left( \|\boldsymbol{a}_1\|^2 + \cdots + \|\boldsymbol{a}_n\|^2 \right) .$$

*Proof.* We use induction on $n$. If $n = 1$ then $p_1 = 1$, so there is nothing to show. For the inductive step, suppose that the statement holds for some $n \geq 1$. Say we are given nonnegative scalars $p_1, \ldots, p_{n+1}$ such that $p_1 + \cdots + p_{n+1} = 1$, and vectors $\boldsymbol{a}_1, \ldots, \boldsymbol{a}_{n+1}$. For the moment, suppose that $p_{n+1} < 1$. Applying Lemma C.3 with $\gamma = \frac{p_{n+1}}{1 - p_{n+1}}$ and using the induction hypothesis, we get

$$\|p_1 \boldsymbol{a}_1 + \cdots + p_n \boldsymbol{a}_n + p_{n+1} \boldsymbol{a}_{n+1}\|^2$$
$$\leq \frac{1}{1 - p_{n+1}} \|p_1 \boldsymbol{a}_1 + \cdots + p_n \boldsymbol{a}_n\|^2 + \frac{1}{p_{n+1}} \|p_{n+1} \boldsymbol{a}_{n+1}\|^2$$
$$= (1 - p_{n+1}) \left\| \frac{p_1}{1 - p_{n+1}} \boldsymbol{a}_1 + \cdots + \frac{p_n}{1 - p_{n+1}} \boldsymbol{a}_n \right\|^2 + p_{n+1} \|\boldsymbol{a}_{n+1}\|^2$$
$$\leq p_1 \|\boldsymbol{a}_1\|^2 + \cdots + p_n \|\boldsymbol{a}_n\|^2 + p_{n+1} \|\boldsymbol{a}_{n+1}\|^2$$

where in last line we used that $p_1 + \cdots + p_n = 1 - p_{n+1}$. Now, if $p_{n+1} = 1$, then we must have $p_1 = \cdots = p_n = 0$, so the claimed inequality holds in this case also. This completes the proof. □

**Lemma C.6.** *Suppose that $\boldsymbol{F}$ is $M$-smooth. Then for any $\boldsymbol{z}$ and $\boldsymbol{w}$ it holds that*

$$\|\boldsymbol{F}\boldsymbol{w} - \boldsymbol{F}\boldsymbol{z} - D\boldsymbol{F}(\boldsymbol{z})(\boldsymbol{w} - \boldsymbol{z})\| \leq \frac{M}{2} \|\boldsymbol{w} - \boldsymbol{z}\|^2 .$$

*Proof.* The proof closely follows the arguments used for Lemma 1.2.4 in [40], by replacing the gradients therein by saddle gradients. The fundamental theorem of calculus with the $M$-smoothness of $\boldsymbol{F}$ gives us

$$\|\boldsymbol{F}\boldsymbol{w} - \boldsymbol{F}\boldsymbol{z} - D\boldsymbol{F}(\boldsymbol{z})(\boldsymbol{w} - \boldsymbol{z})\| = \left\| \int_0^1 D\boldsymbol{F}(\boldsymbol{z} + t(\boldsymbol{w} - \boldsymbol{z})) \, \mathrm{d}t \, (\boldsymbol{w} - \boldsymbol{z}) - D\boldsymbol{F}(\boldsymbol{z})(\boldsymbol{w} - \boldsymbol{z}) \right\|$$
$$\leq \|\boldsymbol{w} - \boldsymbol{z}\| \int_0^1 \|D\boldsymbol{F}(\boldsymbol{z} + t(\boldsymbol{w} - \boldsymbol{z})) - D\boldsymbol{F}(\boldsymbol{z})\| \, \mathrm{d}t$$
$$\leq \|\boldsymbol{w} - \boldsymbol{z}\| \int_0^1 Mt \, \|\boldsymbol{w} - \boldsymbol{z}\| \, \mathrm{d}t$$
$$= \frac{M}{2} \|\boldsymbol{w} - \boldsymbol{z}\|^2 . \qquad \square$$

**Lemma C.7.** *Let $\boldsymbol{F}$ be a $\mu$-strongly monotone operator. Let $\boldsymbol{z}^*$ be a point such that $\boldsymbol{F}\boldsymbol{z}^* = \boldsymbol{0}$, and let $\eta > 0$. Then, for any point $\boldsymbol{z}$ in the domain of $\boldsymbol{F}$ and $\boldsymbol{w} := \boldsymbol{z} - \eta \boldsymbol{F}\boldsymbol{z}$, it holds that*

$$\langle \boldsymbol{F}\boldsymbol{w}, \boldsymbol{w} - \boldsymbol{z}^* \rangle \geq \frac{\mu}{2} \|\boldsymbol{z} - \boldsymbol{z}^*\|^2 - \eta^2 \mu \|\boldsymbol{F}\boldsymbol{z}\|^2 .$$

*Proof.* By the $\mu$-strong monotonicity of $\boldsymbol{F}$ and Lemma C.4 it holds that

$$
\begin{aligned}
\langle \boldsymbol{Fw}, \boldsymbol{w} - \boldsymbol{z}^* \rangle &\geq \mu \left\| \boldsymbol{w} - \boldsymbol{z}^* \right\|^2 \\
&= \mu \left\| \boldsymbol{z} - \eta \boldsymbol{Fz} - \boldsymbol{z}^* \right\|^2 \\
&\geq \frac{\mu}{2} \left\| \boldsymbol{z} - \boldsymbol{z}^* \right\|^2 - \mu \left\| \eta \boldsymbol{Fz} \right\|^2
\end{aligned}
$$

so we are done. $\qquad\square$

The following lemma generalizes Lemma 3.2 in [21] shown for monotone $\boldsymbol{F}$ to $\mu$-strongly monotone $\boldsymbol{F}$ with $\mu > 0$.

**Lemma C.8.** *Let $\boldsymbol{F}$ be a $\mu$-strongly monotone $L$-Lipschitz operator, and let $\boldsymbol{z}$ be any point in the domain of $\boldsymbol{F}$. Then for any $0 < \eta < \frac{1}{L\sqrt{2}}$, it holds that*

$$
\left\| \boldsymbol{F}(\boldsymbol{z} - \eta \boldsymbol{F}(\boldsymbol{z} - \eta \boldsymbol{Fz})) \right\|^2 \leq \left( 1 - \frac{2\eta\mu}{5} \right) \left\| \boldsymbol{Fz} \right\|^2 .
$$

*Proof.* For convenience, let us define $\boldsymbol{w} := \boldsymbol{z} - \eta \boldsymbol{Fz}$ and $\boldsymbol{z}^+ := \boldsymbol{z} - \eta \boldsymbol{F}(\boldsymbol{z} - \eta \boldsymbol{Fz}) = \boldsymbol{z} - \eta \boldsymbol{Fw}$. Because $\boldsymbol{F}$ is $\mu$-strongly monotone, we have

$$
\begin{aligned}
\mu \left\| \boldsymbol{z}^+ - \boldsymbol{z} \right\|^2 &\leq \langle \boldsymbol{Fz}^+ - \boldsymbol{Fz}, \boldsymbol{z}^+ - \boldsymbol{z} \rangle \\
&= \eta \langle \boldsymbol{Fz} - \boldsymbol{Fz}^+, \boldsymbol{Fw} \rangle .
\end{aligned} \tag{19}
$$

Also from the $\mu$-strong monotonicity of $\boldsymbol{F}$ we get

$$
\begin{aligned}
\mu \left\| \boldsymbol{w} - \boldsymbol{z}^+ \right\|^2 &\leq \langle \boldsymbol{Fw} - \boldsymbol{Fz}^+, \boldsymbol{w} - \boldsymbol{z}^+ \rangle \\
&= \eta \langle \boldsymbol{Fw} - \boldsymbol{Fz}^+, \boldsymbol{Fw} - \boldsymbol{Fz} \rangle .
\end{aligned} \tag{20}
$$

Meanwhile, from the $L$-Lipschitzness of $\boldsymbol{F}$ we have

$$
\left\| \boldsymbol{Fw} - \boldsymbol{Fz}^+ \right\|^2 \leq \eta^2 L^2 \left\| \boldsymbol{Fw} - \boldsymbol{Fz} \right\|^2 . \tag{21}
$$

Summing up the inequalities (19), (20), (21) with weights $2/\eta$, $1/2\eta$, and $3/2$ respectively, we obtain

$$
\frac{\mu}{\eta} \left( 2 \left\| \boldsymbol{z}^+ - \boldsymbol{z} \right\|^2 + \frac{1}{2} \left\| \boldsymbol{w} - \boldsymbol{z}^+ \right\|^2 \right) + \frac{3}{2} \left\| \boldsymbol{Fw} - \boldsymbol{Fz}^+ \right\|^2
$$

$$
\leq 2 \langle \boldsymbol{Fz} - \boldsymbol{Fz}^+, \boldsymbol{Fw} \rangle + \frac{1}{2} \langle \boldsymbol{Fw} - \boldsymbol{Fz}^+, \boldsymbol{Fw} - \boldsymbol{Fz} \rangle + \frac{3\eta^2 L^2}{2} \left\| \boldsymbol{Fw} - \boldsymbol{Fz} \right\|^2 .
$$

From this inequality, we can exactly follow the arguments used in the proof of Lemma D.4 in [21] to derive that

$$
\frac{\mu}{\eta} \left( 2 \left\| \boldsymbol{z}^+ - \boldsymbol{z} \right\|^2 + \frac{1}{2} \left\| \boldsymbol{w} - \boldsymbol{z}^+ \right\|^2 \right) + \left\| \boldsymbol{Fz}^+ \right\|^2 \leq \left\| \boldsymbol{Fz} \right\|^2 . \tag{22}
$$

Meanwhile, Young's inequality (Lemma C.3) tells us that

$$
\eta^2 \left\| \boldsymbol{Fz} \right\|^2 = \left\| \boldsymbol{w} - \boldsymbol{z} \right\|^2 \leq \left( 1 + \frac{1}{4} \right) \left\| \boldsymbol{w} - \boldsymbol{z}^+ \right\|^2 + (1 + 4) \left\| \boldsymbol{z}^+ - \boldsymbol{z} \right\|^2 .
$$

Using this to lower bound the left hand side of (22), we get that

$$
\frac{2\eta\mu}{5} \left\| \boldsymbol{Fz} \right\|^2 + \left\| \boldsymbol{Fz}^+ \right\|^2 \leq \left\| \boldsymbol{Fz} \right\|^2 .
$$

It remains to simply rearrange the terms. $\qquad\square$

**Lemma C.9.** *Suppose that $\boldsymbol{F}_i$ is $L$-Lipschitz for all $i = 1, \ldots, n$, and that $\boldsymbol{F} := \frac{1}{n} \sum_{i=1}^{n} \boldsymbol{F}_i$ is $\mu$-strongly monotone with $\mu > 0$. Define $\kappa := L/\mu$ and $\sigma_*^2 := \frac{1}{n} \sum_{i=1}^{n} \left\| \boldsymbol{F}_i \boldsymbol{z}^* \right\|^2$. Then, for any $\boldsymbol{z} \in \mathbb{R}^{d_1 + d_2}$ it holds that*

$$
\frac{1}{n} \sum_{i=1}^{n} \left\| \boldsymbol{F}_i \boldsymbol{z} - \boldsymbol{Fz} \right\|^2 \leq \left( \sqrt{3(1 + \kappa^2)} \left\| \boldsymbol{Fz} \right\| + \sqrt{3} \sigma_* \right)^2 .
$$

*Proof.* For any $\boldsymbol{z}, \boldsymbol{w} \in \mathbb{R}^{d_1 + d_2}$, as Assumption 3.1 holds with $\mu > 0$, by Cauchy-Schwarz inequality

$$\mu \|\boldsymbol{z} - \boldsymbol{w}\|^2 \leq \langle \boldsymbol{F}\boldsymbol{z} - \boldsymbol{F}\boldsymbol{w}, \boldsymbol{z} - \boldsymbol{w} \rangle \leq \|\boldsymbol{F}\boldsymbol{z} - \boldsymbol{F}\boldsymbol{w}\| \|\boldsymbol{z} - \boldsymbol{w}\| ,$$

and as a consequence, $\|\boldsymbol{z} - \boldsymbol{w}\| \leq 1/\mu \|\boldsymbol{F}\boldsymbol{z} - \boldsymbol{F}\boldsymbol{w}\|$. Thus, for any $i \in [n]$, it holds that

$$\begin{aligned}
\|\boldsymbol{F}_i\boldsymbol{z} - \boldsymbol{F}\boldsymbol{z}\|^2 &\leq 3\|\boldsymbol{F}_i\boldsymbol{z} - \boldsymbol{F}_i\boldsymbol{z}^*\|^2 + 3\|\boldsymbol{F}\boldsymbol{z} - \boldsymbol{F}\boldsymbol{z}^*\|^2 + 3\|\boldsymbol{F}_i\boldsymbol{z}^* - \boldsymbol{F}\boldsymbol{z}^*\|^2 \\
&\leq 3L^2 \|\boldsymbol{z} - \boldsymbol{z}^*\|^2 + 3\|\boldsymbol{F}\boldsymbol{z} - \boldsymbol{F}\boldsymbol{z}^*\|^2 + 3\|\boldsymbol{F}_i\boldsymbol{z}^*\|^2 \\
&\leq 3\left(\frac{L^2}{\mu^2} + 1\right)\|\boldsymbol{F}\boldsymbol{z} - \boldsymbol{F}\boldsymbol{z}^*\|^2 + 3\|\boldsymbol{F}_i\boldsymbol{z}^*\|^2 \\
&= 3(1 + \kappa^2)\|\boldsymbol{F}\boldsymbol{z}\|^2 + 3\|\boldsymbol{F}_i\boldsymbol{z}^*\|^2 .
\end{aligned}$$

Summing this inequality over $i = 1, \ldots, n$ and then dividing by $n$ leads to

$$\begin{aligned}
\frac{1}{n}\sum_{i=1}^n \|\boldsymbol{F}_i\boldsymbol{z} - \boldsymbol{F}\boldsymbol{z}\|^2 &\leq 3(1 + \kappa^2)\|\boldsymbol{F}\boldsymbol{z}\|^2 + \frac{3}{n}\sum_{i=1}^n \|\boldsymbol{F}_i\boldsymbol{z}^*\|^2 \\
&= 3(1 + \kappa^2)\|\boldsymbol{F}\boldsymbol{z}\|^2 + 3\sigma_*^2.
\end{aligned}$$

The conclusion follows from the basic inequality $a^2 + b^2 \leq (a + b)^2$ which holds for any $a, b \geq 0$. $\quad\square$

**Lemma C.10** (Nonexpansiveness of the EG operator). *Let $\boldsymbol{F}$ be a monotone $L$-Lipschitz operator, and $\boldsymbol{z}^*$ be a point such that $\boldsymbol{F}\boldsymbol{z}^* = \boldsymbol{0}$. Then, for any point $\boldsymbol{z}$ in the domain of $\boldsymbol{F}$ and $\eta > 0$,*

$$\|\boldsymbol{z} - \eta\boldsymbol{F}(\boldsymbol{z} - \eta\boldsymbol{F}\boldsymbol{z}) - \boldsymbol{z}^*\|^2 \leq \|\boldsymbol{z} - \boldsymbol{z}^*\|^2 - \eta^2(1 - \eta^2 L^2)\|\boldsymbol{F}\boldsymbol{z}\|^2 .$$

*Proof.* This classical result dates back to the original paper on EG by Korpelevich [29]. Here, for completeness, we replicate the proof using our notations.

Expanding the left hand side of the inequality stated, we obtain

$$\begin{aligned}
\|\boldsymbol{z} - \eta\boldsymbol{F}(\boldsymbol{z} - \eta\boldsymbol{F}\boldsymbol{z}) - \boldsymbol{z}^*\|^2 &= \|\boldsymbol{z} - \boldsymbol{z}^*\|^2 - 2\langle \eta\boldsymbol{F}(\boldsymbol{z} - \eta\boldsymbol{F}\boldsymbol{z}), \boldsymbol{z} - \boldsymbol{z}^* \rangle + \|\eta\boldsymbol{F}(\boldsymbol{z} - \eta\boldsymbol{F}\boldsymbol{z})\|^2 \\
&= \|\boldsymbol{z} - \boldsymbol{z}^*\|^2 - 2\eta\langle \boldsymbol{F}(\boldsymbol{z} - \eta\boldsymbol{F}\boldsymbol{z}), \boldsymbol{z} - \eta\boldsymbol{F}\boldsymbol{z} - \boldsymbol{z}^* \rangle \\
&\quad - 2\eta^2\langle \boldsymbol{F}(\boldsymbol{z} - \eta\boldsymbol{F}\boldsymbol{z}), \boldsymbol{F}\boldsymbol{z} \rangle + \eta^2\|\boldsymbol{F}(\boldsymbol{z} - \eta\boldsymbol{F}\boldsymbol{z})\|^2 .
\end{aligned} \tag{23}$$

For the first inner product term, we have

$$-2\eta\langle \boldsymbol{F}(\boldsymbol{z} - \eta\boldsymbol{F}\boldsymbol{z}), \boldsymbol{z} - \eta\boldsymbol{F}\boldsymbol{z} - \boldsymbol{z}^* \rangle \leq 0$$

because $\boldsymbol{F}$ is monotone. For the second inner product term, we use the polarization identity (Lemma C.1) and the $L$-Lipschitzness of $\boldsymbol{F}$ to get

$$\begin{aligned}
-2\langle \boldsymbol{F}(\boldsymbol{z} - \eta\boldsymbol{F}\boldsymbol{z}), \boldsymbol{F}\boldsymbol{z} \rangle &= \|\boldsymbol{F}(\boldsymbol{z} - \eta\boldsymbol{F}\boldsymbol{z}) - \boldsymbol{F}\boldsymbol{z}\|^2 - \|\boldsymbol{F}(\boldsymbol{z} - \eta\boldsymbol{F}\boldsymbol{z})\|^2 - \|\boldsymbol{F}\boldsymbol{z}\|^2 \\
&\leq L^2 \|-\eta\boldsymbol{F}\boldsymbol{z}\|^2 - \|\boldsymbol{F}(\boldsymbol{z} - \eta\boldsymbol{F}\boldsymbol{z})\|^2 - \|\boldsymbol{F}\boldsymbol{z}\|^2 \\
&= -(1 - \eta^2 L^2)\|\boldsymbol{F}\boldsymbol{z}\|^2 - \|\boldsymbol{F}(\boldsymbol{z} - \eta\boldsymbol{F}\boldsymbol{z})\|^2 .
\end{aligned}$$

Applying these two bounds on (23) completes the proof. $\quad\square$

**Lemma C.11.** *Let $\{a_k\}_{k\geq 0}$, $\{b_k\}_{k\geq 0}$, $\{c_k\}_{k\geq 0}$, and $\{d_k\}_{k\geq 0}$ be sequences of nonnegative numbers satisfying the recurrence relation*

$$b_k \leq (1 + a_k)d_k - d_{k+1} + c_k \qquad \forall k \geq 0.$$

*Then for any $k \geq 0$ it holds that*

$$d_{k+1} + \sum_{j=0}^k b_j \leq \left(\prod_{j=0}^k (1 + a_j)\right)\left(d_0 + \sum_{j=0}^k c_j\right).$$

*Proof.* Because $a_k \geq 0$, it suffices to show that

$$\sum_{j=0}^{k}(b_j - c_j) \prod_{i=j+1}^{k} (1 + a_i) \leq -d_{k+1} + d_0 \prod_{j=0}^{k}(1 + a_j), \tag{24}$$

as this implies

$$\sum_{j=0}^{k} b_j \leq \sum_{j=0}^{k} b_j \prod_{i=j+1}^{k} (1 + a_i)$$

$$\leq \left( \sum_{j=0}^{k} c_j \prod_{i=j+1}^{k} (1 + a_i) \right) - d_{k+1} + d_0 \prod_{j=0}^{k}(1 + a_j)$$

$$\leq -d_{k+1} + \left( d_0 + \sum_{j=0}^{k} c_k \right) \prod_{j=0}^{k}(1 + a_j).$$

So, we show that (24) holds, by induction on $k$. For the base case $k = 0$, the recurrence relation tells us that

$$b_0 - c_0 \leq (1 + a_0)d_0 - d_1$$

which is exactly (24) when $k = 0$. Now suppose that (24) holds for some $k \geq 0$. Using the induction hypothesis and the recurrence relation we get

$$\sum_{j=0}^{k+1}(b_j - c_j) \prod_{i=j+1}^{k+1} (1 + a_i) = b_{k+1} - c_{k+1} + (1 + a_{k+1}) \left( \sum_{j=0}^{k}(b_j - c_j) \prod_{i=j+1}^{k} (1 + a_i) \right)$$

$$\leq b_{k+1} - c_{k+1} - (1 + a_{k+1})d_{k+1} + d_0 \prod_{j=0}^{k+1}(1 + a_j)$$

$$\leq -d_{k+2} + d_0 \prod_{j=0}^{k+1}(1 + a_j).$$

This shows that (24) holds also for $k + 1$, and we are done. $\qquad\square$

The subsequent lemma is technical, but it can be derived from elementary calculus.

**Lemma C.12.** *For any $K \geq 1$,*

$$\sum_{k=2}^{K+2} \frac{1}{k^{2/3}(\log k)^2} \geq \frac{(K + 3)^{1/3}}{(\log(K + 3))^2}.$$

*Proof.* Consider the function $h(x) := \frac{1}{x^{2/3}(\log x)^2}$ over the interval $[2, K + 3]$. As

$$h'(x) = -\frac{2}{x^{5/3}(\log x)^3} - \frac{2}{3x^{5/3}(\log x)^2} < 0,$$

$h$ is decreasing. Hence, an upper Riemann sum becomes an upper bound for the integral, so we have

$$\sum_{k=2}^{K+2} \frac{1}{k^{2/3}(\log k)^2} \geq \int_{2}^{K+3} \frac{1}{x^{2/3}(\log x)^2} \, \mathrm{d}x. \tag{25}$$

Now consider a function $g : [1, \infty) \to \mathbb{R}$, defined as

$$g(y) := \int_{2}^{y+3} \frac{1}{x^{2/3}(\log x)^2} \, \mathrm{d}x - \frac{(y + 3)^{1/3}}{(\log(y + 3))^2}.$$

Differentiating, we get

$$g'(y) = \frac{2}{(y + 3)^{2/3}(\log(y + 3))^3} + \frac{2}{3(y + 3)^{2/3}(\log(y + 3))^2} > 0$$

whenever $y \geq 1$. That is, $g$ is increasing on $y \geq 1$. We then show that $g(1) \geq 0$. To this end, let us begin with observing that

$$h''(x) = \frac{6}{x^{8/3}(\log x)^4} + \frac{14}{3x^{8/3}(\log x)^3} + \frac{10}{9x^{8/3}(\log x)^2} > 0,$$

from which we get that $h$ is convex. In particular, it holds that

$$\int_2^3 h(x)\,\mathrm{d}x \geq \int_2^3 h'\left(\frac{5}{2}\right)\left(x - \frac{5}{2}\right) + h\left(\frac{5}{2}\right)\mathrm{d}x = h\left(\frac{5}{2}\right),$$

and similarly, $\int_3^4 h(x)\,\mathrm{d}x \geq h(7/2)$. Thus we indeed have

$$g(1) = \int_2^4 h(x)\,\mathrm{d}x - \frac{4^{1/3}}{(\log 4)^2}$$

$$\geq \frac{1}{(5/2)^{2/3}(\log(5/2))^2} + \frac{1}{(7/2)^{2/3}(\log(7/2))^2} - \frac{4^{1/3}}{(\log 4)^2} \geq 0.$$

Recalling that $g$ is increasing, we have $g(K) \geq g(1) \geq 0$ for all $K \geq 1$. This, with (25), implies that

$$\sum_{k=2}^{K+2} \frac{1}{k^{2/3}(\log k)^2} \geq \int_2^{K+3} \frac{1}{x^{2/3}(\log x)^2}\,\mathrm{d}x \geq \frac{(K+3)^{1/3}}{(\log(K+3))^2}$$

holds whenever $K \geq 1$, which is exactly the claimed. $\qquad\square$

# D  Missing Proofs for Section 5

## D.1  Unravelling the Recurrence of the Generalized SEG in (8) and (12)

In Section 5.1, we considered the method where, in a single epoch (hence omitting all superscripts that are used to denote the epoch number for convenience), the iterates are generated following the recurrence

$$\begin{aligned}
\boldsymbol{w}_i &= \boldsymbol{z}_i - \alpha \boldsymbol{T}_i \boldsymbol{z}_i \\
\boldsymbol{z}_{i+1} &= \boldsymbol{z}_i - \beta \boldsymbol{T}_i \boldsymbol{w}_i
\end{aligned} \tag{26}$$

for $i = 0, 1, \ldots, N-1$, where each $\boldsymbol{T}_i$ are sampled from the set $\{\boldsymbol{F}_1, \ldots, \boldsymbol{F}_n\}$, and an additional anchoring step

$$\boldsymbol{z}^\sharp := \frac{\boldsymbol{z}_N + \theta \boldsymbol{z}_0}{1 + \theta} \tag{27}$$

is performed so that $\boldsymbol{z}^\sharp$ is used as the initial point of the next epoch. Notice that (27) is a generalized anchoring step that incorporates all the settings we are considering, as the versions of SEG where anchoring is not used correspond to taking $\theta = 0$, and the anchoring step (12) that is used in SEG-FFA corresponds to taking $\theta = 1$. In this section we would like to prove the following statement regarding this update rule.

**Proposition D.1** (Proposition 5.1)**.** *It holds that*

$$\boldsymbol{z}^\sharp = \boldsymbol{z}_0 - \frac{\beta}{1+\theta}\sum_{j=0}^{N-1} \boldsymbol{T}_j \boldsymbol{z}_0 + \frac{\alpha\beta}{1+\theta}\sum_{j=0}^{N-1} D\boldsymbol{T}_j(\boldsymbol{z}_0)\boldsymbol{T}_j \boldsymbol{z}_0 + \frac{\beta^2}{1+\theta}\sum_{0 \leq i < j \leq N-1} D\boldsymbol{T}_j(\boldsymbol{z}_0)\boldsymbol{T}_i \boldsymbol{z}_0 + \frac{\boldsymbol{\epsilon}_N}{1+\theta} \tag{28}$$

*for some $\boldsymbol{\epsilon}_N = o\left((\alpha+\beta)^2\right)$.*

*Proof.* Equation (28) immediately follows from Proposition D.2, with (30) giving us the precise definition of $\boldsymbol{\epsilon}_N$. To show that $\boldsymbol{\epsilon}_N = o\left((\alpha+\beta)^2\right)$, we begin with noting that both $\|\boldsymbol{z}_j - \boldsymbol{z}_0\|$ and $\|\boldsymbol{w}_j - \boldsymbol{z}_0\|$ are of $\mathcal{O}(\alpha+\beta)$, because both $\boldsymbol{z}_j$ and $\boldsymbol{w}_j$ are obtained from $\boldsymbol{z}_0$ by performing at most $j$ updates following (26). Thus, the first term in the right hand side of (30) is of $\mathcal{O}(\beta(\alpha+\beta)^2)$ by Lemma C.6, and the remaining terms are of $\mathcal{O}((\alpha+\beta)^3)$ by the $L$-smoothness of the operators $\boldsymbol{F}_1, \ldots, \boldsymbol{F}_n$. $\qquad\square$

**Proposition D.2.** *For any $i = 0, 1, \ldots, N$, it holds that*

$$z_i = z_0 - \beta \sum_{j=0}^{i-1} T_j z_0 + \alpha\beta \sum_{j=0}^{i-1} DT_j(z_0)T_j z_0 + \beta^2 \sum_{0 \le k < j \le i-1} DT_j(z_0)T_k z_0 + \epsilon_i \qquad (29)$$

*where we denote*

$$\begin{aligned}
\epsilon_i :=& -\beta \sum_{j=0}^{i-1}\Big(T_j w_j - T_j z_0 - DT_j(z_0)(w_j - z_0)\Big) \\
&+ \alpha\beta \sum_{j=0}^{i-1} DT_j(z_0)(T_j z_j - T_j z_0) + \beta^2 \sum_{j=0}^{i-1} DT_j(z_0) \sum_{k=0}^{j-1}(T_k w_k - T_k z_0).
\end{aligned} \qquad (30)$$

*Proof.* We use induction on $i$. There is nothing to show for the base case $i = 0$. Now, suppose that (29) and (30) hold for some $i < N$, and write

$$\begin{aligned}
z_{i+1} &= z_i - \beta T_i w_i \\
&= z_i - \beta T_i z_0 - \beta DT_i(z_0)(w_i - z_0) - \beta\Big(T_i w_i - T_i z_0 - DT_i(z_0)(w_i - z_0)\Big).
\end{aligned}$$

Here, notice that by the update rule we have

$$\begin{aligned}
w_i &= z_i - \alpha T_i z_i \\
&= z_0 - \beta \sum_{j=0}^{i-1} T_j w_j - \alpha T_i z_i.
\end{aligned}$$

Using this identity and the induction hypothesis we get

$$\begin{aligned}
z_{i+1} =& \; z_0 - \beta \sum_{j=0}^{i-1} T_j z_0 + \alpha\beta \sum_{j=0}^{i-1} DT_j(z_0)T_j z_0 + \beta^2 \sum_{0 \le k < j \le i-1} DT_j(z_0)T_k z_0 + \epsilon_i \\
&- \beta T_i z_0 - \beta DT_i(z_0)\left(-\beta \sum_{j=0}^{i-1} T_j w_j - \alpha T_i z_i\right) \\
&- \beta\Big(T_i w_i - T_i z_0 - DT_i(z_0)(w_i - z_0)\Big) \\
=& \; z_0 - \beta \sum_{j=0}^{i-1} T_j z_0 + \alpha\beta \sum_{j=0}^{i-1} DT_j(z_0)T_j z_0 + \beta^2 \sum_{0 \le k < j \le i-1} DT_j(z_0)T_k z_0 + \epsilon_i \\
&- \beta T_i z_0 + \beta^2 DT_i(z_0) \sum_{j=0}^{i-1} T_j z_0 + \beta^2 DT_i(z_0) \sum_{j=0}^{i-1}(T_j w_j - T_j z_0) \\
&+ \alpha\beta DT_i(z_0)(T_i z_i - T_i z_0) + \alpha\beta DT_i(z_0)T_i z_0 \\
&- \beta\Big(T_i w_i - T_i z_0 - DT_i(z_0)(w_i - z_0)\Big) \\
=& \; z_0 - \beta \sum_{j=0}^{i} T_j z_0 + \alpha\beta \sum_{j=0}^{i} DT_j(z_0)T_j z_0 + \beta^2 \sum_{0 \le k < j \le i} DT_j(z_0)T_k z_0 + \epsilon_i \\
&+ \beta^2 DT_i(z_0) \sum_{j=0}^{i-1}(T_j w_j - T_j z_0) + \alpha\beta DT_i(z_0)(T_i z_i - T_i z_0) \\
&- \beta\Big(T_i w_i - T_i z_0 - DT_i(z_0)(w_i - z_0)\Big) \\
=& \; z_0 - \beta \sum_{j=0}^{i} T_j z_0 + \alpha\beta \sum_{j=0}^{i} DT_j(z_0)T_j z_0 + \beta^2 \sum_{0 \le k < j \le i} DT_j(z_0)T_k z_0 + \epsilon_{i+1}
\end{aligned}$$

which asserts that (29) also holds for $i + 1$. $\qquad\square$

## D.2 Insufficiency of Only Using Flip-Flop Sampling

Here we prove the following.

**Proposition D.3** (Proposition 5.2). *Suppose we use flip-flop sampling only. In order to make* (10) *and* (11) *hold, we must choose* $\beta = \eta_1/n$ *and* $\alpha = \beta/2$. *However, this leads to* $\eta_2 = 2\eta_1$, *which is the set of parameters that fails to make EG+ converge.*

*Proof.* Suppose that we have already established the upcoming Lemma D.4. Then, we can see by setting $\theta = 0$ in the result of Lemma D.4 that for (11) to hold, the following system of equations should be satisfied:

$$\begin{cases} \eta_1\eta_2 = 2n^2\beta^2, \\ \eta_1\eta_2 = n^2(2\alpha\beta + \beta^2), \\ \eta_2 = 2n\beta. \end{cases}$$

Solving this system of equations, we get $\eta_1 = n\beta$, $\eta_2 = 2n\beta$, and $\alpha = \beta/2$.

For the latter part of the statement on the divergence of EG+ with $\eta_2 = 2\eta_1$, consider the $(1+1)$-dimensional bilinear problem

$$\min_x \max_y \ xy$$

whose unique optimum is $z^* = (0,0)$. A simple computation shows that

$$\boldsymbol{F}\boldsymbol{z} = \begin{bmatrix} 0 & 1 \\ -1 & 0 \end{bmatrix} \boldsymbol{z}.$$

Consequently, for any $\eta > 0$, the update rule of EG+ with $\eta_1 = \eta$ and $\eta_2 = 2\eta$ amounts to

$$\boldsymbol{z}^+ = \boldsymbol{z} - 2\eta\boldsymbol{F}(\boldsymbol{z} - \eta\boldsymbol{F}\boldsymbol{z}) = \begin{bmatrix} 1 - 2\eta^2 & -2\eta \\ 2\eta & 1 - 2\eta^2 \end{bmatrix} \boldsymbol{z}.$$

It follows that

$$\begin{aligned} \left\| \boldsymbol{z}^+ - \boldsymbol{z}^* \right\|^2 &= \left\| \begin{bmatrix} 1 - 2\eta^2 & -2\eta \\ 2\eta & 1 - 2\eta^2 \end{bmatrix} \begin{bmatrix} x \\ y \end{bmatrix} \right\|^2 \\ &= \left( (1 - 2\eta^2)x - 2\eta y \right)^2 + \left( 2\eta x + (1 - 2\eta^2)y \right)^2 \\ &= (1 + 4\eta^4)(x^2 + y^2) \\ &= (1 + 4\eta^4) \left\| \boldsymbol{z} - \boldsymbol{z}^* \right\|^2. \end{aligned}$$

Therefore, the distance from the optimal solution strictly increases every iterate. □

It remains to actually prove Lemma D.4.

**Lemma D.4.** *When flip-flop sampling is used with the generalized anchoring step* (27), *it holds that*

$$\frac{\alpha\beta}{1+\theta} \sum_{j=0}^{N-1} D\boldsymbol{T}_j(\boldsymbol{z}_0)\boldsymbol{T}_j\boldsymbol{z}_0 + \frac{\beta^2}{1+\theta} \sum_{0 \le i < j \le N-1} D\boldsymbol{T}_j(\boldsymbol{z}_0)\boldsymbol{T}_i\boldsymbol{z}_0$$

$$= \frac{2\alpha\beta + \beta^2}{1+\theta} \sum_{j=1}^{n} D\boldsymbol{F}_j(\boldsymbol{z}_0)\boldsymbol{F}_j\boldsymbol{z}_0 + \frac{2\beta^2}{1+\theta} \sum_{i \ne j} D\boldsymbol{F}_j(\boldsymbol{z}_0)\boldsymbol{F}_i\boldsymbol{z}_0.$$

*Proof.* As we are using flip-flop sampling, we have $N = 2n$, and it is clear that

$$\sum_{j=0}^{N-1} D\boldsymbol{T}_j(\boldsymbol{z}_0)\boldsymbol{T}_j\boldsymbol{z}_0 = 2 \sum_{j=1}^{n} D\boldsymbol{F}_j(\boldsymbol{z}_0)\boldsymbol{F}_j\boldsymbol{z}_0.$$

For the second term, as $\boldsymbol{T}_i = \boldsymbol{T}_{2n-1-i}$, we have

$$\sum_{0 \le i < j \le 2n-1} D\boldsymbol{T}_j(\boldsymbol{z}_0)\boldsymbol{T}_i\boldsymbol{z}_0 = \sum_{0 \le i < j \le n-1} D\boldsymbol{T}_j(\boldsymbol{z}_0)\boldsymbol{T}_i\boldsymbol{z}_0 + \sum_{n \le i < j \le 2n-1} D\boldsymbol{T}_j(\boldsymbol{z}_0)\boldsymbol{T}_i\boldsymbol{z}_0$$

$$+ \sum_{i=0}^{n-1} \sum_{j=n}^{2n-2-i} D\boldsymbol{T}_j(\boldsymbol{z}_0)\boldsymbol{T}_i\boldsymbol{z}_0 + \sum_{i=0}^{n-1} \sum_{j=2n-i}^{2n-1} D\boldsymbol{T}_j(\boldsymbol{z}_0)\boldsymbol{T}_i\boldsymbol{z}_0$$

$$+ \sum_{i=0}^{n-1} D\boldsymbol{T}_{2n-1-i}(\boldsymbol{z}_0)\boldsymbol{T}_i\boldsymbol{z}_0$$

$$= \sum_{0 \le i < j \le n-1} D\boldsymbol{T}_j(\boldsymbol{z}_0)\boldsymbol{T}_i\boldsymbol{z}_0 + \sum_{0 \le j < i \le n-1} D\boldsymbol{T}_j(\boldsymbol{z}_0)\boldsymbol{T}_i\boldsymbol{z}_0$$

$$+ \sum_{i=0}^{n-1} \sum_{j=i+1}^{n-1} D\boldsymbol{T}_j(\boldsymbol{z}_0)\boldsymbol{T}_i\boldsymbol{z}_0 + \sum_{i=0}^{n-1} \sum_{j=0}^{i-1} D\boldsymbol{T}_j(\boldsymbol{z}_0)\boldsymbol{T}_i\boldsymbol{z}_0$$

$$+ \sum_{i=0}^{n-1} D\boldsymbol{T}_i(\boldsymbol{z}_0)\boldsymbol{T}_i\boldsymbol{z}_0$$

$$= 2 \sum_{0 \le i < j \le n-1} D\boldsymbol{T}_j(\boldsymbol{z}_0)\boldsymbol{T}_i\boldsymbol{z}_0 + 2 \sum_{0 \le j < i \le n-1} D\boldsymbol{T}_j(\boldsymbol{z}_0)\boldsymbol{T}_i\boldsymbol{z}_0$$

$$+ \sum_{i=0}^{n-1} D\boldsymbol{T}_i(\boldsymbol{z}_0)\boldsymbol{T}_i\boldsymbol{z}_0.$$

The claimed identity can be obtained by taking the weighted sum of the two results. $\square$

## E   Within-Epoch Error Analysis for Upper Bounds

All the upper bounds for SEG-RR, SEG-FF, and SEG-FFA in this paper are established by following the two steps below.

The first step is to decompose the cumulative updates made within an epoch by using the method into a sum of an exact EG update and a within-epoch error term, which we denote by $\boldsymbol{r}^k$. In particular, we show that the error term $\boldsymbol{r}^k$ occurring from any of SEG-RR, SEG-FF, and SEG-FFA can be expressed in a specific unified form (described in Theorem E.1). This will be the main focus of this section.

The second step is establishing a convergence rate that can be applied to *any* method whose update can be decomposed into a sum of an exact EG update and an error term that is of the specific unified form mentioned above. By doing so, the convergence rates of SEG-RR, SEG-FF, and SEG-FFA will automatically follow as special cases of the general convergence result. This step will be dealt in Appendices F and G.

To this end, for any of SEG-RR, SEG-FF, and SEG-FFA, let us decompose the cumulative updates made within an epoch into a sum of an exact EG update and a within-epoch error term $\boldsymbol{r}^k$, as

$$\boldsymbol{z}_0^{k+1} = \boldsymbol{z}_0^k - \eta_k n \boldsymbol{F}(\boldsymbol{z}_0^k - \eta_k n \boldsymbol{F}\boldsymbol{z}_0^k) + \boldsymbol{r}^k.$$

The quality of the method will depend on how small the "noise" term $\boldsymbol{r}^k$ is, as the noise will in general hinder the convergence. As mentioned above, it turns out that, regardless of the method that is in use, the noise term can be bounded in a unified format, as follows.

**Theorem E.1.** *Suppose that Assumptions 3.3 and 3.4 hold. Then, for each of SEG-RR, SEG-FF, and SEG-FFA, there exists a choice of stepsizes that makes the following hold: for an exponent $a$ that depends on the method, there exist constants $C_1$, $D_1$, $V_1$, $C_2$, $D_2$, and $V_2$, all independent of $\eta_k$ and $n$, such that the error term $\boldsymbol{r}^k$ satisfies a deterministic bound*

$$\left\|\boldsymbol{r}^k\right\| \le \eta_k^a n^a C_1 \left\|\boldsymbol{F}\boldsymbol{z}_0^k\right\| + \eta_k^a n^a D_1 \left\|\boldsymbol{F}\boldsymbol{z}_0^k\right\|^2 + \eta_k^a n^a V_1 \tag{31}$$

*and a bound that holds on expectation*

$$\mathbb{E}\left[\left\|\boldsymbol{r}\right\|^2 \Big| \boldsymbol{z}_0^k\right] \le \eta_k^{2a} n^{2a} C_2 \left\|\boldsymbol{F}\boldsymbol{z}_0^k\right\|^2 + \eta_k^{2a} n^{2a} D_2 \left\|\boldsymbol{F}\boldsymbol{z}_0^k\right\|^4 + \eta_k^{2a} n^{2a-1} V_2. \tag{32}$$

*Furthermore, the exponent is $a = 2$ for SEG-RR and SEG-FF, and $a = 3$ for SEG-FFA.*

In other words, SEG-FFA has an error that is an order of magnitude smaller than other methods. Thus, it is now intuitively clear that SEG-FFA should have an advantage in the convergence. The proof of Theorem E.1 is quite long and technical, so we defer it to Appendix E.2.

Within the remaining of this section only, although it is an abuse of notation, for convenience we will write $\boldsymbol{F}_i$ to denote the saddle gradient of the component function chosen in the $i^{\text{th}}$ iteration. More precisely, for indices $i = 0, 1, \ldots, n-1$ we denote $\boldsymbol{F}_{\tau(i+1)}$ by $\boldsymbol{F}_i$. Similarly, in cases of considering SEG-FF or SEG-FFA, for $i \geq n$ we denote $\boldsymbol{F}_{\tau(2n-i)}$ by $\boldsymbol{F}_i$. Also, we omit the superscripts and subscripts denoting the epoch number $k$ unless strictly necessary, as all the iterates that we consider will be from the same epoch.

Let us reformulate the update rule (8) into

$$
\begin{aligned}
\boldsymbol{w}_i &= \boldsymbol{z}_i - \xi\eta\boldsymbol{F}_i\boldsymbol{z}_i, \\
\boldsymbol{z}_{i+1} &= \boldsymbol{z}_i - \eta\boldsymbol{F}_i\boldsymbol{w}_i.
\end{aligned}
\tag{33}
$$

Note that $\xi = 1/2$ for SEG-FFA, and $\xi = 1$ for SEG-RR and SEG-FF.

## E.1 Auxiliary Lemmata

For $j = 1, \ldots, 2n$ we define

$$
\boldsymbol{g}_j := \sum_{i=0}^{j-1} \boldsymbol{F}_i\boldsymbol{z}_0,
\tag{34}
$$

$$
\delta_j := \|\boldsymbol{g}_j - j\boldsymbol{F}\boldsymbol{z}_0\|,
\tag{35}
$$

$$
\Sigma_j := \sum_{i=1}^{j} \delta_i,
\tag{36}
$$

$$
\Psi_j := \sum_{i=1}^{j} \delta_i^2.
\tag{37}
$$

We set $\Sigma_0 = \Psi_0 = 0$, as they are empty sums. Notice that $\delta_j$ is a random variable that depends on the permutation $\tau$.

Meanwhile, by triangle inequality it is immediate that

$$
\|\boldsymbol{g}_j\| \leq j\|\boldsymbol{F}\boldsymbol{z}_0\| + \delta_j,
$$

and by Young's inequality it holds that

$$
\|\boldsymbol{g}_j\|^2 \leq 2j^2\|\boldsymbol{F}\boldsymbol{z}_0\|^2 + 2\delta_j^2.
$$

**Lemma E.2.** *For any index $i \geq 1$, it holds that*

$$
\begin{aligned}
\|\boldsymbol{z}_i - \boldsymbol{z}_0\| &\leq \eta\left(1 + \xi\eta L\right)\|\boldsymbol{g}_i\| \\
&\quad + \eta^2 L\left(2\xi + 2\xi\eta L + \xi^2\eta^2 L^2\right)\sum_{\ell=0}^{i-2}\left(1 + \eta L + \xi\eta^2 L^2\right)^{i-\ell-2}\|\boldsymbol{g}_{\ell+1}\|,
\end{aligned}
\tag{38}
$$

$$
\begin{aligned}
\|\boldsymbol{w}_i - \boldsymbol{z}_0\| &\leq \xi\eta\|\boldsymbol{g}_{i+1}\| + \xi\eta\left((1 - \xi^{-1}) + 2\eta L + \xi\eta^2 L^2\right)\|\boldsymbol{g}_i\| \\
&\quad + \eta(1 + \xi\eta L)\left(2\xi\eta L + 2\xi\eta^2 L^2 + \xi^2\eta^3 L^3\right)\sum_{\ell=0}^{i-2}\left(1 + \eta L + \xi\eta^2 L^2\right)^{i-\ell-2}\|\boldsymbol{g}_{\ell+1}\|.
\end{aligned}
\tag{39}
$$

*Proof.* By the fundamental theorem of calculus for line integrals and the update rule (33), we have

$$
\begin{aligned}
\boldsymbol{w}_i &= \boldsymbol{z}_i - \xi\eta\boldsymbol{F}_i\boldsymbol{z}_i \\
&= \boldsymbol{z}_i - \xi\eta\boldsymbol{F}_i\boldsymbol{z}_0 - \xi\eta(\boldsymbol{F}_i\boldsymbol{z}_i - \boldsymbol{F}_i\boldsymbol{z}_0) \\
&= \boldsymbol{z}_i - \xi\eta\boldsymbol{F}_i\boldsymbol{z}_0 - \xi\eta\int_0^1 D\boldsymbol{F}_i(\boldsymbol{z}_0 + t(\boldsymbol{z}_i - \boldsymbol{z}_0))\,\mathrm{d}t\,(\boldsymbol{z}_i - \boldsymbol{z}_0)
\end{aligned}
$$

and similarly

$$\begin{aligned}
\boldsymbol{z}_{i+1} &= \boldsymbol{z}_i - \eta \boldsymbol{F}_i \boldsymbol{w}_i \\
&= \boldsymbol{z}_i - \eta \boldsymbol{F}_i \boldsymbol{z}_0 - \eta(\boldsymbol{F}_i \boldsymbol{w}_i - \boldsymbol{F}_i \boldsymbol{z}_0) \\
&= \boldsymbol{z}_i - \eta \boldsymbol{F}_i \boldsymbol{z}_0 - \eta \int_0^1 D\boldsymbol{F}_i(\boldsymbol{z}_0 + t(\boldsymbol{w}_i - \boldsymbol{z}_0))\, \mathrm{d}t\, (\boldsymbol{w}_i - \boldsymbol{z}_0).
\end{aligned}$$

Hence, by defining

$$\begin{aligned}
\boldsymbol{A}_i &:= \int_0^1 D\boldsymbol{F}_i(\boldsymbol{z}_0 + t(\boldsymbol{z}_i - \boldsymbol{z}_0))\, \mathrm{d}t \\
\boldsymbol{B}_i &:= \int_0^1 D\boldsymbol{F}_i(\boldsymbol{z}_0 + t(\boldsymbol{w}_i - \boldsymbol{z}_0))\, \mathrm{d}t
\end{aligned} \tag{40}$$

the update rule can be rewritten using these quantities as

$$\boldsymbol{w}_i = \boldsymbol{z}_i - \xi\eta\boldsymbol{F}_i\boldsymbol{z}_0 - \xi\eta\boldsymbol{A}_i(\boldsymbol{z}_i - \boldsymbol{z}_0), \tag{41}$$
$$\boldsymbol{z}_{i+1} = \boldsymbol{z}_i - \eta\boldsymbol{F}_i\boldsymbol{z}_0 - \eta\boldsymbol{B}_i(\boldsymbol{w}_i - \boldsymbol{z}_0). \tag{42}$$

Subtracting $\boldsymbol{z}_0$ from both sides of (41) we get

$$\begin{aligned}
\boldsymbol{w}_i - \boldsymbol{z}_0 &= \boldsymbol{z}_i - \boldsymbol{z}_0 - \xi\eta\boldsymbol{F}_i\boldsymbol{z}_0 - \xi\eta\boldsymbol{A}_i(\boldsymbol{z}_i - \boldsymbol{z}_0) \\
&= \left(\boldsymbol{I} - \xi\eta\boldsymbol{A}_i\right)(\boldsymbol{z}_i - \boldsymbol{z}_0) - \xi\eta\boldsymbol{F}_i\boldsymbol{z}_0,
\end{aligned} \tag{43}$$

and plugging this into (42) gives us

$$\begin{aligned}
\boldsymbol{z}_{i+1} - \boldsymbol{z}_0 &= \boldsymbol{z}_i - \boldsymbol{z}_0 - \eta\boldsymbol{F}_i\boldsymbol{z}_0 - \eta\boldsymbol{B}_i(\boldsymbol{w}_i - \boldsymbol{z}_0) \\
&= \boldsymbol{z}_i - \boldsymbol{z}_0 - \eta\boldsymbol{F}_i\boldsymbol{z}_0 - \eta\boldsymbol{B}_i\left((\boldsymbol{I} - \xi\eta\boldsymbol{A}_i)(\boldsymbol{z}_i - \boldsymbol{z}_0) - \xi\eta\boldsymbol{F}_i\boldsymbol{z}_0\right) \\
&= \left(\boldsymbol{I} - \eta\boldsymbol{B}_i + \xi\eta^2\boldsymbol{B}_i\boldsymbol{A}_i\right)(\boldsymbol{z}_i - \boldsymbol{z}_0) - \eta\left(\boldsymbol{I} - \xi\eta\boldsymbol{B}_i\right)\boldsymbol{F}_i\boldsymbol{z}_0.
\end{aligned} \tag{44}$$

For convenience let us define

$$\begin{aligned}
\boldsymbol{C}_i &:= \boldsymbol{I} - \eta\boldsymbol{B}_i + \xi\eta^2\boldsymbol{B}_i\boldsymbol{A}_i, \\
\boldsymbol{P}_{i,\ell} &:= \boldsymbol{C}_i\boldsymbol{C}_{i-1}\ldots\boldsymbol{C}_{\ell+2}\boldsymbol{C}_{\ell+1}
\end{aligned}$$

and $\boldsymbol{P}_{i,i} := \boldsymbol{I}$ as it denotes an empty product. Observe that for any $j$ we have

$$\|\boldsymbol{C}_j\| = \left\|\boldsymbol{I} - \eta\boldsymbol{B}_i + \xi\eta^2\boldsymbol{B}_i\boldsymbol{A}_i\right\| \le 1 + \eta L + \xi\eta^2 L^2. \tag{45}$$

Also note that for any $\ell$ it holds that

$$\begin{aligned}
(\boldsymbol{I} &- \xi\eta\boldsymbol{B}_{\ell+1}) - \boldsymbol{C}_{\ell+1}(\boldsymbol{I} - \xi\eta\boldsymbol{B}_\ell) \\
&= (\boldsymbol{I} - \xi\eta\boldsymbol{B}_{\ell+1}) - \left(\boldsymbol{I} - \eta\boldsymbol{B}_{\ell+1} + \xi\eta^2\boldsymbol{B}_{\ell+1}\boldsymbol{A}_{\ell+1}\right)(\boldsymbol{I} - \xi\eta\boldsymbol{B}_\ell) \\
&= \xi\eta(\boldsymbol{B}_{\ell+1} + \boldsymbol{B}_\ell) - \xi\eta^2\boldsymbol{B}_{\ell+1}(\boldsymbol{A}_{\ell+1} + \boldsymbol{B}_\ell) + \xi^2\eta^3\boldsymbol{B}_{\ell+1}\boldsymbol{A}_{\ell+1}\boldsymbol{B}_\ell
\end{aligned}$$

and hence

$$\|(\boldsymbol{I} - \xi\eta\boldsymbol{B}_{\ell+1}) - \boldsymbol{C}_{\ell+1}(\boldsymbol{I} - \xi\eta\boldsymbol{B}_\ell)\| \le 2\xi\eta L + 2\xi\eta^2 L^2 + \xi^2\eta^3 L^3. \tag{46}$$

Unravelling the recurrence relation (44) we get

$$\begin{aligned}
\boldsymbol{z}_{i+1} - \boldsymbol{z}_0 &= \boldsymbol{C}_i(\boldsymbol{z}_i - \boldsymbol{z}_0) - \eta\left(\boldsymbol{I} - \xi\eta\boldsymbol{B}_i\right)\boldsymbol{F}_i\boldsymbol{z}_0 \\
&= \boldsymbol{C}_i\big(\boldsymbol{C}_{i-1}(\boldsymbol{z}_{i-1} - \boldsymbol{z}_0) - \eta\left(\boldsymbol{I} - \xi\eta\boldsymbol{B}_{i-1}\right)\boldsymbol{F}_{i-1}\boldsymbol{z}_0\big) - \eta\left(\boldsymbol{I} - \xi\eta\boldsymbol{B}_i\right)\boldsymbol{F}_i\boldsymbol{z}_0 \\
&= \boldsymbol{P}_{i,i-2}(\boldsymbol{z}_{i-1} - \boldsymbol{z}_0) - \eta\sum_{\ell=i-1}^i \boldsymbol{P}_{i,\ell}\left(\boldsymbol{I} - \xi\eta\boldsymbol{B}_\ell\right)\boldsymbol{F}_\ell\boldsymbol{z}_0 \\
&= \boldsymbol{P}_{i,i-2}\big(\boldsymbol{C}_{i-2}(\boldsymbol{z}_{i-2} - \boldsymbol{z}_0) - \eta\left(\boldsymbol{I} - \xi\eta\boldsymbol{B}_{i-2}\right)\boldsymbol{F}_{i-2}\boldsymbol{z}_0\big) - \eta\sum_{\ell=i-1}^i \boldsymbol{P}_{i,\ell}\left(\boldsymbol{I} - \xi\eta\boldsymbol{B}_\ell\right)\boldsymbol{F}_\ell\boldsymbol{z}_0 \\
&= \boldsymbol{P}_{i,i-3}(\boldsymbol{z}_{i-2} - \boldsymbol{z}_0) - \eta\sum_{\ell=i-2}^i \boldsymbol{P}_{i,\ell}\left(\boldsymbol{I} - \xi\eta\boldsymbol{B}_\ell\right)\boldsymbol{F}_\ell\boldsymbol{z}_0 \\
&\;\;\vdots \\
&= \boldsymbol{P}_{i,-1}(\boldsymbol{z}_0 - \boldsymbol{z}_0) - \eta\sum_{\ell=0}^i \boldsymbol{P}_{i,\ell}\left(\boldsymbol{I} - \xi\eta\boldsymbol{B}_\ell\right)\boldsymbol{F}_\ell\boldsymbol{z}_0
\end{aligned}$$

and therefore

$$z_i - z_0 = -\eta \sum_{\ell=0}^{i-1} P_{i-1,\ell} \left( I - \xi\eta B_\ell \right) F_\ell z_0. \tag{47}$$

In order to compute the bound for $\|z_i - z_0\|$, we use summation by parts to get

$$
\begin{aligned}
\frac{1}{\eta}(z_0 - z_i) &= \sum_{\ell=0}^{i-1} P_{i-1,\ell} \left( I - \xi\eta B_\ell \right) F_\ell z_0 \\
&= P_{i-1,i-1} \left( I - \xi\eta B_{i-1} \right) \sum_{\ell=0}^{i-1} F_\ell z_0 \\
&\quad - \sum_{\ell=0}^{i-2} \left( P_{i-1,\ell+1} \left( I - \xi\eta B_{\ell+1} \right) - P_{i-1,\ell} \left( I - \xi\eta B_\ell \right) \right) \sum_{j=0}^{\ell} F_\ell z_0 \\
&= \left( I - \xi\eta B_{i-1} \right) g_i - \sum_{\ell=0}^{i-2} \left( P_{i-1,\ell+1} \left( I - \xi\eta B_{\ell+1} \right) - P_{i-1,\ell} \left( I - \xi\eta B_\ell \right) \right) g_{\ell+1}.
\end{aligned}
$$

Here, observe that

$$
\begin{aligned}
P_{i-1,\ell+1} \left( I - \xi\eta B_{\ell+1} \right) &- P_{i-1,\ell} \left( I - \xi\eta B_\ell \right) \\
&= C_{i-1} C_{i-2} \dots C_{\ell+2} \left( \left( I - \xi\eta B_{\ell+1} \right) - C_{\ell+1} \left( I - \xi\eta B_\ell \right) \right)
\end{aligned}
$$

so by using (45) and (46) we obtain

$$
\begin{aligned}
\| P_{i-1,\ell+1} & \left( I - \xi\eta B_{\ell+1} \right) - P_{i-1,\ell} \left( I - \xi\eta B_\ell \right) \| \\
&\leq \left( 2\xi\eta L + 2\xi\eta^2 L^2 + \xi^2\eta^3 L^3 \right) \left( 1 + \eta L + \xi\eta^2 L^2 \right)^{i-\ell-2}.
\end{aligned}
$$

Therefore, we conclude that

$$\|z_i - z_0\| \leq \eta \left( 1 + \xi\eta L \right) \|g_i\| + \eta^2 L \left( 2\xi + 2\xi\eta L + \xi^2\eta^2 L^2 \right) \sum_{\ell=0}^{i-2} \left( 1 + \eta L + \xi\eta^2 L^2 \right)^{i-\ell-2} \|g_{\ell+1}\|.$$

Meanwhile, substituting (47) back to (43) gives us

$$w_i - z_0 = -\xi\eta F_i z_0 - \eta \sum_{\ell=0}^{i-1} \left( I - \xi\eta A_i \right) P_{i-1,\ell} \left( I - \xi\eta B_\ell \right) F_\ell z_0. \tag{48}$$

For $\ell < i$ let us define

$$
\begin{aligned}
R_{i,\ell} &:= \xi^{-1} \left( I - \xi\eta A_i \right) P_{i-1,\ell} \left( I - \xi\eta B_\ell \right) \\
&= \xi^{-1} \left( I - \xi\eta A_i \right) C_{i-1} C_{i-2} \dots C_{\ell+2} C_{\ell+1} \left( I - \xi\eta B_\ell \right)
\end{aligned}
$$

and for convenience $R_{i,i} := I$ so that (48) can be rewritten as

$$\frac{1}{\xi\eta}(z_0 - w_i) = \sum_{\ell=0}^{i} R_{i,\ell} F_\ell z_0. \tag{49}$$

Applying summation by parts on the above, we obtain

$$
\begin{aligned}
\frac{1}{\xi\eta}(z_0 - w_i) &= R_{i,i} \sum_{\ell=0}^{i} F_\ell z_0 - \sum_{\ell=0}^{i-1} (R_{i,\ell+1} - R_{i,\ell}) \sum_{j=0}^{\ell} F_j z_0 \\
&= g_{i+1} - \sum_{\ell=0}^{i-1} (R_{i,\ell+1} - R_{i,\ell}) g_{\ell+1}
\end{aligned}
$$

and as a consequence we get

$$\frac{1}{\xi\eta} \|w_i - z_0\| \leq \|g_{i+1}\| + \sum_{\ell=0}^{i-1} \|R_{i,\ell+1} - R_{i,\ell}\| \|g_{\ell+1}\|. \tag{50}$$

It remains to bound $\|\boldsymbol{R}_{i,\ell+1} - \boldsymbol{R}_{i,\ell}\|$. For the special case where $\ell = i - 1$, a direct computation leads to

$$\boldsymbol{R}_{i,i} - \boldsymbol{R}_{i,i-1} = \boldsymbol{I} - \xi^{-1}\left(\boldsymbol{I} - \xi\eta\boldsymbol{A}_i\right)\left(\boldsymbol{I} - \xi\eta\boldsymbol{B}_{i-1}\right)$$
$$= (1 - \xi^{-1})\boldsymbol{I} + \eta\boldsymbol{A}_i + \eta\boldsymbol{B}_{i-1} - \xi\eta^2\boldsymbol{A}_i\boldsymbol{B}_{i-1}$$

and thus we have

$$\|\boldsymbol{R}_{i,i} - \boldsymbol{R}_{i,i-1}\| \leq (1 - \xi^{-1}) + 2\eta L + \xi\eta^2 L^2. \tag{51}$$

For the other cases; that is, when $\ell < i - 1$, we have

$$\boldsymbol{R}_{i,\ell+1} - \boldsymbol{R}_{i,\ell} = \xi^{-1}\left(\boldsymbol{I} - \xi\eta\boldsymbol{A}_i\right)\boldsymbol{C}_{i-1}\boldsymbol{C}_{i-2}\ldots\boldsymbol{C}_{\ell+2}\left(\left(\boldsymbol{I} - \xi\eta\boldsymbol{B}_{\ell+1}\right) - \boldsymbol{C}_{\ell+1}\left(\boldsymbol{I} - \xi\eta\boldsymbol{B}_\ell\right)\right)$$

so by using (45) and (46) we get the bound

$$\|\boldsymbol{R}_{i,\ell+1} - \boldsymbol{R}_{i,\ell}\| \leq \xi^{-1}(1 + \xi\eta L)\left(2\xi\eta L + 2\xi\eta^2 L^2 + \xi^2\eta^3 L^3\right)\left(1 + \eta L + \xi\eta^2 L^2\right)^{i-\ell-2}. \tag{52}$$

Applying (51) and (52) on (50) gives the bound for $\|\boldsymbol{w}_i - \boldsymbol{z}_0\|$. $\qquad\square$

**Proposition E.3.** *Suppose that* **SEG-FFA** *is used, $\eta < \frac{1}{nL}$, and let $\nu := 1 + \frac{1}{2n}$. Then for any $i = 1, \ldots, 2n - 1$ we have the bounds*

$$\|\boldsymbol{z}_i - \boldsymbol{z}_0\| \leq \left(\eta\nu i + \frac{\eta\nu^2 e^2 i(i-1)}{2n}\right)\|\boldsymbol{F}\boldsymbol{z}_0\| + \eta\nu\delta_i + \eta^2 L\nu^2 e^2\Sigma_{i-1},$$

$$\|\boldsymbol{w}_i - \boldsymbol{z}_0\| \leq \frac{\eta}{2}\left(1 + 2\nu^2 i + \frac{\nu^3 e^2 i(i-1)}{n}\right)\|\boldsymbol{F}\boldsymbol{z}_0\| + \frac{\eta}{2}\delta_{i+1} + \frac{\eta(2\nu^2 - 1)}{2}\delta_i + \eta^2 L\nu^3 e^2\Sigma_{i-1},$$

$$\|\boldsymbol{w}_i - \boldsymbol{z}_0\|^2 \leq \left(\frac{3\eta^2(i+1)^2}{2} + \frac{3\eta^2(2\nu^2 - 1)^2 i^2}{2} + \frac{\eta^2\nu^6 e^4 i(i-1)^2(2i-1)}{n^2}\right)\|\boldsymbol{F}\boldsymbol{z}_0\|^2$$
$$+ \frac{3\eta^2}{2}\delta_{i+1}^2 + \frac{3\eta^2(2\nu^2 - 1)^2}{2}\delta_i^2 + \frac{6\eta^2\nu^6 e^4(i-1)}{n^2}\Psi_{i-1}.$$

*Proof.* Using elementary calculus one can show that $x \mapsto (1 + \frac{1}{x} + \frac{1}{2x^2})^x$ increases on $x > 0$ and is bounded above by $e$. Hence for all $0 \leq \ell < i \leq 2n$ we have

$$\left(1 + \eta L + \frac{\eta^2 L^2}{2}\right)^{i-\ell-2} \leq \left(1 + \frac{1}{n} + \frac{1}{2n^2}\right)^{2n} \leq e^2.$$

Applying the definitions (35) and (36) on (38) and then substituting $\xi = 1/2$ we get

$$\|\boldsymbol{z}_i - \boldsymbol{z}_0\| \leq \eta\left(1 + \frac{\eta L}{2}\right)\|\boldsymbol{g}_i\| + \eta^2 L\left(1 + \frac{\eta L}{2}\right)^2\sum_{\ell=0}^{i-2}\left(1 + \eta L + \frac{\eta^2 L^2}{2}\right)^{i-\ell-2}\|\boldsymbol{g}_{\ell+1}\|$$

$$\leq \eta\nu\left(i\|\boldsymbol{F}\boldsymbol{z}_0\| + \delta_i\right) + \eta^2 L\nu^2\sum_{\ell=0}^{i-2}e^2\left((\ell+1)\|\boldsymbol{F}\boldsymbol{z}_0\| + \delta_{\ell+1}\right)$$

$$\leq \eta\nu\left(i\|\boldsymbol{F}\boldsymbol{z}_0\| + \delta_i\right) + \frac{\eta\nu^2 e^2 i(i-1)}{2n}\|\boldsymbol{F}\boldsymbol{z}_0\| + \eta^2 L\nu^2 e^2\Sigma_{i-1}.$$

Similarly, from (39) we get

$$\|\boldsymbol{w}_i - \boldsymbol{z}_0\| \leq \frac{\eta}{2}\|\boldsymbol{g}_{i+1}\| + \frac{\eta}{2}\left(1 + 2\eta L + \frac{\eta^2 L^2}{2}\right)\|\boldsymbol{g}_i\|$$

$$+ \eta^2 L\left(1 + \frac{\eta L}{2}\right)^3\sum_{\ell=0}^{i-2}\left(1 + \eta L + \frac{\eta^2 L^2}{2}\right)^{i-\ell-2}\|\boldsymbol{g}_{\ell+1}\|$$

$$\leq \frac{\eta}{2}\left((i+1)\|\boldsymbol{F}\boldsymbol{z}_0\| + \delta_{i+1}\right) + \frac{\eta}{2}\left(1 + \frac{2}{n} + \frac{1}{2n^2}\right)\left(i\|\boldsymbol{F}\boldsymbol{z}_0\| + \delta_i\right)$$

$$+ \eta^2 L\nu^3\sum_{\ell=0}^{i-2}e^2\left((\ell+1)\|\boldsymbol{F}\boldsymbol{z}_0\| + \delta_{\ell+1}\right)$$

$$\leq \frac{\eta}{2}(1 + 2i\nu^2)\|\boldsymbol{F}\boldsymbol{z}_0\| + \frac{\eta\nu^3 e^2 i(i-1)}{2n}\|\boldsymbol{F}\boldsymbol{z}_0\| + \frac{\eta}{2}\delta_{i+1} + \frac{\eta(2\nu^2 - 1)}{2}\delta_i + \eta^2 L\nu^3 e^2\Sigma_{i-1}.$$

Finally, applying generalized Young's inequality on (39) we get

$$\|\boldsymbol{w}_i - \boldsymbol{z}_0\|^2 \le \frac{3\eta^2}{4} \|\boldsymbol{g}_{i+1}\|^2 + \frac{3\eta^2}{4} \left(1 + 2\eta L + \frac{\eta^2 L^2}{2}\right)^2 \|\boldsymbol{g}_i\|^2$$

$$+ 3 \left(\eta^2 L \left(1 + \frac{\eta L}{2}\right)^3 \sum_{\ell=0}^{i-2} \left(1 + \eta L + \frac{\eta^2 L^2}{2}\right)^{i-\ell-2} \|\boldsymbol{g}_{\ell+1}\|\right)^2.$$

Using generalized Young's inequality once more on the last term gives us

$$3 \left(\eta^2 L \left(1 + \frac{\eta L}{2}\right)^3 \sum_{\ell=0}^{i-2} \left(1 + \eta L + \frac{\eta^2 L^2}{2}\right)^{i-\ell-2} \|\boldsymbol{g}_{\ell+1}\|\right)^2 \le 3 \left(\frac{\eta \nu^3 e^2}{n} \sum_{\ell=0}^{i-2} \|\boldsymbol{g}_{\ell+1}\|\right)^2$$

$$\le \frac{3\eta^2 \nu^6 e^4 (i-1)}{n^2} \sum_{\ell=0}^{i-2} \|\boldsymbol{g}_{\ell+1}\|^2.$$

Plugging this back yields

$$\|\boldsymbol{w}_i - \boldsymbol{z}_0\|^2 \le \frac{3\eta^2}{4} \|\boldsymbol{g}_{i+1}\|^2 + \frac{3\eta^2}{4} \left(1 + 2\eta L + \frac{\eta^2 L^2}{2}\right)^2 \|\boldsymbol{g}_i\|^2 + \frac{3\eta^2 \nu^6 e^4 (i-1)}{n^2} \sum_{\ell=0}^{i-2} \|\boldsymbol{g}_{\ell+1}\|^2$$

$$\le \frac{3\eta^2}{4} \left(2(i+1)^2 \|\boldsymbol{F}\boldsymbol{z}_0\|^2 + 2\delta_{i+1}^2\right) + \frac{3\eta^2}{4} \left(2\nu^2 - 1\right)^2 \left(2i^2 \|\boldsymbol{F}\boldsymbol{z}_0\|^2 + 2\delta_i^2\right)$$

$$+ \frac{3\eta^2 \nu^6 e^4 (i-1)}{n^2} \sum_{\ell=0}^{i-2} \left(2(\ell+1)^2 \|\boldsymbol{F}\boldsymbol{z}_0\|^2 + 2\delta_{\ell+1}^2\right)$$

$$\le \frac{3\eta^2}{2} \left((i+1)^2 \|\boldsymbol{F}\boldsymbol{z}_0\|^2 + \delta_{i+1}^2\right) + \frac{3\eta^2}{2} \left(2\nu^2 - 1\right)^2 \left(i^2 \|\boldsymbol{F}\boldsymbol{z}_0\|^2 + \delta_i^2\right)$$

$$+ \frac{\eta^2 \nu^6 e^4 i(i-1)^2 (2i-1)}{n^2} \|\boldsymbol{F}\boldsymbol{z}_0\|^2 + \frac{6\eta^2 \nu^6 e^4 (i-1)}{n^2} \Psi_{i-1}.$$

Now the claimed inequalities can be obtained simply by rearranging the terms appropriately. $\square$

**Proposition E.4.** *Suppose that either SEG-RR or SEG-FF is used with $\alpha = \beta = \eta < \frac{1}{nL}$, and let $\tilde{\nu} := 1 + \frac{1}{n}$. Then for any $i = 1, \ldots, 2n - 1$ we have the bounds*

$$\|\boldsymbol{z}_i - \boldsymbol{z}_0\| \le \left(\eta \tilde{\nu} i + \frac{16\eta \tilde{\nu}^2 i(i-1)}{n}\right) \|\boldsymbol{F}\boldsymbol{z}_0\| + \eta \tilde{\nu} \delta_i + 32\eta^2 L \tilde{\nu}^2 \Sigma_{i-1},$$

$$\|\boldsymbol{w}_i - \boldsymbol{z}_0\| \le \eta \left(1 + i\tilde{\nu}^2 + \frac{16\tilde{\nu}^3 i(i-1)}{n}\right) \|\boldsymbol{F}\boldsymbol{z}_0\| + \eta \delta_{i+1} + \eta(\tilde{\nu}^2 - 1)\delta_i + 32\eta^2 L \tilde{\nu}^3 \Sigma_{i-1},$$

$$\|\boldsymbol{w}_i - \boldsymbol{z}_0\|^2 \le \left(6\eta^2(i+1)^2 + \frac{6\eta^2 (1+\tilde{\nu})^2 i^2}{n^2} + \frac{1024\eta^2 \tilde{\nu}^6 i(i-1)^2 (2i-1)}{n^2}\right) \|\boldsymbol{F}\boldsymbol{z}_0\|^2$$

$$+ 6\eta^2 \delta_{i+1}^2 + \frac{6\eta^2 (1+\tilde{\nu})^2}{n^2} \delta_i^2 + \frac{6144\eta^2 \tilde{\nu}^6 (i-1)}{n^2} \Psi_{i-1}.$$

*Proof.* One can verify that $x \mapsto (1 + \frac{4}{3x})^x$ increases on $x \ge 3$ and is bounded above by $e^{4/3} < 4$. With noting that $(1 + \frac{1}{1} + \frac{1}{1^2})^1 = 3 < 4$, $(1 + \frac{1}{2} + \frac{1}{2^2})^2 = \frac{49}{16} < 4$, and $1 + \frac{1}{x} + \frac{1}{x^2} \le 1 + \frac{4}{3x}$ whenever $x \ge 3$, we see that for all $0 \le \ell < i \le 2n$ it holds that

$$\left(1 + \eta L + \eta^2 L^2\right)^{i-\ell-2} \le \left(1 + \frac{1}{n} + \frac{1}{n^2}\right)^{2n} \le 4^2 = 16.$$

Also, we have

$$2 + 2\eta L + \eta^2 L^2 \le 2 + \frac{2}{n} + \frac{1}{n^2} = 1 + \tilde{\nu}^2 \le 2\tilde{\nu}^2.$$

Applying the definitions (35) and (36) on (38) and then substituting $\xi = 1$ we get

$$\|\mathbf{z}_i - \mathbf{z}_0\| \leq \eta\left(1 + \eta L\right)\|\mathbf{g}_i\| + \eta^2 L\left(2 + 2\eta L + \eta^2 L^2\right)\sum_{\ell=0}^{i-2}\left(1 + \eta L + \eta^2 L^2\right)^{i-\ell-2}\|\mathbf{g}_{\ell+1}\|$$

$$\leq \eta\tilde{\nu}\left(i\|\mathbf{F}\mathbf{z}_0\| + \delta_i\right) + 2\eta^2 L\tilde{\nu}^2\sum_{\ell=0}^{i-2}16\left((\ell+1)\|\mathbf{F}\mathbf{z}_0\| + \delta_{\ell+1}\right)$$

$$\leq \eta\tilde{\nu}\left(i\|\mathbf{F}\mathbf{z}_0\| + \delta_i\right) + \frac{16\eta\tilde{\nu}^2 i(i-1)}{n}\|\mathbf{F}\mathbf{z}_0\| + 32\eta^2 L\tilde{\nu}^2\Sigma_{i-1}.$$

Similarly, from (39) we get

$$\|\mathbf{w}_i - \mathbf{z}_0\| \leq \eta\|\mathbf{g}_{i+1}\| + \eta^2 L\left(2 + \eta L\right)\|\mathbf{g}_i\|$$

$$+ \eta^2 L(1 + \eta L)\left(2 + 2\eta L + \eta^2 L^2\right)\sum_{\ell=0}^{i-2}\left(1 + \eta L + \eta^2 L^2\right)^{i-\ell-2}\|\mathbf{g}_{\ell+1}\|.$$

$$\leq \eta\left((i+1)\|\mathbf{F}\mathbf{z}_0\| + \delta_{i+1}\right) + \frac{\eta}{n}\left(2 + \frac{1}{n}\right)\left(i\|\mathbf{F}\mathbf{z}_0\| + \delta_i\right)$$

$$+ \eta^2 L\tilde{\nu}\left(2\tilde{\nu}^2\right)\sum_{\ell=0}^{i-2}16\left(\ell\|\mathbf{F}\mathbf{z}_0\| + \delta_\ell\right).$$

$$\leq \eta(1 + i\tilde{\nu}^2)\|\mathbf{F}\mathbf{z}_0\| + \frac{16\eta\tilde{\nu}^3 i(i-1)}{n}\|\mathbf{F}\mathbf{z}_0\| + \eta\delta_{i+1} + \eta(\tilde{\nu}^2 - 1)\delta_i + 32\eta^2 L\tilde{\nu}^3\Sigma_{i-1}.$$

Finally, applying Young's inequality on (39) we get

$$\|\mathbf{w}_i - \mathbf{z}_0\|^2 \leq 3\eta^2\|\mathbf{g}_{i+1}\|^2 + \frac{3\eta^2\left(2 + \eta L\right)^2}{n^2}\|\mathbf{g}_i\|$$

$$+ 3\left(\eta^2 L(1 + \eta L)\left(2 + 2\eta L + \eta^2 L^2\right)\sum_{\ell=0}^{i-2}\left(1 + \eta L + \eta^2 L^2\right)^{i-\ell-2}\|\mathbf{g}_{\ell+1}\|\right)^2$$

$$\leq 3\eta^2\|\mathbf{g}_{i+1}\|^2 + \frac{3\eta^2\left(2 + \eta L\right)^2}{n^2}\|\mathbf{g}_i\| + 3\left(\frac{2\eta\tilde{\nu}^3}{n}\sum_{\ell=0}^{i-2}16\|\mathbf{g}_{\ell+1}\|\right)^2.$$

Using Young's inequality once more on the last term gives us

$$3\left(\frac{32\eta\tilde{\nu}^3}{n}\sum_{\ell=0}^{i-2}\|\mathbf{g}_{\ell+1}\|\right)^2 \leq \frac{3072\eta^2\tilde{\nu}^6(i-1)}{n^2}\sum_{\ell=0}^{i-2}\|\mathbf{g}_{\ell+1}\|^2.$$

Plugging this back yields

$$\|\mathbf{w}_i - \mathbf{z}_0\|^2 \leq 3\eta^2\|\mathbf{g}_{i+1}\|^2 + \frac{3\eta^2\left(2 + \eta L\right)^2}{n^2}\|\mathbf{g}_i\| + \frac{3072\eta^2\tilde{\nu}^6(i-1)}{n^2}\sum_{\ell=0}^{i-2}\|\mathbf{g}_{\ell+1}\|^2$$

$$\leq 6\eta^2\left((i+1)^2\|\mathbf{F}\mathbf{z}_0\|^2 + \delta_{i+1}^2\right) + \frac{6\eta^2\left(2 + \eta L\right)^2}{n^2}\left(i^2\|\mathbf{F}\mathbf{z}_0\|^2 + \delta_i^2\right)$$

$$+ \frac{6144\eta^2\tilde{\nu}^6(i-1)}{n^2}\sum_{\ell=0}^{i-2}\left((\ell+1)^2\|\mathbf{F}\mathbf{z}_0\|^2 + \delta_{\ell+1}^2\right)$$

$$\leq 6\eta^2\left((i+1)^2\|\mathbf{F}\mathbf{z}_0\|^2 + \delta_{i+1}^2\right) + \frac{6\eta^2\left(1 + \tilde{\nu}\right)^2}{n^2}\left(i^2\|\mathbf{F}\mathbf{z}_0\|^2 + \delta_i^2\right)$$

$$+ \frac{1024\eta^2\tilde{\nu}^6 i(i-1)^2(2i-1)}{n^2}\|\mathbf{F}\mathbf{z}_0\|^2 + \frac{6144\eta^2\tilde{\nu}^6(i-1)}{n^2}\Psi_{i-1}.$$

Now the claimed inequalities can be obtained simply by rearranging the terms appropriately. $\qquad\square$

Let us now derive the upper bounds for the quantities related to $\delta_j$ and $\Sigma_j$, defined in (35) and (36) respectively, using the upper bound of the variance of saddle gradients (4).

**Lemma E.5.** *For any $j = 1, \ldots, 2n$, it deterministically holds that*

$$\delta_j \leq n(\rho \|\boldsymbol{F}\boldsymbol{z}_0\| + \sigma). \tag{53}$$

*Proof.* For any set of indices $\mathcal{J} \subset \{0, \ldots, n-1\}$, by Assumption 3.4 it holds that

$$\sum_{i \in \mathcal{J}} \|\boldsymbol{F}_i \boldsymbol{z}_0 - \boldsymbol{F}\boldsymbol{z}_0\|^2 \leq \sum_{i=0}^{n-1} \|\boldsymbol{F}_i \boldsymbol{z}_0 - \boldsymbol{F}\boldsymbol{z}_0\|^2 \leq n(\rho \|\boldsymbol{F}\boldsymbol{z}_0\| + \sigma)^2.$$

Hence, for any $j = 1, \ldots, n$ we have

$$\begin{aligned}
\|\boldsymbol{g}_j - j\boldsymbol{F}\boldsymbol{z}_0\|^2 &= \left\|\sum_{i=0}^{j-1} \boldsymbol{F}_i \boldsymbol{z}_0 - j\boldsymbol{F}\boldsymbol{z}_0\right\|^2 \\
&\leq j \sum_{i=0}^{j-1} \|\boldsymbol{F}_i \boldsymbol{z}_0 - \boldsymbol{F}\boldsymbol{z}_0\|^2 \\
&\leq jn(\rho \|\boldsymbol{F}\boldsymbol{z}_0\| + \sigma)^2 \\
&\leq n^2(\rho \|\boldsymbol{F}\boldsymbol{z}_0\| + \sigma)^2,
\end{aligned}$$

and for any $j = n+1, \ldots, 2n$ we have

$$\begin{aligned}
\|\boldsymbol{g}_j - j\boldsymbol{F}\boldsymbol{z}_0\|^2 &= \left\|\sum_{i=0}^{n-1} \boldsymbol{F}_i \boldsymbol{z}_0 + \sum_{i=n}^{j-1} \boldsymbol{F}_i \boldsymbol{z}_0 - j\boldsymbol{F}\boldsymbol{z}_0\right\|^2 \\
&= \left\|\sum_{i=n}^{j-1} \boldsymbol{F}_i \boldsymbol{z}_0 - (j-n)\boldsymbol{F}\boldsymbol{z}_0\right\|^2 \\
&= \left\|\sum_{i=2n-j}^{n-1} \boldsymbol{F}_i \boldsymbol{z}_0 - (j-n)\boldsymbol{F}\boldsymbol{z}_0\right\|^2 \\
&\leq (j-n) \sum_{i=0}^{j-1} \|\boldsymbol{F}_i \boldsymbol{z}_0 - \boldsymbol{F}\boldsymbol{z}_0\|^2 \\
&\leq n^2(\rho \|\boldsymbol{F}\boldsymbol{z}_0\| + \sigma)^2.
\end{aligned} \tag{54}$$

Therefore, in any case we have

$$\|\boldsymbol{g}_j - j\boldsymbol{F}\boldsymbol{z}_0\|^2 \leq n^2(\rho \|\boldsymbol{F}\boldsymbol{z}_0\| + \sigma)^2.$$

Taking square roots on both sides gives us the desired bound. $\qquad\square$

**Lemma E.6.** *For any $j = 1, \ldots, 2n$, it holds that*

$$\mathbb{E}_\tau[\delta_j^2] \leq \frac{n(\rho \|\boldsymbol{F}\boldsymbol{z}_0\| + \sigma)^2}{2}, \tag{55}$$

*Proof.* If $n = 1$ then the left hand side is always 0, so there is nothing to show. So, we may assume that $n \geq 2$. Then, for any $j = 1, \ldots, n$, using Lemma 1 in [35] we obtain

$$\mathbb{E}_\tau \left\|\frac{1}{j}\boldsymbol{g}_j - \boldsymbol{F}\boldsymbol{z}_0\right\|^2 \leq \frac{n-j}{j(n-1)}(\rho \|\boldsymbol{F}\boldsymbol{z}_0\| + \sigma)^2.$$

Multiplying both sides by $j^2$ and applying AM-GM inequality leads to

$$\mathbb{E}_\tau \|\boldsymbol{g}_j - j\boldsymbol{F}\boldsymbol{z}_0\|^2 \leq \frac{j(n-j)}{n-1}(\rho \|\boldsymbol{F}\boldsymbol{z}_0\| + \sigma)^2 \leq \frac{n^2}{4(n-1)}(\rho \|\boldsymbol{F}\boldsymbol{z}_0\| + \sigma)^2 \leq \frac{n}{2}(\rho \|\boldsymbol{F}\boldsymbol{z}_0\| + \sigma)^2.$$

Meanwhile, for $j = n+1, \ldots, 2n$, following the first few steps in (54) we get

$$\|\boldsymbol{g}_j - j\boldsymbol{F}\boldsymbol{z}_0\|^2 = \left\|\sum_{i=2n-j}^{n-1} \boldsymbol{F}_i \boldsymbol{z}_0 - (j-n)\boldsymbol{F}\boldsymbol{z}_0\right\|^2$$

Here, once more applying Lemma 1 of [35], we get

$$
\mathbb{E}_\tau \left\| \boldsymbol{g}_j - j\boldsymbol{F}\boldsymbol{z}_0 \right\|^2 = \mathbb{E}_\tau \left\| \sum_{i=2n-j}^{n-1} \boldsymbol{F}_i \boldsymbol{z}_0 - (j-n)\boldsymbol{F}\boldsymbol{z}_0 \right\|^2
$$

$$
= (j-n)^2 \, \mathbb{E}_\tau \left\| \frac{1}{j-n} \sum_{i=2n-j}^{n-1} \boldsymbol{F}_i \boldsymbol{z}_0 - \boldsymbol{F}\boldsymbol{z}_0 \right\|^2
$$

$$
\leq (j-n)^2 \cdot \frac{n-(j-n)}{(j-n)(n-1)} (\rho \left\| \boldsymbol{F}\boldsymbol{z}_0 \right\| + \sigma)^2
$$

$$
\leq \frac{(j-n)(2n-j)}{n-1} (\rho \left\| \boldsymbol{F}\boldsymbol{z}_0 \right\| + \sigma)^2.
$$

Using AM-GM inequality on the last line gives us

$$
\mathbb{E}_\tau \left\| \boldsymbol{g}_j - j\boldsymbol{F}\boldsymbol{z}_0 \right\|^2 \leq \frac{(j-n)(2n-j)}{n-1} (\rho \left\| \boldsymbol{F}\boldsymbol{z}_0 \right\| + \sigma)^2 \leq \frac{n^2}{4(n-1)} (\rho \left\| \boldsymbol{F}\boldsymbol{z}_0 \right\| + \sigma)^2 \leq \frac{n}{2} (\rho \left\| \boldsymbol{F}\boldsymbol{z}_0 \right\| + \sigma)^2.
$$

Thus, for any case, we have (55). $\square$

**Lemma E.7.** *For any $k, \ell \in \{0, 1, \ldots, 2n\}$, it holds that*

$$
\mathbb{E}_\tau [\Sigma_k \Sigma_\ell] \leq \frac{k\ell n(\rho \left\| \boldsymbol{F}\boldsymbol{z}_0 \right\| + \sigma)^2}{2}. \tag{56}
$$

*Proof.* Expanding the product $\Sigma_k \Sigma_\ell$ and writing in terms of $\delta$, we get

$$
\Sigma_k \Sigma_\ell = \left( \sum_{i=1}^k \delta_i \right) \left( \sum_{j=1}^\ell \delta_j \right) = \sum_{i=1}^k \sum_{j=1}^\ell \delta_i \delta_j
$$

$$
\leq \sum_{i=1}^k \sum_{j=1}^\ell \frac{\delta_i^2 + \delta_j^2}{2}
$$

where the last line follows from the AM-GM inequality. Taking the expectation with respect to $\tau$ and using the bound from Lemma E.6, we obtain

$$
\mathbb{E}_\tau [\Sigma_k \Sigma_\ell] \leq \frac{1}{2} \sum_{i=1}^k \sum_{j=1}^\ell \left( \mathbb{E}_\tau [\delta_i^2] + \mathbb{E}_\tau [\delta_j^2] \right)
$$

$$
\leq \frac{1}{2} \sum_{i=1}^k \sum_{j=1}^\ell \left( \frac{n(\rho \left\| \boldsymbol{F}\boldsymbol{z}_0 \right\| + \sigma)^2}{2} + \frac{n(\rho \left\| \boldsymbol{F}\boldsymbol{z}_0 \right\| + \sigma)^2}{2} \right)
$$

$$
= \frac{k\ell n(\rho \left\| \boldsymbol{F}\boldsymbol{z}_0 \right\| + \sigma)^2}{2}
$$

which is exactly the claimed. $\square$

**Lemma E.8.** *For any $k, \ell \in \{0, 1, \ldots, 2n\}$, it holds that*

$$
\mathbb{E}_\tau \left[ \left( \sum_{i=1}^k \Sigma_i \right) \left( \sum_{j=1}^\ell \Sigma_j \right) \right] \leq \frac{k(k+1)\ell(\ell+1)n(\rho \left\| \boldsymbol{F}\boldsymbol{z}_0 \right\| + \sigma)^2}{8}. \tag{57}
$$

*Proof.* Expanding the product in the left hand side of (57) and applying (56), we get

$$\mathbb{E}_\tau\left[\left(\sum_{i=1}^k \Sigma_i\right)\left(\sum_{j=1}^\ell \Sigma_j\right)\right] = \mathbb{E}_\tau\left[\sum_{i=1}^k\sum_{j=1}^\ell \Sigma_i\Sigma_j\right] = \sum_{i=1}^k\sum_{j=1}^\ell \mathbb{E}_\tau[\Sigma_i\Sigma_j]$$

$$\leq \sum_{i=1}^k\sum_{j=1}^\ell \frac{ijn(\rho\,\|\boldsymbol{F}\boldsymbol{z}_0\| + \sigma)^2}{2}$$

$$\leq \frac{k(k+1)\ell(\ell+1)n(\rho\,\|\boldsymbol{F}\boldsymbol{z}_0\| + \sigma)^2}{8}.\qquad\square$$

## E.2 Upper Bounds of the Within-Epoch Errors

The full proof of Theorem E.1 is quite long and technical, so we divide it into several parts. First we show that (31) and (32) holds with $a = 3$ when SEG-FFA is in use. Then we show that Theorem E.1 also holds for SEG-FF in Appendix E.2.3, and for SEG-RR in Appendix E.2.4.

Throughout the remaining of this section, we always assume that the variance of the saddle gradients satisfies (4).

### E.2.1 Proof of Equation (31) for SEG-FFA

In this section we prove the following.

**Theorem E.9.** *Say we use SEG-FFA. Then, as long as the stepsize used in an epoch satisfies $\eta < \frac{1}{nL}$, it holds that*

$$\|\boldsymbol{r}\| \leq \eta^3 n^3 C_{1A}\,\|\boldsymbol{F}\boldsymbol{z}_0\| + \eta^3 n^3 D_{1A}\,\|\boldsymbol{F}\boldsymbol{z}_0\|^2 + \eta^3 n^3 V_{1A} \tag{58}$$

*for constants*

$$C_{1A} := L^2\left(\frac{1}{2}\left(1 + \frac{2e^2}{3}\right) + \frac{6 + e^2}{3} + 15\rho\right), \tag{59}$$

$$D_{1A} := M\left(\frac{83}{4} + \frac{24e^4}{5} + \rho^2\left(\frac{243}{16} + 27e^4\right)\right), \tag{60}$$

$$V_{1A} := M\sigma^2\left(\frac{243}{16} + 27e^4\right) + 15L^2\sigma. \tag{61}$$

We first list the intermediate results. The actual proof of Theorem E.9 is in page 43, at the end of this section.

**Proposition E.10.** *For using SEG-FFA, the within-epoch update $\boldsymbol{z}^\sharp$ as given by (12) satisfies*

$$\boldsymbol{z}^\sharp = \boldsymbol{z}_0 - n\eta\boldsymbol{F}(\boldsymbol{z}_0 - n\eta\boldsymbol{F}\boldsymbol{z}_0) + \boldsymbol{r}$$

*where we denote*

$$\boldsymbol{r} := n\eta\boldsymbol{F}(\boldsymbol{z}_0 - n\eta\boldsymbol{F}\boldsymbol{z}_0) - n\eta\boldsymbol{F}\boldsymbol{z}_0 + n^2\eta^2 D\boldsymbol{F}(\boldsymbol{z}_0)\boldsymbol{F}\boldsymbol{z}_0 \tag{62a}$$

$$- \frac{\eta}{2}\sum_{j=0}^{2n-1}\left(\boldsymbol{F}_j\boldsymbol{w}_j - \boldsymbol{F}_j\boldsymbol{z}_0 - D\boldsymbol{F}_j(\boldsymbol{z}_0)(\boldsymbol{w}_j - \boldsymbol{z}_0)\right) \tag{62b}$$

$$+ \frac{\eta^2}{4}\sum_{j=0}^{2n-1} D\boldsymbol{F}_j(\boldsymbol{z}_0)(\boldsymbol{F}_j\boldsymbol{z}_j - \boldsymbol{F}_j\boldsymbol{z}_0) \tag{62c}$$

$$+ \frac{\eta^2}{2}\sum_{j=0}^{2n-1} D\boldsymbol{F}_j(\boldsymbol{z}_0)\sum_{k=0}^{j-1}(\boldsymbol{F}_k\boldsymbol{w}_k - \boldsymbol{F}_k\boldsymbol{z}_0). \tag{62d}$$

*Proof.* Setting $\alpha = \eta/2$, $\beta = \eta$, and $\theta = 1$ in (28), we get

$$\boldsymbol{z}^\sharp = \boldsymbol{z}_0 - \frac{\eta}{2}\sum_{j=0}^{2n-1}\boldsymbol{F}_j\boldsymbol{z}_0 + \frac{\eta^2}{4}\sum_{j=0}^{2n-1} D\boldsymbol{F}_j(\boldsymbol{z}_0)\boldsymbol{F}_j\boldsymbol{z}_0 + \frac{\eta^2}{2}\sum_{0\leq k<j\leq 2n-1} D\boldsymbol{F}_j(\boldsymbol{z}_0)\boldsymbol{F}_k\boldsymbol{z}_0 + \frac{1}{2}\boldsymbol{\epsilon}_{2n} \tag{63}$$

where $\epsilon_{2n}$ is defined as in (30). Recall that $\boldsymbol{F}_i = \boldsymbol{F}_{2n-1-i}$ for all $i = 0, 1, \ldots, 2n-1$, and moreover, $\sum_{i=0}^{n-1} \boldsymbol{F}_i = \sum_{i=n}^{2n-1} \boldsymbol{F}_i = n\boldsymbol{F}$. Thus, the first sum in the above is equal to $2n\boldsymbol{F}\boldsymbol{z}_0$, and the second sum is equal to $2\sum_{j=0}^{n-1} D\boldsymbol{F}_j(\boldsymbol{z}_0)\boldsymbol{F}_j\boldsymbol{z}_0$. For the last sum, observe that

$$
\begin{aligned}
\sum_{0 \le k < j \le 2n-1} D\boldsymbol{F}_j(\boldsymbol{z}_0)\boldsymbol{F}_k\boldsymbol{z}_0 ={}& \sum_{0 \le k < j \le n-1} D\boldsymbol{F}_j(\boldsymbol{z}_0)\boldsymbol{F}_k\boldsymbol{z}_0 + \sum_{n \le k < j \le 2n-1} D\boldsymbol{F}_j(\boldsymbol{z}_0)\boldsymbol{F}_k\boldsymbol{z}_0 \\
& + \sum_{\substack{0 \le k \le n-1 \\ n \le j \le 2n-1}} D\boldsymbol{F}_j(\boldsymbol{z}_0)\boldsymbol{F}_k\boldsymbol{z}_0 \\
={}& \sum_{0 \le k < j \le n-1} D\boldsymbol{F}_j(\boldsymbol{z}_0)\boldsymbol{F}_k\boldsymbol{z}_0 + \sum_{n-1 \ge k > j \ge 0} D\boldsymbol{F}_j(\boldsymbol{z}_0)\boldsymbol{F}_k\boldsymbol{z}_0 \\
& + \sum_{\substack{0 \le k \le n-1 \\ n-1 \ge j \ge 0}} D\boldsymbol{F}_j(\boldsymbol{z}_0)\boldsymbol{F}_k\boldsymbol{z}_0 \\
={}& 2\sum_{k \ne j} D\boldsymbol{F}_j(\boldsymbol{z}_0)\boldsymbol{F}_k\boldsymbol{z}_0 + \sum_{j=0}^{n-1} D\boldsymbol{F}_j(\boldsymbol{z}_0)\boldsymbol{F}_j\boldsymbol{z}_0.
\end{aligned}
$$

Hence, (63) is equivalent to

$$
\begin{aligned}
\boldsymbol{z}^{\sharp} &= \boldsymbol{z}_0 - n\eta\boldsymbol{F}\boldsymbol{z}_0 + \frac{\eta^2}{2}\sum_{j=0}^{n-1} D\boldsymbol{F}_j(\boldsymbol{z}_0)\boldsymbol{F}_j\boldsymbol{z}_0 + \frac{\eta^2}{2}\sum_{0 \le k < j \le 2n-1} D\boldsymbol{F}_j(\boldsymbol{z}_0)\boldsymbol{F}_k\boldsymbol{z}_0 + \frac{1}{2}\epsilon_{2n} \\
&= \boldsymbol{z}_0 - n\eta\boldsymbol{F}\boldsymbol{z}_0 + \eta^2\sum_{j=0}^{n-1} D\boldsymbol{F}_j(\boldsymbol{z}_0)\boldsymbol{F}_j\boldsymbol{z}_0 + \eta^2\sum_{k \ne j} D\boldsymbol{F}_j(\boldsymbol{z}_0)\boldsymbol{F}_k\boldsymbol{z}_0 + \frac{1}{2}\epsilon_{2n} \\
&= \boldsymbol{z}_0 - n\eta\boldsymbol{F}\boldsymbol{z}_0 + \eta^2\left(\sum_{j=0}^{n-1} D\boldsymbol{F}_j(\boldsymbol{z}_0)\right)\left(\sum_{j=0}^{n-1} \boldsymbol{F}_j\boldsymbol{z}_0\right) + \frac{1}{2}\epsilon_{2n} \\
&= \boldsymbol{z}_0 - n\eta\boldsymbol{F}\boldsymbol{z}_0 + n^2\eta^2 D\boldsymbol{F}(\boldsymbol{z}_0)\boldsymbol{F}\boldsymbol{z}_0 + \frac{1}{2}\epsilon_{2n}.
\end{aligned}
$$

Observing that the terms (62b), (62c), and (62d) add up to $\frac{1}{2}\epsilon_{2n}$ completes the proof. $\qquad\square$

**Proposition E.11.** *Suppose that $\eta < \frac{1}{nL}$, and let $\nu := 1 + \frac{1}{2n}$. Then the noise term satisfies the bound*

$$
\begin{aligned}
\|\boldsymbol{r}\| \le{}& \eta^3 n^3 L^2 \|\boldsymbol{F}\boldsymbol{z}_0\| \left(\frac{1}{2n}\left(1 + \frac{2e^2}{3}\right) + \frac{4\nu + e^2}{3}\right) \\
& + \eta^3 n^3 M \|\boldsymbol{F}\boldsymbol{z}_0\|^2 \left(\frac{1}{2} + 4\nu^4 + \frac{16\nu e^4}{5}\right) \\
& + \frac{3\eta^3 M}{8}\left(\Psi_{2n} + (2\nu^2 - 1)^2\Psi_{2n-1} + \frac{4\nu^6 e^4}{n^2}\sum_{j=1}^{2n-2} j\Psi_j\right) \\
& + \frac{\eta^3 L^2(\nu+1)}{4}\Sigma_{2n-1} + \frac{\eta^3 L^2\nu^2(1 + \eta Le^2)}{2}\sum_{j=1}^{2n-2}\Sigma_j + \frac{\eta^4 L^3\nu^3 e^2}{2}\sum_{k=1}^{2n-2}(2n - k - 1)\Sigma_{k-1}.
\end{aligned}
$$

*Proof.* We bound each line in equation (62). For (62a), we use Lemma C.6 to get

$$
\begin{aligned}
\left\|n\eta\boldsymbol{F}(\boldsymbol{z}_0 - n\eta\boldsymbol{F}\boldsymbol{z}_0) - n\eta\boldsymbol{F}\boldsymbol{z}_0 + n^2\eta^2 D\boldsymbol{F}(\boldsymbol{z}_0)\boldsymbol{F}\boldsymbol{z}_0\right\| &\le \frac{n\eta M}{2}\|-n\eta\boldsymbol{F}\boldsymbol{z}_0\|^2 \\
&= \frac{n^3\eta^3 M}{2}\|\boldsymbol{F}\boldsymbol{z}_0\|^2.
\end{aligned}
$$

In bounding the remaining three lines we repeatedly use the bounds obtained in Proposition E.3. We will also use the following bounds, which follows from (33), (35), and Young's inequality:

$$\|\boldsymbol{w}_0 - \boldsymbol{z}_0\| = \frac{\eta}{2}\|\boldsymbol{g}_1\| \leq \frac{\eta}{2}\|\boldsymbol{F}\boldsymbol{z}_0\| + \frac{\eta}{2}\delta_1,$$

$$\|\boldsymbol{w}_0 - \boldsymbol{z}_0\|^2 = \frac{\eta^2}{4}\|\boldsymbol{g}_1\|^2 \leq \frac{\eta^2}{2}\|\boldsymbol{F}\boldsymbol{z}_0\|^2 + \frac{\eta^2}{2}\delta_1^2.$$

For (62b), observe that Lemma C.6 gives us

$$\|\boldsymbol{F}_j\boldsymbol{w}_j - \boldsymbol{F}_j\boldsymbol{z}_0 - D\boldsymbol{F}_j(\boldsymbol{z}_0)(\boldsymbol{w}_j - \boldsymbol{z}_0)\| \leq \frac{M}{2}\|\boldsymbol{w}_j - \boldsymbol{z}_0\|^2.$$

Thus, by using the bound obtained in Proposition E.3, we get

$$
\left\| -\frac{\eta}{2}\sum_{j=0}^{2n-1}\left(\boldsymbol{F}_j\boldsymbol{w}_j - \boldsymbol{F}_j\boldsymbol{z}_0 - D\boldsymbol{F}_j(\boldsymbol{z}_0)(\boldsymbol{w}_j - \boldsymbol{z}_0)\right)\right\|
$$

$$
\leq \frac{\eta}{2}\sum_{j=0}^{2n-1}\|\boldsymbol{F}_j\boldsymbol{w}_j - \boldsymbol{F}_j\boldsymbol{z}_0 - D\boldsymbol{F}_j(\boldsymbol{z}_0)(\boldsymbol{w}_j - \boldsymbol{z}_0)\|
$$

$$
\leq \frac{\eta M}{4}\sum_{j=0}^{2n-1}\|\boldsymbol{w}_j - \boldsymbol{z}_0\|^2
$$

$$
\leq \frac{\eta M}{4}\sum_{j=1}^{2n-1}\left(\frac{3\eta^2(j+1)^2}{2} + \frac{3\eta^2(2\nu^2-1)^2 j^2}{2} + \frac{\eta^2\nu^6 e^4 j(j-1)^2(2j-1)}{n^2}\right)\|\boldsymbol{F}\boldsymbol{z}_0\|^2
$$

$$
+ \frac{\eta M}{4}\sum_{j=1}^{2n-1}\left(\frac{3\eta^2}{2}\delta_{j+1}^2 + \frac{3\eta^2(2\nu^2-1)^2}{2}\delta_j^2 + \frac{6\eta^2\nu^6 e^4(j-1)}{n^2}\Psi_{j-1}\right)
$$

$$
+ \frac{\eta M}{4}\|\boldsymbol{w}_0 - \boldsymbol{z}_0\|^2
$$

$$
= \frac{\eta M}{4}\left(\frac{\eta^2 n(1+2n)(1+4n) - 3\eta^2}{2} + \frac{\eta^2(2\nu^2-1)^2 n(2n-1)(4n-1)}{2}\right.
$$

$$
\left. + \frac{\eta^2\nu^6 e^4(n-1)(2n-1)(32n^2-42n+11)}{5n}\right)\|\boldsymbol{F}\boldsymbol{z}_0\|^2
$$

$$
+ \frac{3\eta^3 M}{8}(\Psi_{2n} - \delta_1^2) + \frac{3\eta^3 M(2\nu^2-1)^2}{8}\Psi_{2n-1} + \frac{3\eta^3 M\nu^6 e^4}{2n^2}\sum_{j=1}^{2n-1}(j-1)\Psi_{j-1}
$$

$$
+ \frac{\eta M}{4}\left(\frac{\eta^2}{2}\|\boldsymbol{F}\boldsymbol{z}_0\|^2 + \frac{\eta^2}{2}\delta_1^2\right)
$$

$$
\leq \eta^3 n^3 M\left(\nu^2 + (2\nu^2-1)^2 + \frac{16\nu e^4}{5}\right)\|\boldsymbol{F}\boldsymbol{z}_0\|^2
$$

$$
+ \frac{3\eta^3 M}{8}\Psi_{2n} + \frac{3\eta^3 M(2\nu^2-1)^2}{8}\Psi_{2n-1} + \frac{3\eta^3 M\nu^6 e^4}{2n^2}\sum_{j=1}^{2n-2}j\Psi_j
$$

$$
\leq \eta^3 n^3 M\left(4\nu^4 + \frac{16\nu e^4}{5}\right)\|\boldsymbol{F}\boldsymbol{z}_0\|^2 + \frac{3\eta^3 M}{8}\left(\Psi_{2n} + (2\nu^2-1)^2\Psi_{2n-1} + \frac{4\nu^6 e^4}{n^2}\sum_{j=1}^{2n-2}j\Psi_j\right)
$$

where along the derivation we used the inequality

$$\nu^5(n-1)(2n-1)(32n^2-42n+11) \leq 64n^4$$

which holds for all $n \geq 1$. From now on, we will keep on using similar techniques to reduce the exponents of $\nu$, without explicitly stating the inequalities used, but recovering the inequalities that are used should be clear from context.

For (62c), we use $L$-smoothness of $\boldsymbol{F}_j$, and also the fact that it implies $\|D\boldsymbol{F}_j(\boldsymbol{z}_0)\| \leq L$, to get

$$
\left\| \frac{\eta^2}{4} \sum_{j=0}^{2n-1} D\boldsymbol{F}_j(\boldsymbol{z}_0)(\boldsymbol{F}_j\boldsymbol{z}_j - \boldsymbol{F}_j\boldsymbol{z}_0) \right\|
$$

$$
\leq \frac{\eta^2}{4} \sum_{j=0}^{2n-1} \|D\boldsymbol{F}_j(\boldsymbol{z}_0)\| \, \|\boldsymbol{F}_j\boldsymbol{z}_j - \boldsymbol{F}_j\boldsymbol{z}_0\|
$$

$$
\leq \frac{\eta^2 L^2}{4} \sum_{j=0}^{2n-1} \|\boldsymbol{z}_j - \boldsymbol{z}_0\|
$$

$$
\leq \frac{\eta^2 L^2}{4} \sum_{j=1}^{2n-1} \left( \left( \eta\nu j + \frac{\eta\nu^2 e^2 j(j-1)}{2n} \right) \|\boldsymbol{F}\boldsymbol{z}_0\| + \eta\nu\delta_j + \eta^2 L\nu^2 e^2 \Sigma_{j-1} \right)
$$

$$
= \frac{\eta^2 L^2}{4} \left( \eta\nu n(2n-1) + \frac{2\eta\nu^2 e^2 (n-1)(2n-1)}{3} \right) \|\boldsymbol{F}\boldsymbol{z}_0\|
$$

$$
+ \frac{\eta^3 L^2 \nu}{4} \Sigma_{2n-1} + \frac{\eta^4 L^3 \nu^2 e^2}{4} \sum_{j=1}^{2n-1} \Sigma_{j-1}
$$

$$
\leq \frac{\eta^3 n^2 L^2}{2} \left( 1 + \frac{2e^2}{3} \right) \|\boldsymbol{F}\boldsymbol{z}_0\| + \frac{\eta^3 L^2 \nu}{4} \Sigma_{2n-1} + \frac{\eta^4 L^3 \nu^2 e^2}{4} \sum_{j=1}^{2n-2} \Sigma_j.
$$

By the same logic, each summand in (62d) with $j > 0$ can be bounded as

$$
\left\| D\boldsymbol{F}_j(\boldsymbol{z}_0) \sum_{k=0}^{j-1} (\boldsymbol{F}_k\boldsymbol{w}_k - \boldsymbol{F}_k\boldsymbol{z}_0) \right\|
$$

$$
\leq \|D\boldsymbol{F}_j(\boldsymbol{z}_0)\| \sum_{k=0}^{j-1} \|\boldsymbol{F}_k\boldsymbol{w}_k - \boldsymbol{F}_k\boldsymbol{z}_0\|
$$

$$
\leq L^2 \sum_{k=0}^{j-1} \|\boldsymbol{w}_k - \boldsymbol{z}_0\|
$$

$$
\leq L^2 \left( \frac{\eta}{2} \|\boldsymbol{F}\boldsymbol{z}_0\| + \frac{\eta}{2}\delta_1 \right)
$$

$$
+ L^2 \sum_{k=1}^{j-1} \frac{\eta}{2} \left( 1 + 2\nu^2 k + \frac{\nu^3 e^2 k(k-1)}{n} \right) \|\boldsymbol{F}\boldsymbol{z}_0\|
$$

$$
+ L^2 \sum_{k=1}^{j-1} \left( \frac{\eta}{2}\delta_{k+1} + \frac{\eta(2\nu^2 - 1)}{2}\delta_k + \eta^2 L\nu^3 e^2 \Sigma_{k-1} \right)
$$

$$
= \frac{\eta L^2}{2} (\|\boldsymbol{F}\boldsymbol{z}_0\| + \delta_1) + \frac{\eta L^2}{2} \left( j - 1 + \nu^2 j(j-1) + \frac{\nu^3 e^2 j(j-1)(j-2)}{3n} \right) \|\boldsymbol{F}\boldsymbol{z}_0\|
$$

$$
+ \frac{\eta L^2}{2} (\Sigma_j - \delta_1) + \frac{\eta L^2 (2\nu^2 - 1)}{2} \Sigma_{j-1} + \eta^2 L^3 \nu^3 e^2 \sum_{k=1}^{j-1} \Sigma_{k-1}
$$

$$
= \frac{\eta L^2}{2} \left( j + \nu^2 j(j-1) + \frac{\nu^3 e^2 j(j-1)(j-2)}{3n} \right) \|\boldsymbol{F}\boldsymbol{z}_0\|
$$

$$
+ \frac{\eta L^2}{2} \Sigma_j + \frac{\eta L^2 (2\nu^2 - 1)}{2} \Sigma_{j-1} + \eta^2 L^3 \nu^3 e^2 \sum_{k=1}^{j-1} \Sigma_{k-1},
$$

and when $j = 0$ the sum with respect to $k$ becomes an empty sum. Thus, (62d) in total satisfies the bound

$$\left\| \frac{\eta^2}{2} \sum_{j=0}^{2n-1} D\boldsymbol{F}_j(\boldsymbol{z}_0) \sum_{k=0}^{j-1} (\boldsymbol{F}_k \boldsymbol{w}_k - \boldsymbol{F}_k \boldsymbol{z}_0) \right\|$$

$$\leq \frac{\eta^2}{2} \sum_{j=0}^{2n-1} \left\| D\boldsymbol{F}_j(\boldsymbol{z}_0) \sum_{k=0}^{j-1} (\boldsymbol{F}_k \boldsymbol{w}_k - \boldsymbol{F}_k \boldsymbol{z}_0) \right\|$$

$$\leq \frac{\eta^3 L^2}{4} \sum_{j=1}^{2n-1} \left( j + \nu^2 j(j-1) + \frac{\nu^3 e^2 j(j-1)(j-2)}{3n} \right) \|\boldsymbol{F}\boldsymbol{z}_0\|$$

$$+ \frac{\eta^2}{2} \sum_{j=1}^{2n-1} \left( \frac{\eta L^2}{2} \Sigma_j + \frac{\eta L^2 (2\nu^2 - 1)}{2} \Sigma_{j-1} + \eta^2 L^3 \nu^3 e^2 \sum_{k=1}^{j-1} \Sigma_{k-1} \right)$$

$$= \frac{\eta^3 L^2}{4} \left( n(2n-1) + \frac{4\nu^2 n(n-1)(2n-1)}{3} + \frac{\nu^3 e^2 (n-1)(2n-1)(2n-3)}{3} \right) \|\boldsymbol{F}\boldsymbol{z}_0\|$$

$$+ \frac{\eta^3 L^2}{4} \sum_{j=1}^{2n-1} \Sigma_j + \frac{\eta^3 L^2 (2\nu^2 - 1)}{4} \sum_{j=1}^{2n-1} \Sigma_{j-1} + \frac{\eta^4 L^3 \nu^3 e^2}{2} \sum_{j=1}^{2n-1} \sum_{k=1}^{j-1} \Sigma_{k-1}$$

$$\leq \frac{\eta^3 L^2}{2} \left( n^2 + \frac{4n^3}{3} + \frac{2e^2 n^3}{3} \right) \|\boldsymbol{F}\boldsymbol{z}_0\|$$

$$+ \frac{\eta^3 L^2}{4} \sum_{j=1}^{2n-1} \Sigma_j + \frac{\eta^3 L^2 (2\nu^2 - 1)}{4} \sum_{j=1}^{2n-2} \Sigma_j + \frac{\eta^4 L^3 \nu^3 e^2}{2} \sum_{k=1}^{2n-2} \sum_{j=k+1}^{2n-1} \Sigma_{k-1}$$

$$\leq \eta^3 n^3 L^2 \left( \frac{4\nu + e^2}{3} \right) \|\boldsymbol{F}\boldsymbol{z}_0\|$$

$$+ \frac{\eta^3 L^2}{4} \Sigma_{2n-1} + \frac{\eta^3 L^2 \nu^2}{2} \sum_{j=1}^{2n-2} \Sigma_j + \frac{\eta^4 L^3 \nu^3 e^2}{2} \sum_{k=1}^{2n-2} (2n-k-1)\Sigma_{k-1}.$$

Simply collecting all the inequalities and rearranging the terms leads to the claimed bound. $\square$

Before we proceed, let us write

$$X_1 := \frac{3\eta^3 M}{8} \left( \Psi_{2n} + (2\nu^2 - 1)^2 \Psi_{2n-1} + \frac{4\nu^6 e^4}{n^2} \sum_{j=1}^{2n-2} j \Psi_j \right), \tag{64}$$

$$X_2 := \frac{\eta^3 L^2 (\nu + 1)}{4} \Sigma_{2n-1} + \frac{\eta^3 L^2 \nu^2 (1 + \eta L e^2)}{2} \sum_{j=1}^{2n-2} \Sigma_j + \frac{\eta^4 L^3 \nu^3 e^2}{2} \sum_{k=1}^{2n-2} (2n-k-1)\Sigma_{k-1} \tag{65}$$

so that the bound on $\|\boldsymbol{r}\|$ obtained in Proposition E.11 can be written as

$$\|\boldsymbol{r}\| \leq \eta^3 n^3 L^2 \|\boldsymbol{F}\boldsymbol{z}_0\| \left( \frac{1}{2n} \left( 1 + \frac{2e^2}{3} \right) + \frac{4\nu + e^2}{3} \right)$$

$$+ \eta^3 n^3 M \|\boldsymbol{F}\boldsymbol{z}_0\|^2 \left( \frac{1}{2} + 4\nu^4 + \frac{16\nu e^4}{5} \right) \tag{66}$$

$$+ X_1 + X_2.$$

**Theorem E.12.** *Suppose that $\eta < \frac{1}{nL}$, and let $\nu := 1 + \frac{1}{2n}$. Then the noise term* deterministically *satisfies the bound*

$$\|\boldsymbol{r}\| \le \eta^3 n^3 L^2 \|\boldsymbol{F}\boldsymbol{z}_0\| \left( \frac{1}{2n}\left(1 + \frac{2e^2}{3}\right) + \frac{4\nu + e^2}{3} + 10\nu\rho \right)$$

$$+ \eta^3 n^3 M \|\boldsymbol{F}\boldsymbol{z}_0\|^2 \left( \frac{1}{2} + 4\nu^4 + \frac{16\nu e^4}{5} + \rho^2\left(3\nu^4 + 8\nu^3 e^4\right) \right)$$

$$+ \eta^3 n^3 M \sigma^2 \left(3\nu^4 + 8\nu^3 e^4\right) + 10\nu\eta^3 n^3 L^2 \sigma.$$

*Proof.* From (36), (37), and Lemma E.5, it holds that

$$\Sigma_j = \sum_{i=1}^{j} \delta_i \le jn(\rho\|\boldsymbol{F}\boldsymbol{z}_0\| + \sigma), \tag{67}$$

$$\Psi_j = \sum_{i=1}^{j} \delta_i^2 \le jn^2(\rho\|\boldsymbol{F}\boldsymbol{z}_0\| + \sigma)^2. \tag{68}$$

Plugging the bound for $\Psi_j$ into (64) we get

$$X_1 \le \frac{3\eta^3 M}{8}\left( 2n^3(\rho\|\boldsymbol{F}\boldsymbol{z}_0\| + \sigma)^2 + (2\nu^2 - 1)^2(2n-1)n^2(\rho\|\boldsymbol{F}\boldsymbol{z}_0\| + \sigma)^2 + 4\nu^6 e^4 \sum_{j=1}^{2n-2} j^2(\rho\|\boldsymbol{F}\boldsymbol{z}_0\| + \sigma)^2 \right)$$

$$= \frac{3\eta^3 M}{8}\left( \left(2n^3 + (2\nu^2-1)^2(2n-1)n^2\right) + \frac{4\nu^6 e^4}{3}(n-1)(2n-1)(4n-3) \right)(\rho\|\boldsymbol{F}\boldsymbol{z}_0\| + \sigma)^2$$

$$\le \frac{3\eta^3 M}{8}\left( 4\nu^4 n^3 + \frac{32\nu^3 e^4 n^3}{3} \right)(\rho\|\boldsymbol{F}\boldsymbol{z}_0\| + \sigma)^2$$

$$= \frac{\eta^3 n^3 M(\rho\|\boldsymbol{F}\boldsymbol{z}_0\| + \sigma)^2}{2}\left(3\nu^4 + 8\nu^3 e^4\right).$$

By Young's inequality, it holds that

$$\frac{(\rho\|\boldsymbol{F}\boldsymbol{z}_0\| + \sigma)^2}{2} \le \rho^2\|\boldsymbol{F}\boldsymbol{z}_0\|^2 + \sigma^2,$$

from which we get

$$X_1 \le \eta^3 n^3 M \rho^2 \|\boldsymbol{F}\boldsymbol{z}_0\|^2 \left(3\nu^4 + 8\nu^3 e^4\right) + \eta^3 n^3 M \sigma^2 \left(3\nu^4 + 8\nu^3 e^4\right). \tag{69}$$

Meanwhile, plugging the bound for $\Sigma_j$ into (65) we get

$$X_2 \le \frac{\eta^3 L^2(\nu+1)}{4}(2n-1)n(\rho\|\boldsymbol{F}\boldsymbol{z}_0\| + \sigma) + \frac{\eta^3 L^2 \nu^2(1 + \eta L e^2)}{2}\sum_{j=1}^{2n-2} jn(\rho\|\boldsymbol{F}\boldsymbol{z}_0\| + \sigma)$$

$$+ \frac{\eta^4 L^3 \nu^3 e^2}{2}\sum_{k=1}^{2n-2}(2n-k-1)(k-1)n(\rho\|\boldsymbol{F}\boldsymbol{z}_0\| + \sigma)$$

$$= \frac{\eta^3 L^2(\nu+1)}{4}(2n-1)n(\rho\|\boldsymbol{F}\boldsymbol{z}_0\| + \sigma) + \frac{\eta^3 L^2 \nu^2(1 + \eta L e^2)}{2}(n-1)(2n-1)n(\rho\|\boldsymbol{F}\boldsymbol{z}_0\| + \sigma)$$

$$+ \frac{\eta^4 L^3 \nu^3 e^2}{6}\left(-3 + 11n - 12n^2 + 4n^3\right)n(\rho\|\boldsymbol{F}\boldsymbol{z}_0\| + \sigma)$$

$$\le \eta^3 L^2(\rho\|\boldsymbol{F}\boldsymbol{z}_0\| + \sigma)\left( n^2 + (1 + \eta L e^2)n^3 + \frac{2\eta L e^2}{3}n^4 \right)$$

$$\le \eta^3 n^3 L^2(\rho\|\boldsymbol{F}\boldsymbol{z}_0\| + \sigma)\left( \frac{1}{n} + 1 + \frac{e^2}{n} + \frac{2e^2}{3} \right)$$

where in the last line we used that $\eta < \frac{1}{nL}$. Because the inequality

$$\frac{1}{n} + 1 + \frac{e^2}{n} + \frac{2e^2}{3} \le 10\nu$$

holds for all $n \geq 1$, continuing from above we obtain

$$
\begin{aligned}
X_2 &\leq 10\nu\eta^3 n^3 L^2 (\rho \|\boldsymbol{F}\boldsymbol{z}_0\| + \sigma) \\
&\leq 10\nu\eta^3 n^3 L^2 \rho \|\boldsymbol{F}\boldsymbol{z}_0\| + 10\nu\eta^3 n^3 L^2 \sigma.
\end{aligned}
\tag{70}
$$

Rearranging (66) with applying the bounds (69) and (70) gives us the claimed result. $\qquad\square$

*Proof of Theorem E.9.* As $n \geq 1$, we notice that $1/n \leq 1$ and $\nu \leq 3/2$ where $\nu = 1 + \frac{1}{2n}$ following the notation of Theorem E.12. Then the bound (58) is immediate from Theorem E.12. $\qquad\square$

### E.2.2 Proof of Equation (32) for **SEG-FFA**

In this section, we prove the following.

**Theorem E.13.** *Say we use* **SEG-FFA**. *Then, as long as the stepsize used in an epoch satisfies* $\eta < \frac{1}{nL}$, *it holds that*

$$
\mathbb{E}\left[\|\boldsymbol{r}\|^2 \,\middle|\, \boldsymbol{z}_0\right] \leq \eta^6 n^6 C_{2A} \|\boldsymbol{F}\boldsymbol{z}_0\|^2 + \eta^6 n^6 D_{2A} \|\boldsymbol{F}\boldsymbol{z}_0\|^4 + \eta^6 n^5 V_{2A}
\tag{71}
$$

*for constants*

$$
C_{2A} := 4L^4 \left( \left( \frac{1}{2}\left(1 + \frac{2e^2}{3}\right) + \frac{6 + e^2}{3} \right)^2 + 36\rho^2 e^4 \right),
\tag{72}
$$

$$
D_{2A} := 4M^2 \left( \left( \frac{83}{4} + \frac{24e^4}{5} \right)^2 + \rho^4 \left( \frac{243}{16} + 27e^4 \right)^2 \right),
\tag{73}
$$

$$
V_{2A} := 4M^2\sigma^4 \left( \frac{243}{16} + 27e^4 \right)^2 + 144e^4 L^4 \sigma^2.
\tag{74}
$$

*Proof.* The bound is then immediate from the following Theorem E.14, as $n \geq 1$ implies $1/n \leq 1$ and $\nu \leq 3/2$ for $\nu$ defined in the statement of Theorem E.14. $\qquad\square$

**Theorem E.14.** *Suppose that* $\eta < \frac{1}{nL}$, *and let* $\nu := 1 + \frac{1}{2n}$. *Then,* in expectation, *the noise term satisfies the bound*

$$
\begin{aligned}
\mathbb{E}\left[\|\boldsymbol{r}\|^2 \,\middle|\, \boldsymbol{z}_0\right] &\leq 4\eta^6 n^6 L^4 \|\boldsymbol{F}\boldsymbol{z}_0\|^2 \left( \left( \frac{1}{2n}\left(1 + \frac{2e^2}{3}\right) + \frac{4\nu + e^2}{3} \right)^2 + \frac{36\rho^2 e^4}{n} \right) \\
&\quad + 4\eta^6 n^6 M^2 \|\boldsymbol{F}\boldsymbol{z}_0\|^4 \left( \left( \frac{1}{2} + 4\nu^4 + \frac{16\nu e^4}{5} \right)^2 + \frac{\rho^4 \left(3\nu^4 + 8\nu^3 e^4\right)^2}{n} \right) \\
&\quad + 4\eta^6 n^5 M^2 \sigma^4 \left(3\nu^4 + 8\nu^3 e^4\right)^2 + 144e^4 \eta^6 n^5 L^4 \sigma^2.
\end{aligned}
$$

*Proof.* Notice that, when conditioned on $\boldsymbol{z}_0$, the only source of randomness included in $\Psi_j$ is the random permutation $\tau$ selected for the epoch. Hence, we can use Lemma E.6 to get

$$
\mathbb{E}\left[\Psi_j \mid \boldsymbol{z}_0\right] = \mathbb{E}\left[\sum_{i=1}^j \delta_i^2 \,\middle|\, \boldsymbol{z}_0\right] = \sum_{i=1}^j \mathbb{E}\left[\delta_i^2 \mid \boldsymbol{z}_0\right] \leq \frac{jn(\rho \|\boldsymbol{F}\boldsymbol{z}_0\| + \sigma)^2}{2}.
$$

Applying Young's inequality on (66) we get

$$
\begin{aligned}
\|\boldsymbol{r}\|^2 &\leq 4\eta^6 n^6 L^4 \|\boldsymbol{F}\boldsymbol{z}_0\|^2 \left( \frac{1}{2n}\left(1 + \frac{2e^2}{3}\right) + \frac{4\nu + e^2}{3} \right)^2 \\
&\quad + 4\eta^6 n^6 M^2 \|\boldsymbol{F}\boldsymbol{z}_0\|^4 \left( \frac{1}{2} + 4\nu^4 + \frac{16\nu e^4}{5} \right)^2 \\
&\quad + 4X_1^2 + 4X_2^2.
\end{aligned}
\tag{75}
$$

When conditioned on $\boldsymbol{z}_0$, the first two lines are not random quantities. Thus, it suffices to derive the bounds for $\mathbb{E}\left[X_i^2 \mid \boldsymbol{z}_0\right]$, $i = 1, 2$.

Recall that the bound (69) on $X_1$ holds *deterministically*. Hence, it holds that

$$\mathbb{E}\left[X_1^2 \mid z_0\right] \leq \mathbb{E}\left[X_1\left(\eta^3 n^3 M \rho^2 \left\|\boldsymbol{F}\boldsymbol{z}_0\right\|^2 \left(3\nu^4 + 8\nu^3 e^4\right) + \eta^3 n^3 M \sigma^2 \left(3\nu^4 + 8\nu^3 e^4\right)\right) \,\middle|\, z_0\right]$$

$$= \eta^3 n^3 M \left(3\nu^4 + 8\nu^3 e^4\right)\left(\rho^2 \left\|\boldsymbol{F}\boldsymbol{z}_0\right\|^2 + \sigma^2\right)\mathbb{E}\left[X_1 \mid z_0\right].$$

Now, to compute $\mathbb{E}\left[X_1 \mid z_0\right]$, we apply the linearity of expectation on (64) to get

$$\mathbb{E}\left[X_1 \mid z_0\right]$$

$$= \frac{3\eta^3 M}{8}\left(\mathbb{E}\left[\Psi_{2n} \mid z_0\right] + (2\nu^2 - 1)^2 \mathbb{E}\left[\Psi_{2n-1} \mid z_0\right] + \frac{4\nu^6 e^4}{n^2}\sum_{j=1}^{2n-2} j\,\mathbb{E}\left[\Psi_j \mid z_0\right]\right)$$

$$\leq \frac{3\eta^3 M}{8}\left(n^2(\rho\left\|\boldsymbol{F}\boldsymbol{z}_0\right\| + \sigma)^2 + \frac{(2\nu^2-1)^2(2n-1)n(\rho\left\|\boldsymbol{F}\boldsymbol{z}_0\right\| + \sigma)^2}{2} + \frac{4\nu^6 e^4}{n^2}\sum_{j=1}^{2n-2}\frac{j^2 n(\rho\left\|\boldsymbol{F}\boldsymbol{z}_0\right\| + \sigma)^2}{2}\right)$$

$$= \frac{3\eta^3 M}{8}\left(\frac{2n^2 + (2\nu^2 - 1)^2(2n^2 - n)}{2} + \frac{2\nu^6 e^4(n-1)(2n-1)(4n-3)}{3n}\right)(\rho\left\|\boldsymbol{F}\boldsymbol{z}_0\right\| + \sigma)^2$$

$$\leq \frac{3\eta^3 M}{8}\left(2\nu^4 n^2 + \frac{16\nu^3 e^4 n^2}{3}\right)(\rho\left\|\boldsymbol{F}\boldsymbol{z}_0\right\| + \sigma)^2$$

$$= \frac{\eta^3 n^2 M(\rho\left\|\boldsymbol{F}\boldsymbol{z}_0\right\| + \sigma)^2}{4}\left(3\nu^4 + 8\nu^3 e^4\right).$$

Young's inequality gives us the bound

$$\frac{(\rho\left\|\boldsymbol{F}\boldsymbol{z}_0\right\| + \sigma)^2}{2} \leq \rho^2\left\|\boldsymbol{F}\boldsymbol{z}_0\right\|^2 + \sigma^2 \tag{76}$$

which, with the inequality derived above, leads to

$$\mathbb{E}\left[X_1 \mid z_0\right] \leq \frac{\eta^3 n^2 M}{2}\left(3\nu^4 + 8\nu^3 e^4\right)\left(\rho^2\left\|\boldsymbol{F}\boldsymbol{z}_0\right\|^2 + \sigma^2\right).$$

As a consequence, with using Young's inequality once again, we obtain

$$\mathbb{E}\left[X_1^2 \mid z_0\right] \leq \frac{\eta^6 n^5 M^2}{2}\left(3\nu^4 + 8\nu^3 e^4\right)^2\left(\rho^2\left\|\boldsymbol{F}\boldsymbol{z}_0\right\|^2 + \sigma^2\right)^2$$

$$\leq \eta^6 n^5 M^2\left(3\nu^4 + 8\nu^3 e^4\right)^2\left(\rho^4\left\|\boldsymbol{F}\boldsymbol{z}_0\right\|^4 + \sigma^4\right). \tag{77}$$

To get the bound of $\mathbb{E}\left[X_2^2 \mid z_0\right]$, we begin by using

$$\eta L\nu^2(2n - k - 1) \leq \left(1 + \frac{1}{2n}\right)^2\frac{2n - k - 1}{n}$$

$$= -\frac{k}{4n^3} - \frac{k}{n^2} - \frac{k}{n} - \frac{1}{4n^3} - \frac{1}{2n^2} + \frac{1}{n} + 2 \ \leq\ 2,$$

which holds for all $1 \leq k \leq 2n - 2$, to (65) to obtain

$$X_2 \leq \frac{\eta^3 L^2(\nu + 1)}{4}\Sigma_{2n-1} + \frac{\eta^3 L^2\nu^2(1 + \eta Le^2)}{2}\sum_{j=1}^{2n-2}\Sigma_j + \frac{\eta^3 L^2\nu^3 e^2}{2}\sum_{k=1}^{2n-2}\frac{2n - k - 1}{n}\Sigma_{k-1}$$

$$\leq \frac{\eta^3 L^2(\nu + 1)}{4}\Sigma_{2n-1} + \frac{\eta^3 L^2\nu^2(1 + \eta Le^2)}{2}\sum_{j=1}^{2n-2}\Sigma_j + \eta^3 L^2\nu e^2\sum_{k=1}^{2n-2}\Sigma_{k-1}$$

$$\leq \frac{\eta^3 L^2(\nu + 1)}{4}\Sigma_{2n-1} + \left(\frac{\eta^3 L^2\nu^2(1 + \eta Le^2)}{2} + \eta^3 L^2\nu e^2\right)\sum_{j=1}^{2n-2}\Sigma_j$$

$$\leq \frac{\eta^3 L^2(\nu + 1)}{4}\Sigma_{2n-1} + 3\eta^3 L^2 e^2\sum_{j=1}^{2n-2}\Sigma_j.$$

Then we directly square both sides and expand them to get

$$X_2^2 \leq \left( \frac{\eta^3 L^2 (\nu+1)}{4} \Sigma_{2n-1} + 3\eta^3 L^2 e^2 \sum_{j=1}^{2n-2} \Sigma_j \right)^2$$

$$= \frac{\eta^6 L^4 (\nu+1)^2}{16} \Sigma_{2n-1}^2 + 9\eta^6 L^4 e^4 \left( \sum_{j=1}^{2n-2} \Sigma_j \right)^2 + \frac{3\eta^6 L^4 e^2 (\nu+1)}{2} \sum_{j=1}^{2n-2} \Sigma_{2n-1} \Sigma_j.$$

Here, using Lemma E.7 and Lemma E.8 on the right hand side leads to

$$\mathbb{E}\left[ X_2^2 \,\middle|\, z_0 \right] \leq \frac{\eta^6 L^4 (\nu+1)^2 n (2n-1)^2 (\rho \|Fz_0\| + \sigma)^2}{32} + \frac{9\eta^6 L^4 e^4 n (2n-2)^2 (2n-1)^2 (\rho \|Fz_0\| + \sigma)^2}{8}$$

$$+ \frac{3\eta^6 L^4 e^2 (\nu+1)}{2} \sum_{j=1}^{2n-2} \frac{jn(2n-1)(\rho \|Fz_0\| + \sigma)^2}{2}$$

$$\leq \frac{\eta^6 L^4 (\nu+1)^2 n (2n-1)^2 (\rho \|Fz_0\| + \sigma)^2}{32} + \frac{9\eta^6 L^4 e^4 n (2n-2)^2 (2n-1)^2 (\rho \|Fz_0\| + \sigma)^2}{8}$$

$$+ \frac{3\eta^6 L^4 e^2 (\nu+1) n (n-1)(2n-1)^2 (\rho \|Fz_0\| + \sigma)^2}{4}$$

$$\leq \frac{\eta^6 L^4 n^3 (\rho \|Fz_0\| + \sigma)^2}{2} + \frac{9\eta^6 L^4 e^4 n (2n-2)^2 (2n-1)^2 (\rho \|Fz_0\| + \sigma)^2}{8}$$

$$+ 6\eta^6 L^4 e^2 n^3 (n-1)(\rho \|Fz_0\| + \sigma)^2$$

$$= \eta^6 L^4 \left( \frac{n^3}{2} + \frac{9 e^4 n (2n-2)^2 (2n-1)^2}{8} + 6 e^2 n^3 (n-1) \right) (\rho \|Fz_0\| + \sigma)^2$$

$$\leq 18 e^4 \eta^6 L^4 n^5 (\rho \|Fz_0\| + \sigma)^2.$$

As a consequence, with using (76) once again, we obtain

$$\mathbb{E}\left[ X_2^2 \,\middle|\, z_0 \right] \leq 36 e^4 \eta^6 L^4 n^5 \left( \rho^2 \|Fz_0\|^2 + \sigma^2 \right). \tag{78}$$

Taking the conditional expectation on (75), applying the bounds (77) and (78), and then rearranging the terms leads to the claimed inequality. □

### E.2.3 Upper Bounds of the Within-Epoch Errors for **SEG-FF**

**Theorem E.15.** *Say we use **SEG-FF** with $\alpha = \beta = \eta/2$. Then, as long as the stepsize used in an epoch satisfies $\eta < \frac{1}{nL}$, it holds that*

$$\|r\| \leq \eta^2 n^2 C_{1F} \|Fz_0\| + \eta^2 n^2 D_{1F} \|Fz_0\|^2 + \eta^2 n^2 V_{1F}$$

$$\mathbb{E}\left[ \|r\|^2 \,\middle|\, z_0 \right] \leq \eta^4 n^4 C_{2F} \|Fz_0\|^2 + \eta^4 n^4 D_{2F} \|Fz_0\|^4 + \eta^4 n^3 V_{2F}$$

*for constants $C_{1F}$, $D_{1F}$, $V_{1F}$, $C_{2F}$, $D_{2F}$, and $V_{2F}$ to be determined later in (81) and (82).*

*Proof.* As we have discussed in Section 5.1, we already know that aiming to achieve $\mathcal{O}(\eta^3)$ error without anchoring is futile. Instead, we show that error of magnitude $\mathcal{O}(\eta^2)$ is possible with the chosen stepsizes.

By Proposition D.2 and Lemma D.4 we have For any $i = 0, 1, \ldots, N$, it holds that

$$z_{2n} = z_0 - \frac{\eta}{2} \sum_{j=0}^{2n-1} T_j z_0 + \frac{\eta^2}{4} \sum_{j=0}^{2n-1} DT_j(z_0) T_j z_0 + \frac{\eta^2}{4} \sum_{0 \leq k < j \leq 2n-1} DT_j(z_0) T_k z_0 + \epsilon_{2n}$$

$$= z_0 - \eta \sum_{j=0}^{n-1} F_j z_0 + \frac{3\eta^2}{4} \sum_{j=1}^{n} DF_j(z_0) F_j z_0 + \frac{\eta^2}{2} \sum_{i \neq j} DF_j(z_0) F_i z_0 + \epsilon_{2n}$$

$$= z_0 - \eta n F z_0 + \eta^2 n^2 DF(z_0) F z_0 - \frac{\eta^2}{4} \sum_{j=1}^{n} DF_j(z_0) F_j z_0 - \frac{\eta^2}{2} \sum_{i \neq j} DF_j(z_0) F_i z_0 + \epsilon_{2n}$$

where we denote

$$\epsilon_{2n} := -\frac{\eta}{2} \sum_{j=0}^{2n-1} \Big( \boldsymbol{F}_j \boldsymbol{w}_j - \boldsymbol{F}_j \boldsymbol{z}_0 - D\boldsymbol{F}_j(\boldsymbol{z}_0)(\boldsymbol{w}_j - \boldsymbol{z}_0) \Big)$$

$$+ \frac{\eta^2}{4} \sum_{j=0}^{2n-1} D\boldsymbol{F}_j(\boldsymbol{z}_0)(\boldsymbol{F}_j \boldsymbol{z}_j - \boldsymbol{F}_j \boldsymbol{z}_0) + \frac{\eta^2}{4} \sum_{j=0}^{2n-1} D\boldsymbol{F}_j(\boldsymbol{z}_0) \sum_{k=0}^{j-1} (\boldsymbol{F}_k \boldsymbol{w}_k - \boldsymbol{F}_k \boldsymbol{z}_0). \tag{79}$$

Comparing $\boldsymbol{z}_{2n}$ to a point that would have been the result of a deterministic EG update with stepsize $\eta n$ we get

$$\boldsymbol{z}_{2n} - (\boldsymbol{z}_0 - \eta n \boldsymbol{F}(\boldsymbol{z}_0 - \eta n \boldsymbol{F} \boldsymbol{z}_0)) = \eta n \boldsymbol{F}(\boldsymbol{z}_0 - \eta n \boldsymbol{F} \boldsymbol{z}_0) - \eta n \boldsymbol{F} \boldsymbol{z}_0 + \eta^2 n^2 D\boldsymbol{F}(\boldsymbol{z}_0) \boldsymbol{F} \boldsymbol{z}_0 + \epsilon_{2n}$$

$$- \frac{\eta^2}{4} \sum_{j=1}^{n} D\boldsymbol{F}_j(\boldsymbol{z}_0) \boldsymbol{F}_j \boldsymbol{z}_0 - \frac{\eta^2}{2} \sum_{i \neq j} D\boldsymbol{F}_j(\boldsymbol{z}_0) \boldsymbol{F}_i \boldsymbol{z}_0.$$

Let us define

$$\tilde{\boldsymbol{r}} := \eta n \boldsymbol{F}(\boldsymbol{z}_0 - \eta n \boldsymbol{F} \boldsymbol{z}_0) - \eta n \boldsymbol{F} \boldsymbol{z}_0 + \eta^2 n^2 D\boldsymbol{F}(\boldsymbol{z}_0) \boldsymbol{F} \boldsymbol{z}_0 + \epsilon_{2n}. \tag{80}$$

Noticing the resemblence between (62) and the equations in (79) and (80), we can repeat the same reasoning used for Theorem E.9 and Theorem E.13, but with replacing the bounds given by Proposition E.3 to those in Proposition E.4 (and plugging in $\eta/2$ in place of $\eta$ in the statement of Proposition E.4) to conclude that

$$\|\tilde{\boldsymbol{r}}\| \leq \eta^3 n^3 \tilde{C}_{1\mathsf{A}} \|\boldsymbol{F} \boldsymbol{z}_0\| + \eta^3 n^3 \tilde{D}_{1\mathsf{A}} \|\boldsymbol{F} \boldsymbol{z}_0\|^2 + \eta^3 n^3 \tilde{V}_{1\mathsf{A}}$$

$$\mathbb{E}\left[ \|\tilde{\boldsymbol{r}}\|^2 \,\Big|\, \boldsymbol{z}_0 \right] \leq \eta^6 n^6 \tilde{C}_{2\mathsf{A}} \|\boldsymbol{F} \boldsymbol{z}_0\|^2 + \eta^6 n^6 \tilde{D}_{2\mathsf{A}} \|\boldsymbol{F} \boldsymbol{z}_0\|^4 + \eta^6 n^5 \tilde{V}_{2\mathsf{A}}$$

for some constants $\tilde{C}_{1\mathsf{A}}$, $\tilde{D}_{1\mathsf{A}}$, $\tilde{V}_{1\mathsf{A}}$, $\tilde{C}_{2\mathsf{A}}$, $\tilde{D}_{2\mathsf{A}}$, and $\tilde{V}_{2\mathsf{A}}$. Meanwhile, we also have

$$\left\| \frac{\eta^2}{4} \sum_{j=1}^{n} D\boldsymbol{F}_j(\boldsymbol{z}_0) \boldsymbol{F}_j \boldsymbol{z}_0 + \frac{\eta^2}{2} \sum_{i \neq j} D\boldsymbol{F}_j(\boldsymbol{z}_0) \boldsymbol{F}_i \boldsymbol{z}_0 \right\|$$

$$= \left\| \frac{\eta^2 n^2}{2} D\boldsymbol{F}(\boldsymbol{z}_0) \boldsymbol{F} \boldsymbol{z}_0 - \frac{\eta^2}{4} \sum_{j=1}^{n} D\boldsymbol{F}_j(\boldsymbol{z}_0) \boldsymbol{F}_j \boldsymbol{z}_0 \right\|$$

$$\leq \frac{\eta^2 n^2}{2} \|D\boldsymbol{F}(\boldsymbol{z}_0)\| \|\boldsymbol{F} \boldsymbol{z}_0\| + \frac{\eta^2}{4} \sum_{j=1}^{n} \|D\boldsymbol{F}_j(\boldsymbol{z}_0)\| \|\boldsymbol{F}_j \boldsymbol{z}_0\|$$

$$\leq \frac{\eta^2 n^2}{2} L \|\boldsymbol{F} \boldsymbol{z}_0\| + \frac{\eta^2}{4} \sum_{j=1}^{n} L \left( \|\boldsymbol{F}_j \boldsymbol{z}_0 - \boldsymbol{F} \boldsymbol{z}_0\| + \|\boldsymbol{F} \boldsymbol{z}_0\| \right)$$

$$\leq \frac{\eta^2 (n^2 + n) L}{2} \|\boldsymbol{F} \boldsymbol{z}_0\| + \frac{\eta^2 L}{4} \sum_{j=1}^{n} \|\boldsymbol{F}_j \boldsymbol{z}_0 - \boldsymbol{F} \boldsymbol{z}_0\|$$

$$\leq \eta^2 n^2 L \|\boldsymbol{F} \boldsymbol{z}_0\| + \frac{\eta^2 L}{4} \left( \sum_{j=1}^{n} \|\boldsymbol{F}_j \boldsymbol{z}_0 - \boldsymbol{F} \boldsymbol{z}_0\|^2 \right)^{1/2} \left( \sum_{j=1}^{n} 1 \right)^{1/2}$$

$$= \eta^2 n^2 L \|\boldsymbol{F} \boldsymbol{z}_0\| + \frac{\eta^2 n L}{4} (\rho \|\boldsymbol{F} \boldsymbol{z}_0\| + \sigma)$$

where in the second to the last line we used the Cauchy-Schwarz inequality. Therefore, as $\eta \leq 1/nL$, we conclude that

$$\|\boldsymbol{z}_{2n} - (\boldsymbol{z}_0 - \eta n \boldsymbol{F}(\boldsymbol{z}_0 - \eta n \boldsymbol{F} \boldsymbol{z}_0))\| \leq \eta^2 n^2 C_{1\mathsf{F}} \|\boldsymbol{F} \boldsymbol{z}_0\| + \eta^2 n^2 D_{1\mathsf{F}} \|\boldsymbol{F} \boldsymbol{z}_0\|^2 + \eta^2 n^2 V_{1\mathsf{F}}$$

for constants

$$C_{1\mathsf{F}} = L + \frac{\rho L}{4} + \frac{\tilde{C}_{1\mathsf{A}}}{L}, \quad D_{1\mathsf{F}} = \frac{\tilde{D}_{1\mathsf{A}}}{L}, \quad V_{1\mathsf{F}} = \frac{\sigma L}{4} + \frac{\tilde{V}_{1\mathsf{A}}}{L}. \tag{81}$$

Moreover, using Young's inequality, we see that

$$\left\| \frac{\eta^2}{4} \sum_{j=1}^{n} D\boldsymbol{F}_j(\boldsymbol{z}_0)\boldsymbol{F}_j\boldsymbol{z}_0 + \frac{\eta^2}{2} \sum_{i \neq j} D\boldsymbol{F}_j(\boldsymbol{z}_0)\boldsymbol{F}_i\boldsymbol{z}_0 \right\|^2$$

$$\leq 3\eta^4 n^4 L^2 \left\| \boldsymbol{F}\boldsymbol{z}_0 \right\|^2 + \frac{3\eta^4 n^2 L^2}{16} \rho^2 \left\| \boldsymbol{F}\boldsymbol{z}_0 \right\|^2 + \frac{3\eta^4 n^2 L^2}{16} \sigma^2,$$

so we also conclude that

$$\mathbb{E}\left[ \left\| \boldsymbol{z}_{2n} - (\boldsymbol{z}_0 - \eta n \boldsymbol{F}(\boldsymbol{z}_0 - \eta n \boldsymbol{F}\boldsymbol{z}_0)) \right\|^2 \,\Big|\, \boldsymbol{z}_0 \right] \leq \eta^4 n^4 C_{\mathsf{2F}} \left\| \boldsymbol{F}\boldsymbol{z}_0 \right\|^2 + \eta^4 n^4 D_{\mathsf{2F}} \left\| \boldsymbol{F}\boldsymbol{z}_0 \right\|^4 + \eta^4 n^3 V_{\mathsf{2F}}$$

holds for constants

$$C_{\mathsf{2F}} = 6L^2 + \frac{3\rho^2 L^2}{8} + \frac{2\tilde{C}_{\mathsf{2A}}}{L^2}, \quad D_{\mathsf{2F}} = \frac{2\tilde{D}_{\mathsf{1A}}}{L^2}, \quad V_{\mathsf{2F}} = \frac{3\sigma^2 L^2}{8} + \frac{2\tilde{V}_{\mathsf{1A}}}{L^2}. \tag{82}$$

$$\square$$

### E.2.4 Upper Bounds of the Within-Epoch Errors for **SEG-RR**

**Theorem E.16.** *Say we use* **SEG-RR** *with* $\alpha = \beta = \eta$. *Then, as long as the stepsize used in an epoch satisfies* $\eta < \frac{1}{nL}$, *it holds that*

$$\| \boldsymbol{r} \| \leq \eta^2 n^2 C_{\mathsf{1R}} \| \boldsymbol{F}\boldsymbol{z}_0 \| + \eta^2 n^2 D_{\mathsf{1R}} \| \boldsymbol{F}\boldsymbol{z}_0 \|^2 + \eta^2 n^2 V_{\mathsf{1R}}$$

$$\mathbb{E}\left[ \| \boldsymbol{r} \|^2 \,\Big|\, \boldsymbol{z}_0 \right] \leq \eta^4 n^4 C_{\mathsf{2R}} \| \boldsymbol{F}\boldsymbol{z}_0 \|^2 + \eta^4 n^4 D_{\mathsf{2R}} \| \boldsymbol{F}\boldsymbol{z}_0 \|^4 + \eta^4 n^3 V_{\mathsf{2R}}$$

*for constants* $C_{\mathsf{1R}}$, $D_{\mathsf{1R}}$, $V_{\mathsf{1R}}$, $C_{\mathsf{2R}}$, $D_{\mathsf{2R}}$, *and* $V_{\mathsf{2R}}$ *to be determined later in* (86) *and* (87).

*Proof.* As we have discussed in Section 5.1, we already know that aiming to achieve $\mathcal{O}(\eta^3)$ error with only using random reshuffling is futile. Instead, we show that error of magnitude $\mathcal{O}(\eta^2)$ is possible with the chosen stepsizes.

By Proposition D.2 and Lemma D.4 we have For any $i = 0, 1, \ldots, N$, it holds that

$$\boldsymbol{z}_n = \boldsymbol{z}_0 - \eta \sum_{j=0}^{n-1} \boldsymbol{F}_j\boldsymbol{z}_0 + \eta^2 \sum_{j=0}^{n-1} D\boldsymbol{F}_j(\boldsymbol{z}_0)\boldsymbol{F}_j\boldsymbol{z}_0 + \eta^2 \sum_{0 \leq k < j \leq n-1} D\boldsymbol{F}_j(\boldsymbol{z}_0)\boldsymbol{F}_k\boldsymbol{z}_0 + \boldsymbol{\epsilon}_n$$

$$= \boldsymbol{z}_0 - \eta n \boldsymbol{F}\boldsymbol{z}_0 + \eta^2 n^2 D\boldsymbol{F}(\boldsymbol{z}_0)\boldsymbol{F}\boldsymbol{z}_0 - \eta^2 \sum_{0 \leq j < k \leq n-1} D\boldsymbol{F}_j(\boldsymbol{z}_0)\boldsymbol{F}_k\boldsymbol{z}_0 + \boldsymbol{\epsilon}_n$$

where we denote

$$\boldsymbol{\epsilon}_n := -\eta \sum_{j=0}^{n-1} \Big( \boldsymbol{F}_j\boldsymbol{w}_j - \boldsymbol{F}_j\boldsymbol{z}_0 - D\boldsymbol{F}_j(\boldsymbol{z}_0)(\boldsymbol{w}_j - \boldsymbol{z}_0) \Big)$$

$$+ \eta^2 \sum_{j=0}^{n-1} D\boldsymbol{F}_j(\boldsymbol{z}_0)(\boldsymbol{F}_j\boldsymbol{z}_j - \boldsymbol{F}_j\boldsymbol{z}_0) + \eta^2 \sum_{j=0}^{n-1} D\boldsymbol{F}_j(\boldsymbol{z}_0) \sum_{k=0}^{j-1} (\boldsymbol{F}_k\boldsymbol{w}_k - \boldsymbol{F}_k\boldsymbol{z}_0). \tag{83}$$

Comparing $\boldsymbol{z}_n$ to a point that would have been the result of a deterministic EG update with stepsize $\eta n$ we get

$$\boldsymbol{z}_n - (\boldsymbol{z}_0 - \eta n \boldsymbol{F}(\boldsymbol{z}_0 - \eta n \boldsymbol{F}\boldsymbol{z}_0)) = \eta n \boldsymbol{F}(\boldsymbol{z}_0 - \eta n \boldsymbol{F}\boldsymbol{z}_0) - \eta n \boldsymbol{F}\boldsymbol{z}_0 + \eta^2 n^2 D\boldsymbol{F}(\boldsymbol{z}_0)\boldsymbol{F}\boldsymbol{z}_0 + \boldsymbol{\epsilon}_n$$

$$- \eta^2 \sum_{0 \leq j < k \leq n-1} D\boldsymbol{F}_j(\boldsymbol{z}_0)\boldsymbol{F}_k\boldsymbol{z}_0.$$

Let us define

$$\check{\boldsymbol{r}} := \eta n \boldsymbol{F}(\boldsymbol{z}_0 - \eta n \boldsymbol{F}\boldsymbol{z}_0) - \eta n \boldsymbol{F}\boldsymbol{z}_0 + \eta^2 n^2 D\boldsymbol{F}(\boldsymbol{z}_0)\boldsymbol{F}\boldsymbol{z}_0 + \boldsymbol{\epsilon}_n. \tag{84}$$

Comparing the sums (62b)–(62d) to (83), we can repeat the same reasoning used for Theorem E.9 and Theorem E.13, but with replacing the bounds given by Proposition E.3 to those in Proposition E.4, to conclude that

$$\| \check{\boldsymbol{r}} \| \leq \eta^3 n^3 \check{C}_{\mathsf{1A}} \| \boldsymbol{F}\boldsymbol{z}_0 \| + \eta^3 n^3 \check{D}_{\mathsf{1A}} \| \boldsymbol{F}\boldsymbol{z}_0 \|^2 + \eta^3 n^3 \check{V}_{\mathsf{1A}}$$

$$\mathbb{E}\left[ \| \check{\boldsymbol{r}} \|^2 \,\Big|\, \boldsymbol{z}_0 \right] \leq \eta^6 n^6 \check{C}_{\mathsf{2A}} \| \boldsymbol{F}\boldsymbol{z}_0 \|^2 + \eta^6 n^6 \check{D}_{\mathsf{2A}} \| \boldsymbol{F}\boldsymbol{z}_0 \|^4 + \eta^6 n^5 \check{V}_{\mathsf{2A}}$$

for some constants $\check{C}_{1A}$, $\check{D}_{1A}$, $\check{V}_{1A}$, $\check{C}_{2A}$, $\check{D}_{2A}$, and $\check{V}_{2A}$. Meanwhile, we also have

$$
\sum_{0 \leq j < k \leq n-1} D\boldsymbol{F}_j(\boldsymbol{z}_0)\boldsymbol{F}_k\boldsymbol{z}_0 = \sum_{j=0}^{n-1} D\boldsymbol{F}_j(\boldsymbol{z}_0)(n\boldsymbol{F}\boldsymbol{z}_0 - \boldsymbol{g}_{j+1})
$$
$$
= \sum_{j=0}^{n-1}(n-j-1)D\boldsymbol{F}_j(\boldsymbol{z}_0)\boldsymbol{F}\boldsymbol{z}_0 - \sum_{j=0}^{n-1}D\boldsymbol{F}_j(\boldsymbol{z}_0)(\boldsymbol{g}_{j+1} - (j+1)\boldsymbol{F}\boldsymbol{z}_0)
$$

which leads to

$$
\left\| \sum_{0 \leq j < k \leq n-1} D\boldsymbol{F}_j(\boldsymbol{z}_0)\boldsymbol{F}_k\boldsymbol{z}_0 \right\| \leq \sum_{j=0}^{n-1}(n-j-1)L \left\| \boldsymbol{F}\boldsymbol{z}_0 \right\| + L\sum_{j=0}^{n-1}\delta_{j+1}
$$
$$
\leq \frac{n^2 L}{2} \left\| \boldsymbol{F}\boldsymbol{z}_0 \right\| + L\sum_{j=0}^{n-1}\delta_{j+1}. \tag{85}
$$

Therefore, from $\eta \leq 1/nL$ and Lemma E.5, on one hand we obtain

$$
\left\| \boldsymbol{z}_n - (\boldsymbol{z}_0 - \eta n\boldsymbol{F}(\boldsymbol{z}_0 - \eta n\boldsymbol{F}\boldsymbol{z}_0)) \right\| \leq \eta^2 n^2 C_{1R} \left\| \boldsymbol{F}\boldsymbol{z}_0 \right\| + \eta^2 n^2 D_{1R} \left\| \boldsymbol{F}\boldsymbol{z}_0 \right\|^2 + \eta^2 n^2 V_{1R}
$$

for constants

$$
C_{1R} = \frac{L}{2} + \rho L + \frac{\check{C}_{1A}}{L}, \quad D_{1R} = \frac{\check{D}_{1A}}{L}, \quad V_{1R} = \sigma L + \frac{\check{V}_{1A}}{L}. \tag{86}
$$

On the other hand, applying Young's inequality on (85) we get

$$
\left\| \sum_{0 \leq j < k \leq n-1} D\boldsymbol{F}_j(\boldsymbol{z}_0)\boldsymbol{F}_k\boldsymbol{z}_0 \right\|^2 \leq n^4 L^2 \left\| \boldsymbol{F}\boldsymbol{z}_0 \right\|^2 + 2L^2 \left( \sum_{j=0}^{n-1}\delta_{j+1} \right)^2
$$
$$
\leq n^4 L^2 \left\| \boldsymbol{F}\boldsymbol{z}_0 \right\|^2 + 2nL^2\sum_{j=1}^{n}\delta_j^2.
$$

Taking the expectation conditioned on $\boldsymbol{z}_0$ and applying Lemma E.6, we conclude that

$$
\mathbb{E}\left[ \left\| \boldsymbol{z}_{2n} - (\boldsymbol{z}_0 - \eta n\boldsymbol{F}(\boldsymbol{z}_0 - \eta n\boldsymbol{F}\boldsymbol{z}_0)) \right\|^2 \Big| \boldsymbol{z}_0 \right] \leq \eta^4 n^4 C_{2R} \left\| \boldsymbol{F}\boldsymbol{z}_0 \right\|^2 + \eta^4 n^4 D_{2R} \left\| \boldsymbol{F}\boldsymbol{z}_0 \right\|^4 + \eta^4 n^3 V_{2R}
$$

holds for constants

$$
C_{2R} = 2L^2 + 4\rho^2 L^2 + \frac{2\check{C}_{2A}}{L^2}, \quad D_{2R} = \frac{2\check{D}_{2A}}{L^2}, \quad V_{2R} = 4\sigma^2 L^2 + \frac{2\check{V}_{2A}}{L^2}. \tag{87}
$$

$\square$

# F Convergence Bounds in the Strongly Monotone Setting

In this section, we focus only on the iterates $\{\boldsymbol{z}_0^k\}_{k \geq 0}$. So, we omit the subscript 0 unless necessary, and simply write $\boldsymbol{z}^k$ instead of $\boldsymbol{z}_0^k$.

## F.1 Unified Analysis of the Upper Bounds for Shuffling-Based SEG Methods

When $\boldsymbol{F}$ is $\mu$-strongly monotone with $\mu > 0$, all of SEG-RR, SEG-FF, and SEG-FFA do not diverge. In fact, it is possible to establish the following unified analysis of the methods.

**Theorem F.1** (Theorem F.5, simplified). *Suppose that $\boldsymbol{F}$ is $\mu$-strongly monotone with $\mu > 0$, Assumption 3.3 holds, and an optimization method whose within-epoch error satisfies (31) and (32) for some constant $a > 0$ is run for $K$ epochs. Then, for a sufficiently small constant $\omega$ that does not depend on $K$, we achieve the bound*

$$
\mathbb{E} \left\| \boldsymbol{z}^K - \boldsymbol{z}^* \right\|^2 \leq \exp\left( -\frac{1}{2}\mu\omega nK \right) \left\| \boldsymbol{z}^0 - \boldsymbol{z}^* \right\|^2 + \tilde{\mathcal{O}}\left( \frac{1}{nK^{2a-2}} \right).
$$

The goal of this section is to prove this theorem, whose precise statement is in Theorem F.5. As the polynomial decay will dominate the exponential decay for large enough $K$, the bound we get is essentially $\tilde{\mathcal{O}}\left(1/nK^{2a-2}\right)$. Recall that for SEG-FF and SEG-RR we have $a = 2$ (by Theorems E.15 and E.16) which leads to an upper bound of $\tilde{\mathcal{O}}(1/nK^2)$, whereas for SEG-FFA we have $a = 3$ (by Theorems E.9 and E.13) which gives an upper bound of $\tilde{\mathcal{O}}(1/nK^4)$.

As also mentioned in the beginning of Appendix E, for any of SEG-RR, SEG-FF, and SEG-FFA, we can decompose the update across the epoch into a deterministic EG update plus a noise. In this section, letting $\boldsymbol{w}_\dagger^k := \boldsymbol{z}^k - \eta_k n \boldsymbol{F} \boldsymbol{z}^k$, we define $\widehat{\boldsymbol{F}}^k$ by the relation $\eta_k n \widehat{\boldsymbol{F}}^k = \eta_k n \boldsymbol{F} \boldsymbol{w}_\dagger^k + \boldsymbol{r}^k$ so that

$$\boldsymbol{z}^{k+1} = \boldsymbol{z}^k - \eta_k n \widehat{\boldsymbol{F}}^k. \tag{88}$$

**Proposition F.2.** *Let $\boldsymbol{F}$ be $\mu$-strongly monotone with $\mu > 0$. Then, for any $\eta_k > 0$, it holds that*

$$\eta_k^2 n^2 \left(1 - \frac{3}{2}\mu\eta_k n - \left(1 + \frac{1}{2}\mu\eta_k n\right)\eta_k^2 n^2 L^2\right) \left\|\boldsymbol{F}\boldsymbol{z}^k\right\|^2$$
$$\leq \left(1 - \frac{1}{2}\mu\eta_k n\right)\left\|\boldsymbol{z}^k - \boldsymbol{z}^*\right\|^2 - \left\|\boldsymbol{z}^{k+1} - \boldsymbol{z}^*\right\|^2 + \frac{2 + \mu\eta_k n}{\mu\eta_k n}\left\|\boldsymbol{r}^k\right\|^2. \tag{89}$$

*Proof.* From (88), using Lemma C.7 we get

$$\left\|\boldsymbol{z}^{k+1} - \boldsymbol{z}^*\right\|^2 = \left\|\boldsymbol{z}^k - \boldsymbol{z}^*\right\|^2 - 2\left\langle \eta_k n \widehat{\boldsymbol{F}}^k, \boldsymbol{z}^k - \boldsymbol{z}^* \right\rangle + \left\|\eta_k n \widehat{\boldsymbol{F}}^k\right\|^2$$
$$= \left\|\boldsymbol{z}^k - \boldsymbol{z}^*\right\|^2 - 2\eta_k n \left\langle \boldsymbol{F}\boldsymbol{w}_\dagger^k, \boldsymbol{w}_\dagger^k - \boldsymbol{z}^* \right\rangle - 2\eta_k^2 n^2 \left\langle \boldsymbol{F}\boldsymbol{w}_\dagger^k, \boldsymbol{F}\boldsymbol{z}^k \right\rangle$$
$$- 2\left\langle \boldsymbol{r}^k, \boldsymbol{z}^k - \boldsymbol{z}^* \right\rangle + \left\|\eta_k n \widehat{\boldsymbol{F}}^k\right\|^2$$
$$\leq \left\|\boldsymbol{z}^k - \boldsymbol{z}^*\right\|^2 - \mu\eta_k n \left\|\boldsymbol{z}^k - \boldsymbol{z}^*\right\|^2 - 2\eta_k^2 n^2 \left\langle \boldsymbol{F}\boldsymbol{w}_\dagger^k, \boldsymbol{F}\boldsymbol{z}^k \right\rangle$$
$$- 2\left\langle \boldsymbol{r}^k, \boldsymbol{z}^k - \boldsymbol{z}^* \right\rangle + \left\|\eta_k n \widehat{\boldsymbol{F}}^k\right\|^2 + 2\mu\eta_k^3 n^3 \left\|\boldsymbol{F}\boldsymbol{z}^k\right\|^2.$$

Meanwhile, using the polarization identity (Lemma C.1) and the $L$-smoothness of $\boldsymbol{F}$ we get

$$-2\left\langle \boldsymbol{F}\boldsymbol{w}_\dagger^k, \boldsymbol{F}\boldsymbol{z}^k \right\rangle = \left\|\boldsymbol{F}\boldsymbol{w}_\dagger^k - \boldsymbol{F}\boldsymbol{z}^k\right\|^2 - \left\|\boldsymbol{F}\boldsymbol{w}_\dagger^k\right\|^2 - \left\|\boldsymbol{F}\boldsymbol{z}^k\right\|^2$$
$$\leq L^2 \left\|\boldsymbol{w}_\dagger^k - \boldsymbol{z}^k\right\|^2 - \left\|\boldsymbol{F}\boldsymbol{w}_\dagger^k\right\|^2 - \left\|\boldsymbol{F}\boldsymbol{z}^k\right\|^2$$
$$\leq -(1 - \eta_k^2 n^2 L^2) \left\|\boldsymbol{F}\boldsymbol{z}^k\right\|^2 - \left\|\boldsymbol{F}\boldsymbol{w}_\dagger^k\right\|^2.$$

Combining the two inequalities and using the definition of $\widehat{\boldsymbol{F}}$ we obtain

$$\left\|\boldsymbol{z}^{k+1} - \boldsymbol{z}^*\right\|^2 \leq (1 - \mu\eta_k n)\left\|\boldsymbol{z}^k - \boldsymbol{z}^*\right\|^2 - \eta_k^2 n^2(1 - \eta_k^2 n^2 L^2)\left\|\boldsymbol{F}\boldsymbol{z}^k\right\|^2 - \eta_k^2 n^2 \left\|\boldsymbol{F}\boldsymbol{w}_\dagger^k\right\|^2$$
$$- 2\left\langle \boldsymbol{r}^k, \boldsymbol{z}^k - \boldsymbol{z}^* \right\rangle + \left\|\eta_k n \boldsymbol{F}\boldsymbol{w}_\dagger^k + \boldsymbol{r}^k\right\|^2 + 2\mu\eta_k^3 n^3 \left\|\boldsymbol{F}\boldsymbol{z}^k\right\|^2$$
$$\leq (1 - \mu\eta_k n)\left\|\boldsymbol{z}^k - \boldsymbol{z}^*\right\|^2 - \eta_k^2 n^2(1 - 2\mu\eta_k n - \eta_k^2 n^2 L^2)\left\|\boldsymbol{F}\boldsymbol{z}^k\right\|^2$$
$$- 2\left\langle \boldsymbol{r}^k, \boldsymbol{z}^k - \boldsymbol{z}^* \right\rangle + 2\left\langle \boldsymbol{r}^k, \eta_k n \boldsymbol{F}\boldsymbol{w}_\dagger^k \right\rangle + \left\|\boldsymbol{r}^k\right\|^2$$
$$\leq (1 - \mu\eta_k n)\left\|\boldsymbol{z}^k - \boldsymbol{z}^*\right\|^2 - \eta_k^2 n^2(1 - 2\mu\eta_k n - \eta_k^2 n^2 L^2)\left\|\boldsymbol{F}\boldsymbol{z}^k\right\|^2$$
$$- 2\left\langle \boldsymbol{r}^k, \boldsymbol{z}^k - \eta_k n \boldsymbol{F}\boldsymbol{w}_\dagger^k - \boldsymbol{z}^* \right\rangle + \left\|\boldsymbol{r}^k\right\|^2.$$

Let us consider the inner product term in the last line above. By Lemma C.2 and the nonexpansiveness of the EG update (Lemma C.10), for any $\gamma_k > 0$ we have

$$-2\left\langle \boldsymbol{r}^k, \boldsymbol{z}^k - \eta_k n \boldsymbol{F}\boldsymbol{w}_\dagger^k - \boldsymbol{z}^* \right\rangle \leq \frac{1}{\gamma_k}\left\|\boldsymbol{r}^k\right\|^2 + \gamma_k \left\|\boldsymbol{z}^k - \eta_k n \boldsymbol{F}\boldsymbol{w}_\dagger^k - \boldsymbol{z}^*\right\|^2$$
$$\leq \frac{1}{\gamma_k}\left\|\boldsymbol{r}^k\right\|^2 + \gamma_k \left\|\boldsymbol{z}^k - \boldsymbol{z}^*\right\|^2 - \gamma_k \eta_k^2 n^2(1 - \eta_k^2 n^2 L^2)\left\|\boldsymbol{F}\boldsymbol{z}^k\right\|^2.$$

Plugging this back we get

$$\eta_k^2 n^2 (1 + \gamma_k - 2\mu\eta_k n - (1 + \gamma_k)\eta_k^2 n^2 L^2) \left\| \boldsymbol{F} \boldsymbol{z}^k \right\|^2$$

$$\leq (1 + \gamma_k - \mu\eta_k n) \left\| \boldsymbol{z}^k - \boldsymbol{z}^* \right\|^2 - \left\| \boldsymbol{z}^{k+1} - \boldsymbol{z}^* \right\|^2 + \left( 1 + \frac{1}{\gamma_k} \right) \left\| \boldsymbol{r}^k \right\|^2. \tag{90}$$

Choosing $\gamma_k = \frac{\mu\eta_k n}{2}$ completes the proof. $\qquad\square$

**Proposition F.3.** *Let $\boldsymbol{F}$ be a $\mu$-strongly monotone and $L$-Lipschitz operator. Then, whenever $\eta_k \leq \frac{1}{nL\sqrt{2}}$, it holds that*

$$\left\| \boldsymbol{F} \boldsymbol{z}^{k+1} \right\| \leq \left( 1 - \frac{\mu n \eta_k}{5} \right) \left\| \boldsymbol{F} \boldsymbol{z}^k \right\| + L \left\| \boldsymbol{r}^k \right\|.$$

*Proof.* Let $\boldsymbol{z}_\dagger^{k+1} := \boldsymbol{z}^k - \eta_k n \boldsymbol{F}(\boldsymbol{z}^k - \eta_k n \boldsymbol{F} \boldsymbol{z}^k)$, so that we have $\left\| \boldsymbol{z}^{k+1} - \boldsymbol{z}_\dagger^{k+1} \right\| = \left\| \boldsymbol{r}^k \right\|$. Then, the $L$-smoothness of $\boldsymbol{F}$ and Lemma C.8 implies

$$\left\| \boldsymbol{F} \boldsymbol{z}^{k+1} \right\| \leq \left\| \boldsymbol{F} \boldsymbol{z}^{k+1} - \boldsymbol{F} \boldsymbol{z}_\dagger^{k+1} \right\| + \left\| \boldsymbol{F} \boldsymbol{z}_\dagger^{k+1} \right\|$$

$$\leq L \left\| \boldsymbol{z}^{k+1} - \boldsymbol{z}_\dagger^{k+1} \right\| + \left\| \boldsymbol{F} \boldsymbol{z}_\dagger^{k+1} \right\|$$

$$\leq L \left\| \boldsymbol{r}^k \right\| + \left( 1 - \frac{2\mu\eta_k n}{5} \right)^{1/2} \left\| \boldsymbol{F} \boldsymbol{z}^k \right\|$$

$$\leq L \left\| \boldsymbol{r}^k \right\| + \left( 1 - \frac{\mu\eta_k n}{5} \right) \left\| \boldsymbol{F} \boldsymbol{z}^k \right\|$$

where in the last line we apply a simple inequality $1 - 2x \leq (1 - x)^2$ which holds for all $x \in \mathbb{R}$. $\quad\square$

**Lemma F.4.** *Suppose that (31) holds. Say we use a constant stepsize $\eta_k = \eta$, where $\eta$ satisfies $\eta \leq \frac{1}{nL\sqrt{2}}$ and*

$$\eta^{a-1} n^{a-1} \leq \frac{1}{10} \min \left\{ \frac{1}{L^2}, \frac{\mu}{L(C_1 + D_1(\|\boldsymbol{F} \boldsymbol{z}_0\| + V_1/\mu L))} \right\}. \tag{91}$$

*Then for any $k = 0, 1, \ldots,$ the following two inequalities both hold:*

$$\left\| \boldsymbol{F} \boldsymbol{z}^{k+1} \right\| \leq \left( 1 - \frac{\mu\eta n}{10} \right) \left\| \boldsymbol{F} \boldsymbol{z}^k \right\| + \eta^a n^a L V_1, \tag{92}$$

$$\left\| \boldsymbol{F} \boldsymbol{z}^k \right\| \leq \left\| \boldsymbol{F} \boldsymbol{z}^0 \right\| + \frac{V_1}{\mu L}. \tag{93}$$

*Proof.* For the case $k = 0$, the inequality (93) clearly holds. For the remaining cases, we use strong induction on $k$. More precisely, assuming that (93) holds for all $0, 1, \ldots, k$, we will show that (92) holds, and from that the inequality

$$\left\| \boldsymbol{F} \boldsymbol{z}^{k+1} \right\| \leq \left\| \boldsymbol{F} \boldsymbol{z}^0 \right\| + \frac{V_1}{\mu L} \tag{94}$$

follows. To this end, let us begin from noting that Proposition F.3, (31), and the induction hypothesis (93) implies

$$\left\| \boldsymbol{F} \boldsymbol{z}^{k+1} \right\| \leq \left( 1 - \frac{\mu\eta n}{5} \right) \left\| \boldsymbol{F} \boldsymbol{z}^k \right\| + \eta^a n^a L \left( C_1 \left\| \boldsymbol{F} \boldsymbol{z}^k \right\| + D_1 \left\| \boldsymbol{F} \boldsymbol{z}^k \right\|^2 + V_1 \right)$$

$$\leq \left( 1 - \frac{\mu\eta n}{5} + \eta^a n^a L C_1 + \eta^a n^a L D_1 \left( \left\| \boldsymbol{F} \boldsymbol{z}^0 \right\| + \frac{V_1}{\mu L} \right) \right) \left\| \boldsymbol{F} \boldsymbol{z}^k \right\| + \eta^a n^a L V_1. \tag{95}$$

Here, from the choice of the stepsize (91), we have

$$\eta^a n^a L C_1 + \eta^a n^a L D_1 \left( \left\| \boldsymbol{F} \boldsymbol{z}^0 \right\| + \frac{V_1}{\mu L} \right) \leq \frac{\mu\eta n}{10}.$$

Hence, from (95) we get

$$\left\|\boldsymbol{F}\boldsymbol{z}^{k+1}\right\| \leq \left(1 - \frac{\mu\eta n}{10}\right)\left\|\boldsymbol{F}\boldsymbol{z}^{k}\right\| + \eta^a n^a LV_1.$$

which is exactly (92). Now, considering that we are assuming (93) holds for all $0, 1, \ldots, k$, we must also have (92) for all $0, 1, \ldots, k$. Thus we can unravel the recurrence to get

$$
\begin{aligned}
\left\|\boldsymbol{F}\boldsymbol{z}^{k+1}\right\| &\leq \left(1 - \frac{\mu\eta n}{10}\right)\left\|\boldsymbol{F}\boldsymbol{z}^{k}\right\| + \eta^a n^a LV_1 \\
&\leq \left(1 - \frac{\mu\eta n}{10}\right)^2\left\|\boldsymbol{F}\boldsymbol{z}^{k-1}\right\| + \left(1 - \frac{\mu\eta n}{10}\right)\eta^a n^a LV_1 + \eta^a n^a LV_1 \\
&\leq \ldots \\
&\leq \left(1 - \frac{\mu\eta n}{10}\right)^{k+1}\left\|\boldsymbol{F}\boldsymbol{z}^{0}\right\| + \eta^a n^a LV_1 \sum_{j=0}^{k}\left(1 - \frac{\mu\eta n}{10}\right)^j && (96) \\
&\leq \left\|\boldsymbol{F}\boldsymbol{z}^{0}\right\| + \frac{\eta^a n^a LV_1}{1 - \left(1 - \frac{\mu\eta n}{10}\right)} \\
&= \left\|\boldsymbol{F}\boldsymbol{z}^{0}\right\| + \frac{10\eta^{a-1}n^{a-1}LV_1}{\mu}.
\end{aligned}
$$

As (91) also implies $10\eta^{a-1}n^{a-1}L \leq 1/L$, we obtain (94), as claimed. This completes the proof. $\square$

**Theorem F.5** (Theorem F.1). *Suppose that $\boldsymbol{F}$ is $\mu$-strongly monotone with $\mu > 0$, Assumption 3.3 holds, and an optimization method whose within-epoch error satisfies (31) and (32) for some constant $a > 0$ is run for $K$ epochs. Let us define a constant*

$$\Phi := C_2 + D_2\left(\left\|\boldsymbol{F}\boldsymbol{z}^{0}\right\| + \frac{V_1}{\mu L}\right)^2.$$

*Say we use a constant stepsize $\eta_k = \eta$, where $\eta$ is chosen as*

$$\eta = \min\left\{\frac{2}{5nL},\right. \tag{97a}$$

$$\frac{1}{n(10L^2)^{1/(a-1)}}, \tag{97b}$$

$$\frac{\mu^{1/(a-1)}}{n(10L(C_1 + D_1(\|\boldsymbol{F}\boldsymbol{z}_0\| + V_1/\mu L)))^{1/(a-1)}}, \tag{97c}$$

$$\frac{1}{(12\Phi/\mu)^{1/(2a-3)}n}, \tag{97d}$$

$$\left.\frac{4(a-1)\log(n^{1/(2a-2)}K)}{\mu n K}\right\}. \tag{97e}$$

*Then for $\omega$ denoting the minimum among (97a)–(97d), it holds that*

$$\mathbb{E}\left\|\boldsymbol{z}^{K} - \boldsymbol{z}^{*}\right\|^2 \leq \exp\left(-\frac{1}{2}\mu\omega nK\right)\left\|\boldsymbol{z}^{0} - \boldsymbol{z}^{*}\right\|^2 + \mathcal{O}\left(\frac{(\log(n^{1/(2a-2)}K))^{2a-2}}{nK^{2a-2}}\right). \tag{98}$$

As a reminder, for SEG-FF and SEG-RR we have $a = 2$, and for SEG-FFA we have $a = 3$.

*Proof.* Notice that (97b) and (97c) together implies (91), and that $\eta_k = \eta \leq \frac{2}{5nL} \leq \frac{1}{nL\sqrt{2}} < \frac{1}{nL}$. So, we can utilize (32) and Lemma F.4 to get

$$
\begin{aligned}
\mathbb{E}\left[\left\|\boldsymbol{r}\right\|^2 \,\middle|\, \boldsymbol{z}^k\right] &\leq \eta^{2a}n^{2a}C_2\left\|\boldsymbol{F}\boldsymbol{z}^{k}\right\|^2 + \eta^{2a}n^{2a}D_2\left\|\boldsymbol{F}\boldsymbol{z}^{k}\right\|^4 + \eta^{2a}n^{2a-1}V_2 \\
&\leq \eta^{2a}n^{2a}C_2\left\|\boldsymbol{F}\boldsymbol{z}^{k}\right\|^2 + \eta^{2a}n^{2a}D_2\left(\left\|\boldsymbol{F}\boldsymbol{z}^{0}\right\| + \frac{V_1}{\mu L}\right)^2\left\|\boldsymbol{F}\boldsymbol{z}^{k}\right\|^2 + \eta^{2a}n^{2a-1}V_2 \\
&= \eta^{2a}n^{2a}\Phi\left\|\boldsymbol{F}\boldsymbol{z}^{k}\right\|^2 + \eta^{2a}n^{2a-1}V_2.
\end{aligned}
$$

Taking the conditional expectation on (89) and applying the bound just derived, we obtain

$$\eta^2 n^2 \left(1 - \frac{3}{2}\mu\eta n - \left(1 + \frac{1}{2}\mu\eta n\right)\eta^2 n^2 L^2\right)\left\|\boldsymbol{F}\boldsymbol{z}^k\right\|^2$$

$$\leq \left(1 - \frac{1}{2}\mu\eta n\right)\left\|\boldsymbol{z}^k - \boldsymbol{z}^*\right\|^2 - \mathbb{E}\left[\left\|\boldsymbol{z}^{k+1} - \boldsymbol{z}^*\right\|^2 \,\Big|\, \boldsymbol{z}^k\right]$$

$$+ \frac{2 + \mu\eta n}{\mu}\left(\eta^{2a-1}n^{2a-1}\Phi\left\|\boldsymbol{F}\boldsymbol{z}^k\right\|^2 + \eta^{2a-1}n^{2a-2}V_2\right).$$

A simple rearrangement of the terms leads to

$$\eta^2 n^2 \left(1 - \frac{3}{2}\mu\eta n - \left(1 + \frac{1}{2}\mu\eta n\right)\eta^2 n^2 L^2 - \frac{2 + \mu\eta n}{\mu}\cdot\eta^{2a-3}n^{2a-3}\Phi\right)\left\|\boldsymbol{F}\boldsymbol{z}^k\right\|^2$$

$$\leq \left(1 - \frac{1}{2}\mu\eta n\right)\left\|\boldsymbol{z}^k - \boldsymbol{z}^*\right\|^2 - \mathbb{E}\left[\left\|\boldsymbol{z}^{k+1} - \boldsymbol{z}^*\right\|^2 \,\Big|\, \boldsymbol{z}^k\right] + \frac{2 + \mu\eta n}{\mu}\cdot\eta^{2a-1}n^{2a-2}V_2. \tag{99}$$

Notice that by assuming (97a) and (97d), it holds that

$$\frac{3}{2}\mu\eta n + \left(1 + \frac{1}{2}\mu\eta n\right)\eta^2 n^2 L^2 + \frac{2 + \mu\eta n}{\mu}\cdot\eta^{2a-3}n^{2a-3}\Phi$$

$$\leq \frac{3}{2}\cdot\frac{2}{5} + \left(1 + \frac{1}{2}\cdot\frac{2}{5}\right)\left(\frac{2}{5}\right)^2 + \frac{12\Phi}{5\mu}\cdot\frac{\mu}{12\Phi} = \frac{124}{125},$$

so we can guarantee that the left hand side of (99) is nonnegative. It then follows that

$$\mathbb{E}\left[\left\|\boldsymbol{z}^{k+1} - \boldsymbol{z}^*\right\|^2 \,\Big|\, \boldsymbol{z}^k\right] \leq \left(1 - \frac{1}{2}\mu\eta n\right)\left\|\boldsymbol{z}^k - \boldsymbol{z}^*\right\|^2 + \frac{2 + \mu\eta n}{\mu}\cdot\eta^{2a-1}n^{2a-2}V_2.$$

Applying the law of total expectation, from the above we obtain

$$\mathbb{E}\left\|\boldsymbol{z}^{k+1} - \boldsymbol{z}^*\right\|^2 \leq \left(1 - \frac{1}{2}\mu\eta n\right)\mathbb{E}\left\|\boldsymbol{z}^k - \boldsymbol{z}^*\right\|^2 + \frac{2 + \mu\eta n}{\mu}\cdot\eta^{2a-1}n^{2a-2}V_2.$$

We can now unravel this recurrence over $k = 0, 1, \ldots, K - 1$ as done in (96) to get

$$\mathbb{E}\left\|\boldsymbol{z}^K - \boldsymbol{z}^*\right\|^2 \leq \left(1 - \frac{1}{2}\mu\eta n\right)\mathbb{E}\left\|\boldsymbol{z}^{K-1} - \boldsymbol{z}^*\right\|^2 + \frac{2 + \mu\eta n}{\mu}\cdot\eta^{2a-1}n^{2a-2}V_2$$

$$\leq \ldots$$

$$\leq \left(1 - \frac{1}{2}\mu\eta n\right)^K\left\|\boldsymbol{z}^0 - \boldsymbol{z}^*\right\|^2 + \frac{2 + \mu\eta n}{\mu}\cdot\eta^{2a-1}n^{2a-2}V_2\sum_{j=0}^{K-1}\left(1 - \frac{1}{2}\mu\eta n\right)^j$$

$$\leq \left(1 - \frac{1}{2}\mu\eta n\right)^K\left\|\boldsymbol{z}^0 - \boldsymbol{z}^*\right\|^2 + \frac{4 + 2\mu\eta n}{\mu^2\eta n}\cdot\eta^{2a-1}n^{2a-2}V_2$$

$$\leq \exp\left(-\frac{1}{2}\mu\eta n K\right)\left\|\boldsymbol{z}^0 - \boldsymbol{z}^*\right\|^2 + \frac{24}{5\mu^2}\cdot\eta^{2a-2}n^{2a-3}V_2$$

where in the last line we used the basic inequality $1 + x \leq e^x$ which holds for all $x \in \mathbb{R}$. With the choice of the stepsize (97e), we arrive at

$$\mathbb{E}\left\|\boldsymbol{z}^K - \boldsymbol{z}^*\right\|^2 \leq \exp\left(-\frac{1}{2}\mu\eta n K\right)\left\|\boldsymbol{z}^0 - \boldsymbol{z}^*\right\|^2 + \frac{24\cdot(4a-4)^{2a-2}V_2}{5\mu^{2a}}\cdot\frac{\left(\log(n^{1/(2a-2)}K)\right)^{2a-2}}{nK^{2a-2}}. \tag{100}$$

Now, recall that $\eta$ is chosen to be the smallest one among (97a)–(97e). Notice that the options (97a)–(97d) are independent with respect to $K$, and (97e) is the only one that depends on $K$. Let us consider these two cases separately.

(i) $\eta$ is chosen to be the minimum among (97a)–(97d).

This is the case where we have $\eta = \omega$. Notice that the constant $\omega$ that does not depend on $K$. The inequality (100) then takes the form

$$\mathbb{E}\left\|\boldsymbol{z}^K - \boldsymbol{z}^*\right\|^2 \leq \exp\left(-\frac{\mu\omega n K}{2}\right)\left\|\boldsymbol{z}^0 - \boldsymbol{z}^*\right\|^2 + \mathcal{O}\left(\frac{\left(\log(n^{1/(2a-2)}K)\right)^{2a-2}}{nK^{2a-2}}\right).$$

(ii) $\eta$ is chosen to be (97e), that is, $\eta = \frac{4(a-1)\log(n^{1/(2a-2)}K)}{\mu n K}$.

In this case, the exponential factor of the first term in the right hand side of (100) reduces to

$$\exp\left(-\frac{1}{2}\mu\eta nK\right) = \frac{1}{nK^{2a-2}}.$$

Thus, the second term in (100) dominates the first term, and in total (100) becomes

$$\mathbb{E}\left\|z^K - z^*\right\|^2 = \mathcal{O}\left(\frac{\left(\log(n^{1/(2a-2)}K)\right)^{2a-2}}{nK^{2a-2}}\right).$$

Therefore, in both cases we have

$$\mathbb{E}\left\|z^K - z^*\right\|^2 \leq \exp\left(-\frac{1}{2}\mu\omega nK\right)\left\|z^0 - z^*\right\|^2 + \mathcal{O}\left(\frac{\left(\log(n^{1/(2a-2)}K)\right)^{2a-2}}{nK^{2a-2}}\right)$$

which is exactly (98). This completes the proof. $\qquad\square$

*Remark* F.6. To compare the convergence rate of SEG-FFA in the strongly monotone setting with that of SEG-RR by Emmanouilidis et al. [18] more in depth, let us make an estimation on the size of $\omega$ appearing in Theorem F.5 when $a = 3$.

To this end, we need estimates on the constants $C_{1A}$, $D_{1A}$, $V_{1A}$, $C_{2A}$, and $D_{2A}$. From their definitions in (59)–(61), (72), and (73) we have $C_{1A} \asymp L^2$, $D_{1A} \asymp M$, $V_{1A} \asymp M + L^2$, $C_{2A} \asymp L^4$, and $D_{2A} \asymp M^2$. In general, there is not a direct relation between $L$ and $M$. For example, recall that if all components are quadratic, then $M = 0$. Meanwhile, Gorbunov et al. [21] has argued that $M$ can be much larger than $L$ in certain cases, by providing an example where $M \asymp L^{3/2}$. For our purposes, however, let us allow $M$ to be even as large as $M \asymp L^2$, so that the situation is simplified into $C_{1A} \asymp D_{1A} \asymp V_{1A} \asymp L^2$ and $C_{2A} \asymp D_{2A} \asymp L^4$.

Then, we get the estimate of (97c),

$$\frac{\mu^{1/2}}{n(10L(C_{1A} + D_{1A}(\|\boldsymbol{F}z_0\| + V_{1A}/\mu L)))^{1/2}} \asymp \frac{\mu}{nL^2}.$$

Meanwhile, as for the constant $\Phi$ it holds that

$$\Phi = C_{2A} + D_{2A}\left(\|\boldsymbol{F}z^0\| + \frac{V_{1A}}{\mu L}\right)^2 \asymp \frac{L^6}{\mu^2},$$

for (97d) we have

$$\frac{1}{(12\Phi/\mu)^{1/3}n} \asymp \frac{\mu}{nL^2}.$$

As (97a) while (97b) are both $\Theta(1/nL)$ and $\mu \leq L$, we essentially have $\omega \asymp \mu/nL^2$. Or equivalently, for some $b = \Theta(1)$, the convergence rate (98) reads

$$\mathbb{E}\left\|z^K - z^*\right\|^2 \leq \exp\left(-\frac{b\mu^2 K}{L^2}\right)\left\|z^0 - z^*\right\|^2 + \mathcal{O}\left(\frac{\left(\log(n^{1/4}K)\right)^4}{nK^4}\right). \tag{101}$$

On the other hand, Theorem 2.1 of [18] states that, for some $b' = \Theta(1)$, SEG-RR exhibits a rate of

$$\mathbb{E}\left\|z^K - z^*\right\|^2 \leq \exp\left(-\frac{b'\mu^2 K}{L^2}\right)\left\|z^0 - z^*\right\|^2 + \mathcal{O}\left(\frac{\left(\log(n^{1/2}K)\right)^2}{nK^2}\right). \tag{102}$$

Comparing (101) with (102), the exponents in the exponentially decaying term are of the same order of $-\frac{\mu^2 K}{L^2}$, so SEG-FFA having a faster polynomially decaying term $\tilde{\mathcal{O}}(1/nK^4)$ enjoys an improved convergence rate.

# G Convergence Rate of SEG-FFA in the Monotone Setting

## G.1 Star-monotonicity

Notice that we only used Assumptions 3.3 and 3.4 in deriving the results in Appendices D and E, and in particular, the monotonicity assumption on $\boldsymbol{F}$ was not necessary. Moreover, among the lemmata listed in Appendix C, Lemma C.10 is the only one that possibly uses the (non-strongly) monotone assumption, but that lemma is not used in this section.

In fact, as it turns out in Appendix G.2, in the convergence analysis of SEG-FFA, we need not fully exploit the inequality (3) provided by the monotonicity assumption. Rather, all the results on the performance of SEG-FFA can be established with only assuming the following condition (which has been also briefly mentioned in Appendix B.2).

**Assumption G.1** (Star-monotonicity). Given an operator $\boldsymbol{F}$ with a point $\boldsymbol{z}^* \in \mathbb{R}^{d_1+d_2}$ such that $\boldsymbol{F}\boldsymbol{z}^* = \boldsymbol{0}$, we say that $\boldsymbol{F}$ is star-monotone if, for any $\boldsymbol{z} \in \mathbb{R}^{d_1+d_2}$, it holds that

$$\langle \boldsymbol{F}\boldsymbol{z}, \boldsymbol{z} - \boldsymbol{z}^* \rangle \geq 0. \tag{103}$$

Monotone and strongly-monotone operators are clearly star-monotone, as they satisfy (3). On the other hand, there exist operators that are star-monotone but not monotone: see, *e.g.*, [31, Appendix A.6].

Recall that when $\boldsymbol{F}$ is monotone, Assumption 3.2 is equivalent to assuming the existence of a point $\boldsymbol{z}^*$ that satisfies $\boldsymbol{F}\boldsymbol{z}^* = \boldsymbol{0}$. Hence, after simply replacing the optimality condition in Assumption 3.2 with $\boldsymbol{F}\boldsymbol{z}^* = \boldsymbol{0}$, our convergence analyses not only will show that our SEG-FFA finds an optimum on monotone problems, but also that it can be also used to find stationary points in "star-monotone" problems, allowing the objective function $f$ to be nonconvex-nonconcave.

Star-monotonicity is also known as the *variational stability condition* [25], and has much been studied in the literature. For further details on star-monotonicity, we refer to [25, 31] and the references therein.

## G.2 Convergence Analysis of SEG-FFA in the (Star-)Monotone Setting

Let us in particular consider SEG-FFA. As in the previous section, we focus only on the iterates $\{\boldsymbol{z}_0^k\}_{k \geq 0}$, so again, we omit the subscript 0 unless necessary, and simply write $\boldsymbol{z}^k$ instead of $\boldsymbol{z}_0^k$.

Decompose the update across the epoch into a deterministic EG update plus a noise, as

$$
\begin{aligned}
\boldsymbol{w}_\dagger^k &:= \boldsymbol{z}^k - \eta_k n \boldsymbol{F}\boldsymbol{z}^k, \\
\boldsymbol{z}^{k+1} &= \boldsymbol{z}^k - \eta_k n \widehat{\boldsymbol{F}}^k.
\end{aligned}
\tag{104}
$$

for $\widehat{\boldsymbol{F}}^k$ defined by the equation

$$\eta_k n \widehat{\boldsymbol{F}}^k = \eta_k n \boldsymbol{F}\boldsymbol{w}_\dagger^k + \boldsymbol{r}^k. \tag{105}$$

**Lemma G.2.** *Let $\boldsymbol{F}$ be a (star-)monotone operator with a point $\boldsymbol{z}^*$ that satisfies $\boldsymbol{F}\boldsymbol{z}^* = \boldsymbol{0}$, and suppose that Assumption 3.3 holds. Then for any $\eta_k > 0$ and $\gamma_k > 0$, it holds that*

$$0 \leq \left\| \boldsymbol{z}^k - \boldsymbol{z}^* \right\|^2 - \frac{1}{1+\gamma_k} \left\| \boldsymbol{z}^{k+1} - \boldsymbol{z}^* \right\|^2 - \eta_k^2 n^2 (1 - \eta_k^2 n^2 L^2) \left\| \boldsymbol{F}\boldsymbol{z}^k \right\|^2 + \frac{1}{\gamma_k} \left\| \boldsymbol{r}^k \right\|^2. \tag{106}$$

*Proof.* By (104) and (105) we get

$$
\begin{aligned}
\left\| \boldsymbol{z}^{k+1} - \boldsymbol{z}^* \right\|^2 &= \left\| \boldsymbol{z}^k - \eta_k n \widehat{\boldsymbol{F}}^k - \boldsymbol{z}^* \right\|^2 \\
&= \left\| \boldsymbol{z}^k - \boldsymbol{z}^* \right\|^2 - 2 \left\langle \eta_k n \widehat{\boldsymbol{F}}^k, \boldsymbol{z}^k - \boldsymbol{z}^* \right\rangle + \left\| \eta_k n \widehat{\boldsymbol{F}}^k \right\|^2 \\
&= \left\| \boldsymbol{z}^k - \boldsymbol{z}^* \right\|^2 - 2 \left\langle \eta_k n \boldsymbol{F}\boldsymbol{w}_\dagger^k, \boldsymbol{w}_\dagger^k - \boldsymbol{z}^* \right\rangle - 2 \left\langle \eta_k n \boldsymbol{F}\boldsymbol{w}_\dagger^k, \boldsymbol{z}^k - \boldsymbol{w}_\dagger^k \right\rangle - 2 \left\langle \boldsymbol{r}^k, \boldsymbol{z}^k - \boldsymbol{z}^* \right\rangle \\
&\quad + \left\| \eta_k n \boldsymbol{F}\boldsymbol{w}_\dagger^k \right\|^2 + 2 \left\langle \boldsymbol{r}^k, \eta_k n \boldsymbol{F}\boldsymbol{w}_\dagger^k \right\rangle + \left\| \boldsymbol{r}^k \right\|^2 \\
&= \left\| \boldsymbol{z}^k - \boldsymbol{z}^* \right\|^2 - 2\eta_k n \left\langle \boldsymbol{F}\boldsymbol{w}_\dagger^k, \boldsymbol{w}_\dagger^k - \boldsymbol{z}^* \right\rangle - 2 \left\langle \eta_k n \boldsymbol{F}\boldsymbol{w}_\dagger^k, \eta_k n \boldsymbol{F}\boldsymbol{z}^k \right\rangle \\
&\quad + \left\| \eta_k n \boldsymbol{F}\boldsymbol{w}_\dagger^k \right\|^2 - 2 \left\langle \boldsymbol{r}^k, \boldsymbol{z}^k - \eta_k n \boldsymbol{F}\boldsymbol{w}_\dagger^k - \boldsymbol{z}^* \right\rangle + \left\| \boldsymbol{r}^k \right\|^2
\end{aligned}
$$

$$= \left\| \boldsymbol{z}^k - \boldsymbol{z}^* \right\|^2 - 2\eta_k n \left\langle \boldsymbol{F}\boldsymbol{w}_{\dagger}^k, \boldsymbol{w}_{\dagger}^k - \boldsymbol{z}^* \right\rangle - 2\eta_k^2 n^2 \left\langle \boldsymbol{F}\boldsymbol{w}_{\dagger}^k, \boldsymbol{F}\boldsymbol{z}^k \right\rangle$$
$$+ \eta_k^2 n^2 \left\| \boldsymbol{F}\boldsymbol{w}_{\dagger}^k \right\|^2 - 2 \left\langle \boldsymbol{r}^k, \boldsymbol{z}^{k+1} - \boldsymbol{z}^* \right\rangle - \left\| \boldsymbol{r}^k \right\|^2.$$

We now bound the inner products. On one hand, by the polarization identity (Lemma C.1) and the $L$-smoothness of $f$, we have

$$-2 \left\langle \boldsymbol{F}\boldsymbol{w}_{\dagger}^k, \boldsymbol{F}\boldsymbol{z}^k \right\rangle = \left\| \boldsymbol{F}\boldsymbol{w}_{\dagger}^k - \boldsymbol{F}\boldsymbol{z}^k \right\|^2 - \left\| \boldsymbol{F}\boldsymbol{w}_{\dagger}^k \right\|^2 - \left\| \boldsymbol{F}\boldsymbol{z}^k \right\|^2$$
$$\leq L^2 \left\| -\eta_k n \boldsymbol{F}\boldsymbol{z}^k \right\|^2 - \left\| \boldsymbol{F}\boldsymbol{w}_{\dagger}^k \right\|^2 - \left\| \boldsymbol{F}\boldsymbol{z}^k \right\|^2$$
$$= -(1 - \eta_k^2 n^2 L^2) \left\| \boldsymbol{F}\boldsymbol{z}^k \right\|^2 - \left\| \boldsymbol{F}\boldsymbol{w}_{\dagger}^k \right\|^2.$$

On the other hand, by the weighted AM-GM inequality (Lemma C.2), for any number $a_k \in (0, 1)$ it holds that

$$-2 \left\langle \boldsymbol{r}^k, \boldsymbol{z}^{k+1} - \boldsymbol{z}^* \right\rangle \leq \frac{1}{a_k} \left\| \boldsymbol{r}^k \right\|^2 + a_k \left\| \boldsymbol{z}^{k+1} - \boldsymbol{z}^* \right\|^2.$$

Using these two bounds, we get

$$\left\| \boldsymbol{z}^{k+1} - \boldsymbol{z}^* \right\|^2 \leq \left\| \boldsymbol{z}^k - \boldsymbol{z}^* \right\|^2 - 2\eta_k n \left\langle \boldsymbol{F}\boldsymbol{w}_{\dagger}^k, \boldsymbol{w}_{\dagger}^k - \boldsymbol{z}^* \right\rangle - \eta_k^2 n^2 (1 - \eta_k^2 n^2 L^2) \left\| \boldsymbol{F}\boldsymbol{z}^k \right\|^2$$
$$- \eta_k^2 n^2 \left\| \boldsymbol{F}\boldsymbol{w}_{\dagger}^k \right\|^2 + \eta_k^2 n^2 \left\| \boldsymbol{F}\boldsymbol{w}_{\dagger}^k \right\|^2 + a_k \left\| \boldsymbol{z}^{k+1} - \boldsymbol{z}^* \right\|^2 + \left( \frac{1}{a_k} - 1 \right) \left\| \boldsymbol{r}^k \right\|^2.$$

Choosing $a_k = \frac{\gamma_k}{1+\gamma_k}$ and rearranging the terms, we obtain

$$2\eta_k n \left\langle \boldsymbol{F}\boldsymbol{w}_{\dagger}^k, \boldsymbol{w}_{\dagger}^k - \boldsymbol{z}^* \right\rangle \leq \left\| \boldsymbol{z}^k - \boldsymbol{z}^* \right\|^2 - \frac{1}{1+\gamma_k} \left\| \boldsymbol{z}^{k+1} - \boldsymbol{z}^* \right\|^2$$
$$- \eta_k^2 n^2 (1 - \eta_k^2 n^2 L^2) \left\| \boldsymbol{F}\boldsymbol{z}^k \right\|^2 + \frac{1}{\gamma_k} \left\| \boldsymbol{r}^k \right\|^2. \tag{107}$$

The left hand side of (107) is nonnegative by the star-monotonicity of $\boldsymbol{F}$ (103), and the claimed inequality follows. $\qquad\square$

Now we show that choosing the appropriate stepsizes leads to $\left\| \boldsymbol{F}\boldsymbol{z}^k \right\|$ being bounded uniformly over $k$.

**Proposition G.3.** *Let $\boldsymbol{F}$ be a (star-)monotone operator with a point $\boldsymbol{z}^*$ that satisfies $\boldsymbol{F}\boldsymbol{z}^* = \boldsymbol{0}$, and suppose that Assumptions 3.3 and 3.4 hold. Say we are using SEG-FFA, or any optimization method whose within-epoch error satisfies (58) and (71). Let the sequence of stepsizes $\{\eta_k\}_{k \geq 0}$ be nonincreasing, with*

$$S := \sum_{k=0}^{\infty} \eta_k^3 n^3 L^3 < \infty. \tag{108}$$

*Suppose that initial stepsize $\eta_0$ is chosen sufficiently small so that*

$$\eta_0^2 n^2 L^2 + \frac{3\eta_0 n C_{1A}^2}{L^3} + \frac{3\eta_0 n D_{1A}^2}{L} \cdot e^S \left( \left\| \boldsymbol{z}^0 - \boldsymbol{z}^* \right\|^2 + \frac{6 S V_{1A}^2}{L^6} \right) \leq 1 \tag{109}$$

*for constants $C_{1A}$, $D_{1A}$, and $V_{1A}$ defined in (59)–(61). Then for all $k \geq 0$,*

$$\left\| \boldsymbol{F}\boldsymbol{z}^k \right\|^2 \leq e^S L^2 \left( \left\| \boldsymbol{z}^0 - \boldsymbol{z}^* \right\|^2 + \frac{6 S V_{1A}^2}{L^6} \right). \tag{110}$$

*Proof.* We use induction on $k$, to establish a stronger inequality

$$\left\| \boldsymbol{z}^k - \boldsymbol{z}^* \right\|^2 \leq e^S \left( \left\| \boldsymbol{z}^0 - \boldsymbol{z}^* \right\|^2 + \frac{6 S V_{1A}^2}{L^6} \right). \tag{111}$$

To see that (111) indeed implies (110), notice that by the $L$-smoothness of $f$ it holds that

$$\left\| \boldsymbol{F}\boldsymbol{z}^k \right\|^2 = \left\| \boldsymbol{F}\boldsymbol{z}^k - \boldsymbol{F}\boldsymbol{z}^* \right\|^2 \leq L^2 \left\| \boldsymbol{z}^k - \boldsymbol{z}^* \right\|^2.$$

For the case when $k = 0$, as $S > 0$, it is clear that (111) holds. Now suppose that (111) holds for some $k \geq 0$. Applying Young's inequality on (31) leads to

$$\left\| r^k \right\|^2 \leq 3\eta_k^6 n^6 \left( C_{1A}^2 \left\| F z^k \right\|^2 + D_{1A}^2 \left\| F z^k \right\|^4 + V_{1A}^2 \right).$$

Applying this bound on $\left\| r^k \right\|^2$ on (106), we obtain

$$\eta_k^2 n^2 \left( 1 - \eta_k^2 n^2 L^2 - \frac{3\eta_k^4 n^4 C_{1A}^2}{\gamma_k} - \frac{3\eta_k^4 n^4 D_{1A}^2}{\gamma_k} \left\| F z^k \right\|^2 \right) \left\| F z^k \right\|^2$$
$$\leq \left\| z^k - z^* \right\|^2 - \frac{1}{1 + \gamma_k} \left\| z^{k+1} - z^* \right\|^2 + \frac{3\eta_k^6 n^6 V_{1A}^2}{\gamma_k}. \tag{112}$$

Choose $\gamma_k = \eta_k^3 n^3 L^3$. Notice that (109) implies $\eta_0 n L \leq 1$, henceforth $\eta_k \leq \eta_0 \leq 1/nL$. This, with the induction hypothesis (110), implies

$$\eta_k^2 n^2 L^2 + \frac{3\eta_k^4 n^4 C_{1A}^2}{\gamma_k} + \frac{3\eta_k^4 n^4 D_{1A}^2}{\gamma_k} \left\| F z^k \right\|^2$$
$$= \eta_k^2 n^2 L^2 + \frac{3\eta_k n C_{1A}^2}{L^3} + \frac{3\eta_k n D_{1A}^2}{L^3} \left\| F z^k \right\|^2$$
$$\leq \eta_0^2 n^2 L^2 + \frac{3\eta_0 n C_{1A}^2}{L^3} + \frac{3\eta_0 n D_{1A}^2}{L^3} \left\| F z^k \right\|^2$$
$$\leq \eta_0^2 n^2 L^2 + \frac{3\eta_0 n C_{1A}^2}{L^3} + \frac{3\eta_0 n D_{1A}^2}{L^3} \cdot e^S L^2 \left( \left\| z^0 - z^* \right\|^2 + \frac{6 S V_{1A}^2}{L^6} \right)$$
$$\leq 1.$$

That is, the left hand side of (112) becomes nonnegative. Then it is immediate from (112) that

$$\left\| z^{k+1} - z^* \right\|^2 \leq (1 + \gamma_k) \left\| z^k - z^* \right\|^2 + \frac{3\eta_k^6 n^6 (1 + \gamma_k) V_{1A}^2}{\gamma_k}$$
$$\leq \left( 1 + \eta_k^3 n^3 L^3 \right) \left\| z^k - z^* \right\|^2 + \frac{6\eta_k^3 n^3 V_{1A}^2}{L^3}.$$

Using Lemma C.11 to unravel this recurrence relation, we obtain

$$\left\| z^{k+1} - z^* \right\|^2 \leq \left( \prod_{j=0}^k \left( 1 + \eta_j^3 n^3 L^3 \right) \right) \left( \left\| z^0 - z^* \right\|^2 + \sum_{j=0}^k \frac{6\eta_j^3 n^3 V_{1A}^2}{L^3} \right)$$
$$\leq e^{\sum_{j=0}^k \eta_j^3 n^3 L^3} \left( \left\| z^0 - z^* \right\|^2 + \frac{6 V_{1A}^2}{L^6} \sum_{j=0}^k \eta_j^3 n^3 L^3 \right)$$
$$\leq e^S \left( \left\| z^0 - z^* \right\|^2 + \frac{6 S V_{1A}^2}{L^6} \right)$$

which shows that (111) also holds when $k$ is replaced by $k + 1$. This completes the proof. $\qquad \square$

**Theorem G.4** (Formal version of Theorem 5.4). *Let $F$ be a (star-)monotone operator with a point $z^*$ that satisfies $F z^* = 0$, and suppose that Assumptions 3.3 and 3.4 hold. Say that we are using SEG-FFA, or any optimization method whose within-epoch error satisfies (58) and (71), with $\beta_k = \eta_k = \frac{\eta_0 \sqrt[3]{2} \log 2}{(k+2)^{1/3} \log(k+2)}$ and $\alpha_k = \beta_k/2$ for $k = 0, 1, \ldots$, where, for $S := \sum_{k=0}^\infty \eta_k^3 n^3 L^3$, the initial stepsize $\eta_0$ is chosen so that*

$$\eta_0^2 n^2 L^2 + \frac{3\eta_0 n C_{1A}^2}{L^3} + \frac{3\eta_0 n D_{1A}^2}{L} \cdot e^S \left( \left\| z^0 - z^* \right\|^2 + \frac{6 S V_{1A}^2}{L^6} \right) \leq 1 \tag{113}$$

*for constants $C_{1A}$, $D_{1A}$, and $V_{1A}$ defined in (59)–(61), and there exists a positive constant $\lambda > 0$ such that*

$$\eta_0^2 n^2 L^2 + \frac{\eta_0 n C_{2A}}{L^3} + \frac{\eta_0 n D_{2A}}{L} \cdot e^S \left( \left\| z^0 - z^* \right\|^2 + \frac{6 S V_{1A}^2}{L^6} \right) \leq 1 - \lambda \tag{114}$$

*for constants $C_{2A}$, $D_{2A}$, and $V_{2A}$ defined in (72)–(74). Then for any $K \geq 1$, it holds that*

$$\min_{k=0,1,\ldots,K} \mathbb{E}\left\|\boldsymbol{F}\boldsymbol{z}^k\right\|^2 \leq \frac{(\log(K+3))^2}{(K+3)^{1/3}} \cdot \left(\frac{\left\|\boldsymbol{z}^0 - \boldsymbol{z}^*\right\|^2 + \frac{3V_{2A}}{nL^6}}{\lambda e^{-3/2}(\sqrt[3]{2}\log 2)^2}\eta_0^2 n^2\right). \tag{115}$$

*Proof.* As the sequence of stepsizes $\{\eta_k\}_{k\geq 0}$ is nonincreasing and (113) asserts that $\eta_0 \leq 1/nL$, we can use the bounds established in Theorem E.9 and Theorem E.13. Also, the premises required for Proposition G.3 are also satisfied, so the bound (110) holds.

Setting $\gamma_k = \eta_k^3 n^3 L^3$ in (106) and then taking the conditional expectation given $\boldsymbol{z}^k$, with using (71) and (114), we obtain

$$0 \leq \left\|\boldsymbol{z}^k - \boldsymbol{z}^*\right\|^2 - \frac{1}{1+\gamma_k}\mathbb{E}\left[\left\|\boldsymbol{z}^{k+1} - \boldsymbol{z}^*\right\|^2 \Big| \boldsymbol{z}^k\right] - \eta_k^2 n^2(1 - \eta_k^2 n^2 L^2)\left\|\boldsymbol{F}\boldsymbol{z}^k\right\|^2 + \frac{1}{\gamma_k}\mathbb{E}\left[\left\|\boldsymbol{r}^k\right\|^2 \Big| \boldsymbol{z}^k\right]$$

$$\leq \left\|\boldsymbol{z}^k - \boldsymbol{z}^*\right\|^2 - \frac{1}{1+\gamma_k}\mathbb{E}\left[\left\|\boldsymbol{z}^{k+1} - \boldsymbol{z}^*\right\|^2 \Big| \boldsymbol{z}^k\right]$$
$$- \eta_k^2 n^2(1 - \eta_k^2 n^2 L^2)\left\|\boldsymbol{F}\boldsymbol{z}^k\right\|^2 + \frac{1}{L^3}\left(\eta_k^3 n^3 C_{2A}\left\|\boldsymbol{F}\boldsymbol{z}^k\right\|^2 + \eta_k^3 n^3 D_{2A}\left\|\boldsymbol{F}\boldsymbol{z}^k\right\|^4 + \eta_k^3 n^2 V_{2A}\right)$$

$$\leq \left\|\boldsymbol{z}^k - \boldsymbol{z}^*\right\|^2 - \frac{1}{1+\gamma_k}\mathbb{E}\left[\left\|\boldsymbol{z}^{k+1} - \boldsymbol{z}^*\right\|^2 \Big| \boldsymbol{z}^k\right]$$
$$- \eta_k^2 n^2\left(1 - \eta_k^2 n^2 L^2 - \frac{\eta_k n C_{2A}}{L^3} - \frac{\eta_k n D_{2A}}{L^3}\left\|\boldsymbol{F}\boldsymbol{z}^k\right\|^2\right)\left\|\boldsymbol{F}\boldsymbol{z}^k\right\|^2 + \frac{\eta_k^3 n^2 V_{2A}}{L^3}$$

$$\leq \left\|\boldsymbol{z}^k - \boldsymbol{z}^*\right\|^2 - \frac{1}{1+\gamma_k}\mathbb{E}\left[\left\|\boldsymbol{z}^{k+1} - \boldsymbol{z}^*\right\|^2 \Big| \boldsymbol{z}^k\right] - \lambda\eta_k^2 n^2\left\|\boldsymbol{F}\boldsymbol{z}^k\right\|^2 + \frac{\eta_k^3 n^2 V_{2A}}{L^3}.$$

By the law of total expectation, and that $\gamma_k = \eta_k^3 n^3 L^3 < 1$, from the above we get

$$(1+\gamma_k)\lambda\eta_k^2 n^2 \mathbb{E}\left\|\boldsymbol{F}\boldsymbol{z}^k\right\|^2 \leq (1+\gamma_k)\mathbb{E}\left\|\boldsymbol{z}^k - \boldsymbol{z}^*\right\|^2 - \mathbb{E}\left\|\boldsymbol{z}^{k+1} - \boldsymbol{z}^*\right\|^2 + \frac{(1+\gamma_k)\eta_k^3 n^2 V_{2A}}{L^3}$$
$$\leq (1+\gamma_k)\mathbb{E}\left\|\boldsymbol{z}^k - \boldsymbol{z}^*\right\|^2 - \mathbb{E}\left\|\boldsymbol{z}^{k+1} - \boldsymbol{z}^*\right\|^2 + \frac{2\eta_k^3 n^2 V_{2A}}{L^3}.$$

This recurrence can be unraveled using Lemma C.11, giving us

$$\mathbb{E}\left\|\boldsymbol{z}^{K+1} - \boldsymbol{z}^*\right\|^2 + \sum_{k=0}^{K}(1+\gamma_k)\lambda\eta_j^2 n^2 \mathbb{E}\left\|\boldsymbol{F}\boldsymbol{z}^k\right\|^2$$
$$\leq \left(\prod_{k=0}^{K}(1+\gamma_k)\right)\left(\left\|\boldsymbol{z}^0 - \boldsymbol{z}^*\right\|^2 + \sum_{k=0}^{K}\frac{2\eta_k^3 n^2 V_{2A}}{L^3}\right). \tag{116}$$

For the left hand side of (116), we have

$$\mathbb{E}\left\|\boldsymbol{z}^{K+1} - \boldsymbol{z}^*\right\|^2 + \sum_{k=0}^{K}(1+\gamma_k)\lambda\eta_k^2 n^2 \mathbb{E}\left\|\boldsymbol{F}\boldsymbol{z}^k\right\|^2 \geq \lambda\sum_{k=0}^{K}\eta_k^2 n^2 \mathbb{E}\left\|\boldsymbol{F}\boldsymbol{z}^k\right\|^2$$
$$\geq \lambda \min_{k=0,1,\ldots,K}\mathbb{E}\left\|\boldsymbol{F}\boldsymbol{z}^k\right\|^2 \sum_{k=0}^{K}\eta_k^2 n^2. \tag{117}$$

From Lemma C.12, we know that whenever $K \geq 1$,

$$\sum_{k=0}^{K}\eta_k^2 n^2 = \eta_0^2 n^2(\sqrt[3]{2}\log 2)^2 \sum_{k=0}^{K}\frac{1}{(k+2)^{2/3}(\log(k+2))^2}$$
$$\geq \eta_0^2 n^2(\sqrt[3]{2}\log 2)^2 \cdot \frac{(K+3)^{1/3}}{(\log(K+3))^2}.$$

Meanwhile, as $x \mapsto \frac{2(\log 2)^3}{(x+2)(\log(x+2))^3}$ is a decreasing function, we have

$$\sum_{k=0}^{\infty} \frac{2(\log 2)^3}{(k+2)(\log(k+2))^3} \le 1 + \frac{2(\log 2)^3}{3(\log 3)^3} + \int_1^{\infty} \frac{2(\log 2)^3}{(x+2)(\log(x+2))^3} \, \mathrm{d}x$$

$$\le 1 + \frac{2(\log 2)^3}{3(\log 3)^3} + \frac{(\log 2)^3}{(\log 3)^2} \quad \le \frac{3}{2}$$

and thus

$$S = \sum_{k=0}^{\infty} \eta_k^3 n^3 L^3 = \eta_0^3 n^3 L^3 \sum_{k=0}^{\infty} \frac{2(\log 2)^3}{(k+2)(\log(k+2))^3} \le \frac{3}{2} \eta_0^3 n^3 L^3 \le \frac{3}{2}.$$

Thus, for the right hand side of (116), it holds that

$$\left( \prod_{k=0}^{K} (1 + \gamma_k) \right) \left( \left\| z^0 - z^* \right\|^2 + \sum_{k=0}^{K} \frac{2 \eta_k^3 n^2 V_{2A}}{L^3} \right) \le e^{\sum_{k=0}^{K} \gamma_k} \left( \left\| z^0 - z^* \right\|^2 + \sum_{k=0}^{K} \frac{2 \eta_k^3 n^2 V_{2A}}{L^3} \right)$$

$$\le e^S \left( \left\| z^0 - z^* \right\|^2 + \frac{2 S V_{2A}}{n L^6} \right)$$

$$\le e^{3/2} \left( \left\| z^0 - z^* \right\|^2 + \frac{3 V_{2A}}{n L^6} \right). \tag{118}$$

Therefore, from (116) we get

$$\lambda \eta_0^2 n^2 (\sqrt[3]{2} \log 2)^2 \cdot \frac{(K+3)^{1/3}}{(\log(K+3))^2} \cdot \min_{k=0,1,\dots,K} \mathbb{E} \left\| F z^k \right\|^2 \le e^{3/2} \left( \left\| z^0 - z^* \right\|^2 + \frac{3 V_{2A}}{n L^6} \right).$$

Simply rearranging the terms gives us the desired inequality. $\qquad \square$

*Remark* G.5. While $\eta_0$ should be chosen so that both (113) and (114) hold, in practice, there is a way to circumvent this complication. Notice that in deriving the upper bound (118) of the right hand side of (116), it suffices to have $\eta_k \le \frac{\eta_0 \sqrt[3]{2} \log 2}{(k+2)^{1/3} \log(k+2)}$, and the lower bound (117) of the left hand side holds for any $\eta_k \ge 0$. In other words, if we have had chosen $\eta_k = \Theta\left(1/(k+1)^q\right)$ for $q > \frac{1}{3}$ so that $S = \sum_{k=0}^{\infty} \eta_k^3 n^3 L^3 < \infty$, as long as $\eta_0$ satisfies (113) and (114), we would still have obtained the inequality

$$\min_{k=0,1,\dots,K} \mathbb{E} \left\| F z^k \right\|^2 \le \frac{e^S}{\lambda n^2 \sum_{k=0}^{K} \eta_k^2} \left( \left\| z^0 - z^* \right\|^2 + \frac{2 S V_{2A}}{n L^6} \right). \tag{119}$$

In particular, if we additionally assume that $q < \frac{1}{2}$ then

$$\sum_{k=0}^{K} \eta_k^2 \asymp \sum_{k=1}^{K} \frac{1}{k^{2q}} \asymp K^{1-2q},$$

so from (119) we would have obtained the convergence rate

$$\min_{k=0,1,\dots,K} \mathbb{E} \left\| F z^k \right\|^2 = \mathcal{O}\left( \frac{1}{K^{1-2q}} \right). \tag{120}$$

We now claim that, if one accepts a slight sacrifice of the convergence rate from $\tilde{\mathcal{O}}(1/K^{1/3})$ to $\mathcal{O}\left(1/K^{1-2q}\right)$ for $1/3 < q < 1/2$, one can simply choose the stepsizes as $\eta_k = \eta_{00}/(k+1)^q$ for a sufficiently small $\eta_{00}$. To see why this is the case, let us fix $\eta_0$ to be a number that satisfies the inequalities (113) and (114). Then, because $\eta_{00}/(k+1)^q = o\left( \frac{1}{(k+2)^{1/3} \log(k+2)} \right)$, there will exist a nonnegative integer $k_0$ such that $\eta_k \le \frac{\eta_0 \sqrt[3]{2} \log 2}{(k+2)^{1/3} \log(k+2)}$ for all $k \ge k_0$. So, by ignoring the first $k_0$ terms if necessary—that is, considering as if the $k_0$th iteration is the 0th iteration—it follows from the discussions made above in obtaining (120) that we get the rate of convergence $\mathcal{O}\left(1/K^{1-2q}\right)$.

This discussion also justifies the choice of stepsizes $\eta_k = \Theta\left(1/(1+k/10)^{0.34}\right)$ used in the experiments for the monotone setting.

# H Proof of Lower Bounds

## H.1 Proof of the Divergence of SEG-US, SEG-RR and SEG-FF

We prove the divergence of SEG-US, SEG-RR and SEG-FF in each proposition below, using the same worst-case problem for $n = 2$. These constitute the proof of Theorem 4.1.

**Proposition H.1** (Part of Theorem 4.1). *For $n = 2$, there exists a convex-concave minimax problem $f(x, y) = \frac{1}{2} \sum_{i=1}^{2} f_i(x, y)$ having a monotone $\boldsymbol{F}$, consisting of L-smooth quadratic $f_i$'s satisfying Assumption 3.4 with $(\rho, \sigma) = (1, 0)$ such that SEG-US diverges in expectation for any choice of stepsizes $\{\alpha_t\}_{t \geq 0}$ and $\{\beta_t\}_{t \geq 0}$. That is, for all $t \geq 0$,*

$$\mathbb{E}\left[\|\boldsymbol{z}_{t+1}\|^2\right] > \mathbb{E}\left[\|\boldsymbol{z}_t\|^2\right], \quad \mathbb{E}\left[\|\boldsymbol{F}\boldsymbol{z}_{t+1}\|^2\right] > \mathbb{E}\left[\|\boldsymbol{F}\boldsymbol{z}_t\|^2\right].$$

*Proof.* We consider the case of

$$f_1(x, y) = -\frac{L}{4}x^2 + \frac{L}{2}xy - \frac{L}{4}y^2,$$

$$f_2(x, y) = \frac{L}{4}x^2 + \frac{L}{2}xy + \frac{L}{4}y^2,$$

which result in a bilinear (and hence convex-concave) objective function

$$f(x, y) = \frac{1}{2} \sum_{i=1}^{2} f_i(x, y) = \frac{L}{2}xy. \tag{121}$$

One can quickly check from the definitions of the component functions $f_1$ and $f_2$ that the corresponding saddle gradient operators are given as

$$\boldsymbol{F}_1\boldsymbol{z} = \underbrace{\begin{bmatrix} -L/2 & L/2 \\ -L/2 & L/2 \end{bmatrix}}_{:=\boldsymbol{A}_1} \boldsymbol{z}, \quad \boldsymbol{F}_2\boldsymbol{z} = \underbrace{\begin{bmatrix} L/2 & L/2 \\ -L/2 & -L/2 \end{bmatrix}}_{:=\boldsymbol{A}_2} \boldsymbol{z}, \quad \boldsymbol{F}\boldsymbol{z} = \begin{bmatrix} 0 & L/2 \\ -L/2 & 0 \end{bmatrix} \boldsymbol{z}$$

where $\boldsymbol{z} = (x, y) \in \mathbb{R}^2$. From the fact that $\|\boldsymbol{A}_i\| \leq L$ for all $i$'s, we can confirm that $f_i$'s are indeed $L$-smooth. As for Assumption 3.4, we can verify that

$$\frac{1}{2} \sum_{i=1}^{2} \|\boldsymbol{F}_i\boldsymbol{z} - \boldsymbol{F}\boldsymbol{z}\|^2 = \frac{L^2}{4} \|\boldsymbol{z}\|^2 = \|\boldsymbol{F}\boldsymbol{z}\|^2,$$

thus proving that our example $f$ indeed satisfies Assumption 3.4 with $(\rho, \sigma) = (1, 0)$.

We now proceed to show that for this particular worst-case example $f$, SEG-US diverges in expectation. For $t \geq 0$, the $(t + 1)$-th iteration of SEG-US starts at $\boldsymbol{z}_t$, and the algorithm uniformly chooses an index $i(t)$ from $[n]$. The algorithm then makes an update

$$\boldsymbol{w}_t = \boldsymbol{z}_t - \alpha_t \boldsymbol{F}_{i(t)}\boldsymbol{z}_t,$$

$$\boldsymbol{z}_{t+1} = \boldsymbol{z}_t - \beta_t \boldsymbol{F}_{i(t)}\boldsymbol{w}_t.$$

In our worst-case example $f$, the updates can be compactly written as

$$\boldsymbol{z}_{t+1} = (\boldsymbol{I} - \beta_t \boldsymbol{A}_{i(t)} + \alpha_t \beta_t \boldsymbol{A}_{i(t)}^2)\boldsymbol{z}_t.$$

Since we have $n = 2$, the update can be summarized as

$$\boldsymbol{z}_{t+1} = \begin{cases} (\boldsymbol{I} - \beta_t \boldsymbol{A}_1 + \alpha_t \beta_t \boldsymbol{A}_1^2)\boldsymbol{z}_t & \text{with probability } 1/2, \\ (\boldsymbol{I} - \beta_t \boldsymbol{A}_2 + \alpha_t \beta_t \boldsymbol{A}_2^2)\boldsymbol{z}_t & \text{with probability } 1/2. \end{cases}$$

By the definition of $\boldsymbol{A}_1$ and $\boldsymbol{A}_2$ and using $\boldsymbol{A}_1^2 = \boldsymbol{A}_2^2 = \boldsymbol{0}$, we can verify that

$$\boldsymbol{N}_1 := \boldsymbol{I} - \beta_t \boldsymbol{A}_1 + \alpha_t \beta_t \boldsymbol{A}_1^2 = \begin{bmatrix} 1 + \frac{\beta_t L}{2} & -\frac{\beta_t L}{2} \\ \frac{\beta_t L}{2} & 1 - \frac{\beta_t L}{2} \end{bmatrix},$$

$$\boldsymbol{N}_2 := \boldsymbol{I} - \beta_t \boldsymbol{A}_2 + \alpha_t \beta_t \boldsymbol{A}_2^2 = \begin{bmatrix} 1 - \frac{\beta_t L}{2} & -\frac{\beta_t L}{2} \\ \frac{\beta_t L}{2} & 1 + \frac{\beta_t L}{2} \end{bmatrix}.$$

From this, we notice that the expectation of $\|z_{t+1}\|^2$ conditional on $z_t$ reads

$$\mathbb{E}\left[\|z_{t+1}\|^2 \,\Big|\, z_t\right] = z_t^\top \left(\frac{N_1^\top N_1 + N_2^\top N_2}{2}\right) z_t.$$

Working out the calculations, we can check that

$$\frac{N_1^\top N_1 + N_2^\top N_2}{2} = \begin{bmatrix} 1 + \frac{\beta_t^2 L^2}{2} & 0 \\ 0 & 1 + \frac{\beta_t^2 L^2}{2} \end{bmatrix},$$

thus resulting in

$$\mathbb{E}\left[\|z_{t+1}\|^2 \,\Big|\, z_t\right] = \left(1 + \frac{\beta_t^2 L^2}{2}\right) \|z_t\|^2.$$

Since this holds for all $t \geq 0$, SEG-US diverges in expectation, for any positive stepsizes $\{\alpha_t\}_{t\geq 0}$ and $\{\beta_t\}_{t\geq 0}$. The statement on $\|Fz_t\|$ follows by realizing that $\|Fz\| = \frac{L}{2}\|z\|$. □

**Proposition H.2** (Part of Theorem 4.1). *For $n = 2$, there exists a convex-concave minimax problem $f(x,y) = \frac{1}{2}\sum_{i=1}^2 f_i(x,y)$ having a monotone $F$, consisting of $L$-smooth quadratic $f_i$'s satisfying Assumption 3.4 with $(\rho, \sigma) = (1, 0)$ such that SEG-RR diverges in expectation for any choice of stepsizes $\{\alpha_k\}_{k\geq 0}$ and $\{\beta_k\}_{k\geq 0}$. That is, for any $k \geq 0$,*

$$\mathbb{E}\left[\|z_0^{k+1}\|^2\right] > \mathbb{E}\left[\|z_0^k\|^2\right], \quad \mathbb{E}\left[\|Fz_0^{k+1}\|^2\right] > \mathbb{E}\left[\|Fz_0^k\|^2\right].$$

*Proof.* The proof uses the same example as Proposition H.1, outlined in (121). We show that for this particular worst-case example $f$, SEG-RR diverges in expectation. For $k \geq 0$, the $(k + 1)$-th epoch of SEG-RR starts at $z_0^k$, and the algorithm randomly chooses a permutation $\tau_k : [n] \to [n]$. The algorithm then goes through a series of updates

$$w_i^k = z_i^k - \alpha_k F_{\tau_k(i+1)} z_i^k,$$
$$z_{i+1}^k = z_i^k - \beta_k F_{\tau_k(i+1)} w_i^k,$$

for $i = 0, \ldots, n-1$. In our worst-case example $f$, the updates can be compactly written as

$$z_{i+1}^k = (I - \beta_k A_{\tau_k(i+1)} + \alpha_k \beta_k A_{\tau_k(i+1)}^2) z_i^k.$$

Since we have $n = 2$ and there are only two possible permutations, the updates over an epoch can be summarized as

$$z_0^{k+1} = z_n^k = \begin{cases} (I - \beta_k A_1 + \alpha_k \beta_k A_1^2)(I - \beta_k A_2 + \alpha_k \beta_k A_2^2) z_0^k & \text{with probability } 1/2, \\ (I - \beta_k A_2 + \alpha_k \beta_k A_2^2)(I - \beta_k A_1 + \alpha_k \beta_k A_1^2) z_0^k & \text{with probability } 1/2. \end{cases}$$

By the definition of $A_1$ and $A_2$ and using $A_1^2 = A_2^2 = 0$, we can verify that

$$M_1 := (I - \beta_k A_1 + \alpha_k \beta_k A_1^2)(I - \beta_k A_2 + \alpha_k \beta_k A_2^2) = \begin{bmatrix} 1 - \frac{\beta_k^2 L^2}{2} & -\beta_k L - \frac{\beta_k^2 L^2}{2} \\ \beta_k L - \frac{\beta_k^2 L^2}{2} & 1 - \frac{\beta_k^2 L^2}{2} \end{bmatrix}, \quad (122)$$

$$M_2 := (I - \beta_k A_2 + \alpha_k \beta_k A_2^2)(I - \beta_k A_1 + \alpha_k \beta_k A_1^2) = \begin{bmatrix} 1 - \frac{\beta_k^2 L^2}{2} & -\beta_k L + \frac{\beta_k^2 L^2}{2} \\ \beta_k L + \frac{\beta_k^2 L^2}{2} & 1 - \frac{\beta_k^2 L^2}{2} \end{bmatrix}. \quad (123)$$

From this, we notice that the expectation of $\|z_0^{k+1}\|^2$ conditional on $z_0^k$ reads

$$\mathbb{E}\left[\|z_0^{k+1}\|^2 \,\Big|\, z_0^k\right] = (z_0^k)^\top \left(\frac{M_1^\top M_1 + M_2^\top M_2}{2}\right) z_0^k.$$

Working out the calculations, we can check that

$$\frac{M_1^\top M_1 + M_2^\top M_2}{2} = \begin{bmatrix} 1 + \frac{\beta_k^4 L^4}{2} & 0 \\ 0 & 1 + \frac{\beta_k^4 L^4}{2} \end{bmatrix},$$

thus resulting in

$$\mathbb{E}\left[\|z_0^{k+1}\|^2 \,\Big|\, z_0^k\right] = \left(1 + \frac{\beta_k^4 L^4}{2}\right) \|z_0^k\|^2.$$

Since this holds for all $k \geq 0$, SEG-RR diverges in expectation, for any positive stepsizes $\{\alpha_k\}_{k\geq 0}$ and $\{\beta_k\}_{k\geq 0}$. The statement on $\|Fz_0^k\|$ follows by realizing that $\|Fz\| = \frac{L}{2}\|z\|$. □

**Proposition H.3** (Part of Theorem 4.1). *For $n = 2$, there exists a convex-concave minimax problem $f(x, y) = \frac{1}{2} \sum_{i=1}^{2} f_i(x, y)$ having a monotone $\boldsymbol{F}$, consisting of L-smooth quadratic $f_i$'s satisfying Assumption 3.4 with $(\rho, \sigma) = (1, 0)$ such that SEG-FF diverges in expectation for any positive stepsizes $\{\alpha_k\}_{k \geq 0}$ and $\{\beta_k\}_{k \geq 0}$. That is, for any $k \geq 0$,*

$$\mathbb{E}\left[\left\|\boldsymbol{z}_0^{k+1}\right\|^2\right] > \mathbb{E}\left[\left\|\boldsymbol{z}_0^{k}\right\|^2\right], \quad \mathbb{E}\left[\left\|\boldsymbol{F}\boldsymbol{z}_0^{k+1}\right\|^2\right] > \mathbb{E}\left[\left\|\boldsymbol{F}\boldsymbol{z}_0^{k}\right\|^2\right].$$

*Proof.* The proof uses the same example as Proposition H.1, outlined in (121). We prove that SEG-FF also diverges for this $f$. For $k \geq 0$, the $(k + 1)$-th epoch of SEG-FF starts at $\boldsymbol{z}_0^k$, and the algorithm randomly chooses a permutation $\tau_k : [n] \to [n]$, as in the case of SEG-RR. The algorithm then goes through a series of updates for $i = 0, \ldots, n - 1$:

$$\boldsymbol{w}_i^k = \boldsymbol{z}_i^k - \alpha_k \boldsymbol{F}_{\tau_k(i+1)} \boldsymbol{z}_i^k,$$
$$\boldsymbol{z}_{i+1}^k = \boldsymbol{z}_i^k - \beta_k \boldsymbol{F}_{\tau_k(i+1)} \boldsymbol{w}_i^k,$$

which are the same as SEG-RR; but then, it performs another series of $n$ updates, in the reverse order. For $i = n, \ldots, 2n - 1$,

$$\boldsymbol{w}_i^k = \boldsymbol{z}_i^k - \alpha_k \boldsymbol{F}_{\tau_k(2n-i)} \boldsymbol{z}_i^k,$$
$$\boldsymbol{z}_{i+1}^k = \boldsymbol{z}_i^k - \beta_k \boldsymbol{F}_{\tau_k(2n-i)} \boldsymbol{w}_i^k.$$

Using the definition of $\boldsymbol{M}_1$ and $\boldsymbol{M}_2$ from (122) and (123), one can verify that the $2n = 4$ updates over an epoch of SEG-FF can be summarized as

$$\boldsymbol{z}_0^{k+1} = \boldsymbol{z}_{2n}^k = \begin{cases} \boldsymbol{M}_2 \boldsymbol{M}_1 \boldsymbol{z}_0^k & \text{with probability } 1/2, \\ \boldsymbol{M}_1 \boldsymbol{M}_2 \boldsymbol{z}_0^k & \text{with probability } 1/2. \end{cases}$$

From this, we notice that the expectation of $\left\|\boldsymbol{z}_0^{k+1}\right\|^2$ conditional on $\boldsymbol{z}_0^k$ reads

$$\mathbb{E}\left[\left\|\boldsymbol{z}_0^{k+1}\right\|^2 \,\Big|\, \boldsymbol{z}_0^k\right] = (\boldsymbol{z}_0^k)^\top \left(\frac{\boldsymbol{M}_1^\top \boldsymbol{M}_2^\top \boldsymbol{M}_2 \boldsymbol{M}_1 + \boldsymbol{M}_2^\top \boldsymbol{M}_1^\top \boldsymbol{M}_1 \boldsymbol{M}_2}{2}\right) \boldsymbol{z}_0^k.$$

Working out the calculations, we can check that

$$\frac{\boldsymbol{M}_1^\top \boldsymbol{M}_2^\top \boldsymbol{M}_2 \boldsymbol{M}_1 + \boldsymbol{M}_2^\top \boldsymbol{M}_1^\top \boldsymbol{M}_1 \boldsymbol{M}_2}{2} = \begin{bmatrix} 1 + 2\beta_k^6 L^6 & 0 \\ 0 & 1 + 2\beta_k^6 L^6 \end{bmatrix},$$

thus resulting in

$$\mathbb{E}\left[\left\|\boldsymbol{z}_0^{k+1}\right\|^2 \,\Big|\, \boldsymbol{z}_0^k\right] = \left(1 + 2\beta_k^6 L^6\right) \left\|\boldsymbol{z}_0^k\right\|^2.$$

Since this holds for all $k \geq 0$, SEG-FF diverges in expectation, for any positive stepsizes $\{\alpha_k\}_{k \geq 0}$ and $\{\beta_k\}_{k \geq 0}$. The statement on $\left\|\boldsymbol{F}\boldsymbol{z}_0^k\right\|$ follows by realizing that $\|\boldsymbol{F}\boldsymbol{z}\| = \frac{L}{2} \|\boldsymbol{z}\|$. $\qquad \square$

## H.2 Proof of Limited Convergence of SEG-US in Monotone Cases

In [17, 20], the authors study the same-sample and independent-sample versions of SEG-US, with step sizes $\alpha_t$ and $\beta_t$ satisfying a constant ratio: $\beta_t = \gamma \alpha_t$ for $\gamma \in (0, 1]$. While the authors show convergence in the monotone $\boldsymbol{F}$ case, there is one important limitation shared by the existing analyses. In order to achieve $\min_{t=0,\ldots,T} \mathbb{E}[\|\boldsymbol{F}\boldsymbol{z}_t\|^2] \leq \epsilon^2$ for an arbitrarily chosen $\epsilon$, the algorithms must repeat the same query to the stochastic gradient oracle $b = \mathcal{O}(\frac{1}{\epsilon^2})$ times at every iteration to reduce the gradient variance from $\sigma^2$ to $\frac{\sigma^2}{b}$. In other words, the convergence bounds for SEG-US in the monotone case have an additive term $\mathcal{O}(\sigma^2)$ that cannot be reduced to zero by proper choices of stepsizes. Below, we prove that such a $\sigma^2$ term is in fact inevitable for any choices of stepsizes, if the ratio $\gamma$ is fixed constant. This indicates that SEG-US considered in the existing results can never converge all the way to the optimum if $b = 1$ is maintained throughout training. In contrast, our SEG-FFA shows convergence in the monotone case even when $b = 1$.

**Theorem H.4.** *For $n = 2$, there exists a convex-concave minimax problem $f(x, y) = \frac{1}{2} \sum_{i=1}^{2} f_i(x, y)$ having a monotone $\boldsymbol{F}$, consisting of L-smooth quadratic $f_i$'s satisfying Assumption 3.4 with $(\rho, \sigma) =$*

$(0, \sigma)$ *such that* **SEG-US** *with any positive stepsizes* $\{\alpha_t\}_{t\geq 0}$ *and* $\{\beta_t\}_{t\geq 0}$ *satisfying* $\beta_t = \gamma\alpha_t$ *for* $\gamma > 0$ *cannot converge beyond a certain fixed constant* $\Omega(\sigma^2)$. *More concretely, for any* $t \geq 0$,

$$\mathbb{E}\left[\|\boldsymbol{F}\boldsymbol{z}_t\|^2\right] \geq \min\left\{\|\boldsymbol{F}\boldsymbol{z}_0\|^2, \frac{\gamma\sigma^2}{2}\right\}$$

*regardless of the stepsizes. This holds for both **same-sample** and **independent-sample** SEG-US.*

*Proof.* We consider the case of

$$f_1(x, y) = Lxy + \nu x - \nu y,$$
$$f_2(x, y) = Lxy - \nu x + \nu y,$$

which results in a bilinear (and hence convex-concave) objective function

$$f(x, y) = \frac{1}{2}\sum_{i=1}^{2} f_i(x, y) = Lxy.$$

One can quickly check from the definitions of the component functions $f_1$ and $f_2$ that the corresponding saddle gradient operators are given as

$$\boldsymbol{F}_1\boldsymbol{z} = \underbrace{\begin{bmatrix} 0 & L \\ -L & 0 \end{bmatrix}}_{:=\boldsymbol{A}}\boldsymbol{z} + \nu\boldsymbol{1}, \quad \boldsymbol{F}_2\boldsymbol{z} = \boldsymbol{A}\boldsymbol{z} - \nu\boldsymbol{1}, \quad \boldsymbol{F}\boldsymbol{z} = \boldsymbol{A}\boldsymbol{z},$$

where $\boldsymbol{z} = (x, y) \in \mathbb{R}^2$. From the fact that $\|\boldsymbol{A}\| \leq L$, we can confirm that $f_i$'s are indeed $L$-smooth. As for Assumption 3.4, we can verify that

$$\frac{1}{2}\sum_{i=1}^{2}\|\boldsymbol{F}_i\boldsymbol{z} - \boldsymbol{F}\boldsymbol{z}\|^2 = \frac{1}{2}\sum_{i=1}^{2} 2\nu^2 = 2\nu^2.$$

Therefore, by choosing $\nu^2 = \frac{\sigma^2}{2}$, our example $f$ indeed satisfies Assumption 3.4 with $(\rho, \sigma) = (0, \sigma)$.

The proof is outlined as follows. For the example constructed above, we will calculate the $\mathbb{E}[\|\boldsymbol{z}_{t+1}\|^2]$ and show that the expectation is identical for both same-sample and independent sample versions of **SEG-US**. We will then show that the update on the expected squared distance to equilibrium $\mathbb{E}[\|\boldsymbol{z}_{t+1}\|^2]$ for given $\boldsymbol{z}_t$ can only belong to two categories: either $\|\boldsymbol{z}_t\|^2 \leq \mathbb{E}[\|\boldsymbol{z}_{t+1}\|^2]$ (expected squared distance increases) or $\|\boldsymbol{z}_t\|^2 \geq \mathbb{E}[\|\boldsymbol{z}_{t+1}\|^2] \geq \frac{\gamma\sigma^2}{2L}$ (expected squared distance shrinks but is bounded from below by a constant). Since the two cases hold for any $t \geq 0$ and any choices of $\alpha_t$ and $\beta_t = \gamma\alpha_t$, we show that the "convergence" can happen only up to a neighborhood of equilibrium.

At iteration $t$, **SEG-US** samples component indices $i(t), j(t) \in \{1, 2\}$ for its extrapolation step and update step, respectively. In the independent-sample version $i(t)$ and $j(t)$ are independently sampled from $\text{Unif}(\{1, 2\})$, and in the same-sample version $i(t)$ is sampled uniformly at random and $j(t)$ is set to be equal to $i(t)$. With the indices sampled as above, **SEG-US** then makes an update

$$\boldsymbol{w}_t = \boldsymbol{z}_t - \alpha_t\boldsymbol{F}_{i(t)}\boldsymbol{z}_t,$$
$$\boldsymbol{z}_{t+1} = \boldsymbol{z}_t - \beta_t\boldsymbol{F}_{j(t)}\boldsymbol{w}_t.$$

In our worst-case example $f$, the updates can be written as

$$\boldsymbol{w}_t = \boldsymbol{z}_t - \alpha_t\boldsymbol{A}\boldsymbol{z}_t - s_{i(t)}\alpha_t\nu\boldsymbol{1}$$
$$= (\boldsymbol{I} - \alpha_t\boldsymbol{A})\boldsymbol{z}_t - s_{i(t)}\alpha_t\nu\boldsymbol{1}$$
$$\boldsymbol{z}_{t+1} = \boldsymbol{z}_t - \beta_t\boldsymbol{A}\boldsymbol{w}_t - s_{j(t)}\beta_t\nu\boldsymbol{1}$$
$$= (\boldsymbol{I} - \beta_t\boldsymbol{A} + \alpha_t\beta_t\boldsymbol{A}^2)\boldsymbol{z}_t + s_{i(t)}\alpha_t\beta_t\nu\boldsymbol{A}\boldsymbol{1} - s_{j(t)}\beta_t\nu\boldsymbol{1},$$

where we defined $s_1 = +1$ and $s_2 = -1$ for simplicity of notation.

We now calculate the expected value of $\|\boldsymbol{z}_{t+1}\|^2$.

$$\|\boldsymbol{z}_{t+1}\|^2 = \left\|(\boldsymbol{I} - \beta_t\boldsymbol{A} + \alpha_t\beta_t\boldsymbol{A}^2)\boldsymbol{z}_t\right\|^2 + \alpha_t^2\beta_t^2\nu^2\|\boldsymbol{A}\boldsymbol{1}\|^2 + \beta_t^2\nu^2\|\boldsymbol{1}\|^2$$
$$+ 2s_{i(t)}\alpha_t\beta_t\nu\langle(\boldsymbol{I} - \beta_t\boldsymbol{A} + \alpha_t\beta_t\boldsymbol{A}^2)\boldsymbol{z}_t, \boldsymbol{A}\boldsymbol{1}\rangle - 2s_{j(t)}\beta_t\nu\langle(\boldsymbol{I} - \beta_t\boldsymbol{A} + \alpha_t\beta_t\boldsymbol{A}^2)\boldsymbol{z}_t, \boldsymbol{1}\rangle$$
$$- 2s_{i(t)}s_{j(t)}\alpha_t\beta_t^2\nu^2\langle\boldsymbol{A}\boldsymbol{1}, \boldsymbol{1}\rangle.$$

For the independent-sample case, since $s_{i(t)}$ and $s_{j(t)}$ are independent mean-zero random variables,

$$\mathbb{E}_{i(t),j(t)}[\|z_{t+1}\|^2] = \|(I - \beta_t A + \alpha_t \beta_t A^2)z_t\|^2 + \alpha_t^2 \beta_t^2 \nu^2 \|A1\|^2 + \beta_t^2 \nu^2 \|1\|^2. \quad (124)$$

In the same-sample case, $s_{i(t)} = s_{j(t)}$ is a mean-zero random variable, so

$$\mathbb{E}_{i(t)}[\|z_{t+1}\|^2] = \|(I - \beta_t A + \alpha_t \beta_t A^2)z_t\|^2 + \alpha_t^2 \beta_t^2 \nu^2 \|A1\|^2 + \beta_t^2 \nu^2 \|1\|^2 - 2\alpha_t \beta_t^2 \nu^2 \langle A1, 1 \rangle,$$

but once we realize that $\langle A1, 1 \rangle = 0$, the expectation becomes identical to (124); hence, the rest of the analysis is the same for the two versions.

We now expand and arrange the RHS of (124). It is easy to check that

$$(I - \beta_t A + \alpha_t \beta_t A^2)z_t = \begin{bmatrix} 1 - \alpha_t \beta_t L^2 & -\beta_t L \\ \beta_t L & 1 - \alpha_t \beta_t L^2 \end{bmatrix} \begin{bmatrix} x_t \\ y_t \end{bmatrix} = \begin{bmatrix} (1 - \alpha_t \beta_t L^2)x_t - \beta_t L y_t \\ \beta_t L x_t + (1 - \alpha_t \beta_t L^2)y_t \end{bmatrix}$$

and hence

$$\|(I - \beta_t A + \alpha_t \beta_t A^2)z_t\|^2 = \left((1 - \alpha_t \beta_t L^2)^2 + \beta_t^2 L^2\right)\|z_t\|^2$$
$$= \left(1 - 2\alpha_t \beta_t L^2 + \beta_t^2 L^2(1 + \alpha_t^2 L^2)\right)\|z_t\|^2.$$

From this, we get

$$\mathbb{E}[\|z_{t+1}\|^2] = \|z_t\|^2 - \left(2\alpha_t \beta_t L^2 - \beta_t^2 L^2(1 + \alpha_t^2 L^2)\right)\|z_t\|^2 + 2\alpha_t^2 \beta_t^2 L^2 \nu^2 + 2\beta_t^2 \nu^2$$
$$= \|z_t\|^2 - \left(2\alpha_t \beta_t L^2 - \beta_t^2 L^2(1 + \alpha_t^2 L^2)\right)\|z_t\|^2 + \beta_t^2 \sigma^2(1 + \alpha_t^2 L^2),$$

where we used the choice $\nu_2 = \frac{\sigma^2}{2}$ as above.

The rest of the proof proceeds as follows: we show that, regardless of $t \geq 0$ and the choices of $\alpha_t$ and $\beta_t = \gamma \alpha_t$, the expected value of $\|z_{t+1}\|^2$ given $z_t$ can be categorized into only two cases:

1. $\|z_t\|^2 \leq \mathbb{E}[\|z_{t+1}\|^2]$. That is, the iterate moves away from the equilibrium in expectation.

2. $\|z_t\|^2 \geq \mathbb{E}[\|z_{t+1}\|^2] \geq \frac{\gamma \sigma^2}{2L^2}$. That is, the expected squared distance shrinks but is lower bounded by a certain constant independent of the stepsizes.

Showing this immediately finishes the proof, because there is no way that any $\mathbb{E}[\|z_t\|^2]$ can get smaller than $\min\{\|z_0\|^2, \frac{\gamma \sigma^2}{2L^2}\}$, and $\|Fz\| = L\|z\|$ for our example $f$.

The remaining proof is simple, by noticing that $\|z_t\|^2 \leq \mathbb{E}[\|z_{t+1}\|^2]$ is equivalent to

$$\left(2\alpha_t \beta_t L^2 - \beta_t^2 L^2(1 + \alpha_t^2 L^2)\right)\|z_t\|^2 \leq \beta_t^2 \sigma^2(1 + \alpha_t^2 L^2). \quad (125)$$

Hence, if $\alpha_t$, $\beta_t$, and $z_t$ satisfies (125), we belong to the first category. Otherwise, we are in the second category, for which we need to additionally show $\mathbb{E}[\|z_{t+1}\|^2] \geq \frac{\gamma \sigma^2}{2L^2}$. When the inequality (125) is satisfied with the opposite sign, we must have $2\alpha_t \beta_t L^2 - \beta_t^2 L^2(1 + \alpha_t^2 L^2) > 0$ and

$$\|z_t\|^2 \geq \frac{\beta_t^2 \sigma^2(1 + \alpha_t^2 L^2)}{2\alpha_t \beta_t L^2 - \beta_t^2 L^2(1 + \alpha_t^2 L^2)}.$$

Also, notice that

$$2\alpha_t \beta_t L^2 - \beta_t^2 L^2(1 + \alpha_t^2 L^2) = 1 - \left((1 - \alpha_t \beta_t L^2)^2 + \beta_t^2 L^2\right) < 1.$$

Using $2\alpha_t \beta_t L^2 - \beta_t^2 L^2(1 + \alpha_t^2 L^2) \in (0, 1)$ and substituting the lower bound on $\|z_t\|^2$ into the update equation, we find that

$$\mathbb{E}[\|z_{t+1}\|^2] = \|z_t\|^2 - \left(2\alpha_t \beta_t L^2 - \beta_t^2 L^2(1 + \alpha_t^2 L^2)\right)\|z_t\|^2 + \beta_t^2 \sigma^2(1 + \alpha_t^2 L^2)$$
$$\geq \frac{\beta_t^2 \sigma^2(1 + \alpha_t^2 L^2)}{2\alpha_t \beta_t L^2 - \beta_t^2 L^2(1 + \alpha_t^2 L^2)} = \frac{1}{L^2 \left(\frac{2\alpha_t}{\beta_t \sigma^2(1 + \alpha_t^2 L^2)} - 1\right)}$$

Lastly, substituting $\beta_t = \gamma \alpha_t$ into the RHS gives

$$\mathbb{E}[\|z_{t+1}\|^2] \geq \frac{1}{L^2 \left(\frac{2}{\gamma \sigma^2(1 + \alpha_t^2 L^2)} - 1\right)} \geq \frac{\gamma \sigma^2}{2L^2}.$$

This finishes the proof. □

*Remark* H.5. We remark that, while Theorem H.4 successfully shows that SEG-US as studied in [17, 20] cannot converge to an optimal point unless the batch size is increased every iteration, it does not contradict the (almost sure) convergence result of independent-sample SEG by Hsieh et al. [25]. Indeed, in [25], the stepsizes $\{\alpha_t\}_{t\geq 0}$ and $\{\beta_t\}_{t\geq 0}$ are chosen so that they decay to 0 with a *different* rate and hence the corresponding ratio $\gamma$ approaches 0, while Theorem H.4 considers the case where $\alpha_t$ and $\beta_t$ differ by a constant factor $\gamma$.

## H.3 Proof of SGDA-RR and SEG-RR Lower Bounds

**Theorem H.6.** *Suppose $n \geq 2$ and $L, \mu > 0$ satisfies $L/\mu \geq 2$. There exists a $\mu$-strongly-convex-strongly-concave minimax problem $f(\boldsymbol{z}) = \frac{1}{n}\sum_{i=1}^{n} f_i(\boldsymbol{z})$ consisting of $L$-smooth quadratic $f_i$'s satisfying Assumption 3.4 with $(\rho, \sigma) = (0, \sigma)$ and initialization $\boldsymbol{z}_0^0$ such that SEG-RR with any constant stepsize $\alpha_k = \alpha > 0$, $\beta_k = \beta > 0$ satisfies*

$$
\mathbb{E}\left[\left\|\boldsymbol{z}_0^K - \boldsymbol{z}^*\right\|^2\right] = \begin{cases} \Omega\left(\frac{\sigma^2}{L\mu nK}\right) & \text{if } K \leq L/\mu, \\ \Omega\left(\frac{L\sigma^2}{\mu^3 nK^3}\right) & \text{if } K > L/\mu. \end{cases}
$$

*where $\boldsymbol{z}^*$ is the unique equilibrium point of $f$. For a similar choice of problem $f$ (this time with $(\rho, \sigma) = (1, \sigma)$), SGDA-RR with any constant stepsize $\alpha_k = \alpha > 0$ satisfies*

$$
\mathbb{E}\left[\left\|\boldsymbol{z}_0^K - \boldsymbol{z}^*\right\|^2\right] = \begin{cases} \Omega\left(\frac{\sigma^2}{L\mu nK}\right) & \text{if } K \leq L/\mu, \\ \Omega\left(\frac{\sigma^2}{\mu^2 n^2 K^2} + \frac{L\sigma^2}{\mu^3 nK^3}\right) & \text{if } K > L/\mu. \end{cases}
$$

*Remark* H.7. In Theorem H.6, we adopt techniques from the existing lower bounds for SGD-RR to prove lower bounds for the minimax algorithms SGDA-RR and SEG-RR. In the literature, there are two types of lower bounds for SGD-RR when $K \gtrsim L/\mu$: namely, $\Omega(\frac{1}{n^2K^2} + \frac{1}{nK^3})$ bounds for strongly convex *quadratic* functions [49, 50] and $\Omega(\frac{1}{nK^2})$ bounds for strongly convex *non-quadratic* functions [11, 45, 56]. Upper bounds that match the lower bounds in $n$ and $K$ are also known, indicating that SGD-RR is one of the rare examples of minimization algorithms whose tight convergence rates for quadratic vs. non-quadratic functions differ, within the narrow scope of strongly convex and smooth functions. While it is tempting to aim for a tighter $\Omega(\frac{1}{nK^2})$ lower bound for our algorithms of interest, we note that the existing $\Omega(\frac{1}{nK^2})$ bounds for SGD-RR are proven for piecewise-quadratic functions whose Hessian is *discontinuous*. Since the discontinuous Hessian violates our Assumption 3.3, we instead adhere to the quadratic case to prove lower bounds $\Omega(\frac{1}{nK^3})$ for both SGDA-RR and SEG-RR (when $K \geq L/\mu$). These bounds may not be the tightest possible (since they are restricted to quadratics), but they still suffice to demonstrate that SEG-FFA is provably superior to both SGDA-RR and SEG-RR.

### H.3.1 Existing Lower Bound for SGD-RR

For the proof of lower bounds for SGDA-RR and SEG-RR, we utilize the results and techniques from the lower bounds proven for SGD-RR; thus, it would be profitable to summarize the existing result.

In case of SGD-RR, it is known from Theorem 2 of Safran and Shamir [50] that there exists a minimization problem $g(\boldsymbol{x})$ such that SGD-RR satisfies a lower bound of $\Omega(\frac{1}{n^2K^2} + \frac{1}{nK^3})$ for large enough values of $K$. We rewrite the theorem in a version in accordance with our notation and assumptions:

**Theorem H.8** (Theorem 2 of Safran and Shamir [50]). *For any $n \geq 2$ and $L, \mu > 0$ satisfying $L/\mu \geq 2$, there exists a $\mu$-strongly convex minimization problem $g(\boldsymbol{x}) = \frac{1}{n}\sum_{i=1}^{n} g_i(\boldsymbol{x})$ consisting of $L$-smooth quadratic $g_i$'s satisfying Assumption 3.4 with $(\rho, \sigma) = (1, \sigma)$ such that SGD-RR using any constant stepsize $\alpha_k = \alpha > 0$ satisfies*

$$
\mathbb{E}\left[\left\|\boldsymbol{x}_0^K - \boldsymbol{x}^*\right\|^2\right] = \Omega\left(\frac{\sigma^2}{L\mu nK} \cdot \min\left\{1, \frac{L}{\mu nK} + \frac{L^2}{\mu^2 K^2}\right\}\right).
$$

The statement is equivalent to saying that for SGD-RR with constant stepsize $\alpha > 0$, the bound $\Omega(\frac{\sigma^2}{L\mu nK})$ holds for $K \lesssim L/\mu$ and $\Omega(\frac{\sigma^2}{\mu^2 n^2 K^2} + \frac{L\sigma^2}{\mu^3 nK^3})$ for $K \gtrsim L/\mu$.

The function $g = \frac{1}{n}\sum_{i=1}^{n} g_i$ used in the theorem is defined by the following component functions:

$$g_i(\boldsymbol{x}) = g_i(x_1, x_2, x_3) \coloneqq \frac{\mu}{2}x_1^2 + \frac{L}{2}x_2^2 + \begin{cases} \frac{\sigma}{2}x_2 + \frac{L}{2}x_3^2 + \frac{\sigma}{2}x_3 & i \leq \frac{n}{2}, \\ -\frac{\sigma}{2}x_2 - \frac{\sigma}{2}x_3 & i > \frac{n}{2}, \end{cases} \tag{126}$$

thus making the objective function

$$g(x_1, x_2, x_3) \coloneqq \frac{\mu}{2}x_1^2 + \frac{L}{2}x_2^2 + \frac{L}{4}x_3^2.$$

One can notice that the linear terms in $g_i$ (126) change signs depending on $i \leq \frac{n}{2}$ or not, and handling these sign flips is the key to the proof of the lower bound.

### H.3.2 Proof of Lower Bound for SGDA-RR

For the SGDA-RR lower bound, we consider the following minimax optimization problem:

$$f(\boldsymbol{x}, y) = \frac{1}{n}\sum_{i=1}^{n} f_i(\boldsymbol{x}, y), \text{ where } \boldsymbol{x} \in \mathbb{R}^3, \ y \in \mathbb{R},$$

$$f_i(\boldsymbol{x}, y) = g_i(\boldsymbol{x}) - \frac{\mu}{2}y^2, \tag{127}$$

where $g_i$'s are from (126). We need to first check if the problem instance satisfies the assumptions listed in the theorem statement. Since $f(\boldsymbol{x}, y) = g(\boldsymbol{x}) - \frac{\mu}{2}y^2$ and $g$ is a $\mu$-strongly convex function, $f$ is $\mu$-strongly-convex-strongly-concave as claimed. Also, it is easy to check from the definition of $g_i$ that each component $f_i(\boldsymbol{x}, y)$ is $L$-smooth quadratic.

Lastly, to verify Assumption 3.4, we first define $s_1, \ldots, s_n$ as $s_i = 1$ for $i \leq \frac{n}{2}$ and $s_i = 0$ for $i > \frac{n}{2}$. Using this notation, The function $g_i$ can be compactly written as the following:

$$g_i(x_1, x_2, x_3) = \frac{\mu}{2}x_1^2 + \frac{L}{2}x_2^2 + \frac{\sigma}{2}(2s_i - 1)x_2 + \frac{L}{2}s_ix_3^2 + \frac{\sigma}{2}(2s_i - 1)x_3.$$

Therefore, the saddle gradient operators $\boldsymbol{F}_i$ of $f_i$ and $\boldsymbol{F}$ of $f$ evaluate to

$$\boldsymbol{F}_i\boldsymbol{z} \coloneqq \begin{bmatrix} \nabla g_i(\boldsymbol{x}) \\ \mu y \end{bmatrix} = \begin{bmatrix} \mu x_1 \\ Lx_2 + \frac{\sigma}{2}(2s_i - 1) \\ Ls_ix_3 + \frac{\sigma}{2}(2s_i - 1) \\ \mu y \end{bmatrix}, \quad \boldsymbol{F}\boldsymbol{z} = \begin{bmatrix} \mu x_1 \\ Lx_2 \\ \frac{L}{2}x_3 \\ \mu y \end{bmatrix},$$

which in turn yields

$$\|\boldsymbol{F}_i\boldsymbol{z} - \boldsymbol{F}\boldsymbol{z}\|^2 = \frac{\sigma^2}{4} + \left(\frac{L}{2}x_3 + \frac{\sigma}{2}\right)^2 \leq \left(\frac{L}{2}|x_3| + \sigma\right)^2 \leq (\|\boldsymbol{F}\boldsymbol{z}\| + \sigma)^2$$

for all $i = 1, \ldots, n$. This confirms that the function $f = \frac{1}{n}\sum_i f_i$ satisfies Assumption 3.4 with $(\rho, \sigma) = (1, \sigma)$.

If we run SGDA-RR on this problem, the updates on $\boldsymbol{x}$ done by SGDA-RR is exactly identical to what SGD-RR would perform for the minimization problem $g(\boldsymbol{x}) = \frac{1}{n}\sum_i g_i(\boldsymbol{x})$ with the same choices of random permutations. Therefore, after $K$ epochs of SGDA-RR, it follows from Theorem H.8 that

$$\mathbb{E}\left[\|\boldsymbol{z}_0^K - \boldsymbol{z}^*\|^2\right] \geq \mathbb{E}\left[\|\boldsymbol{x}_0^K - \boldsymbol{x}^*\|^2\right] = \Omega\left(\frac{\sigma^2}{L\mu nK} \cdot \min\left\{1, \frac{L}{\mu nK} + \frac{L^2}{\mu^2 K^2}\right\}\right),$$

which is in fact a tighter lower bound for SGDA-RR than what is stated in Theorem H.6. This finishes the proof.

### H.3.3 Proof of Lower Bound for SEG-RR

In this subsection, we prove the lower bound for SEG-RR. We will first define a new problem instance $f$ to be used here, and verify that the assumptions in the theorem statement are indeed satisfied by this new $f$. We will then spell out the update equation of SEG-RR for this example, which will serve as a basis for the case analysis that follows: we will divide the choices of stepsizes

$\alpha, \beta > 0$ to four regimes and prove a lower bound for each of them. Combining the regimes will result in the desired lower bound.

For SEG-RR, we use a slightly different problem from (127). This time, we consider

$$f(\boldsymbol{x}, y) = \frac{1}{n} \sum_{i=1}^{n} f_i(\boldsymbol{x}, y), \text{ where } \boldsymbol{x} \in \mathbb{R}^2, \ y \in \mathbb{R},$$

$$f_i(\boldsymbol{x}, y) = \frac{L}{2} x_1^2 + \frac{L}{4} x_2^2 + \sigma(2s_i - 1)x_2 - \frac{\mu}{2} y^2,$$

(128)

where $s_i = 1$ for $i \leq \frac{n}{2}$ and $s_i = 0$ for $i > \frac{n}{2}$, as defined above.

We first check if the problem (128) satisfies the assumptions in the theorem statement. Since

$$f(\boldsymbol{x}, y) = \frac{L}{2} x_1^2 + \frac{L}{4} x_2^2 - \frac{\mu}{2} y^2$$

and $L/2 \geq \mu$ by assumption, $f$ is $\mu$-strongly-convex-strongly-concave. Also, it is straightforward to see that each $f_i$ is an $L$-smooth quadratic function. It is left to check Assumption 3.4. The saddle gradient operators $\boldsymbol{F}_i$ of $f_i$ and $\boldsymbol{F}$ of $f$ evaluate to

$$\boldsymbol{F}_i \boldsymbol{z} = \begin{bmatrix} Lx_1 \\ \frac{L}{2}x_2 + \sigma(2s_i - 1) \\ \mu y \end{bmatrix}, \quad \boldsymbol{F}\boldsymbol{z} = \begin{bmatrix} Lx_1 \\ \frac{L}{2}x_2 \\ \mu y \end{bmatrix},$$

which in turn yields

$$\|\boldsymbol{F}_i \boldsymbol{z} - \boldsymbol{F}\boldsymbol{z}\|^2 = \sigma^2,$$

for all $i = 1, \ldots, n$. This confirms that the function $f = \frac{1}{n} \sum_i f_i$ satisfies Assumption 3.4 with $(\rho, \sigma) = (0, \sigma)$, as required by the theorem.

For $k \geq 0$, the $(k+1)$-th epoch of SEG-RR starts at $\boldsymbol{z}_0^k = (\boldsymbol{x}_0^k, y_0^k)$ and the algorithm chooses a random permutation $\tau_k$. The algorithm then goes through a series of updates

$$\boldsymbol{w}_i^k = \boldsymbol{z}_i^k - \alpha \boldsymbol{F}_{\tau_k(i+1)} \boldsymbol{z}_i^k,$$

$$\boldsymbol{z}_{i+1}^k = \boldsymbol{z}_i^k - \beta \boldsymbol{F}_{\tau_k(i+1)} \boldsymbol{w}_i^k,$$

for $i = 0, \ldots, n-1$. For our example $f$ (128), it can be checked that a single iteration by SEG-RR reads

$$\boldsymbol{z}_{i+1}^k = \begin{bmatrix} x_{i+1,1}^k \\ x_{i+1,2}^k \\ y_{i+1}^k \end{bmatrix} = \begin{bmatrix} (1 - \beta L + \alpha\beta L^2)x_{i,1}^k \\ (1 - \frac{\beta L}{2} + \frac{\alpha\beta L^2}{4})x_{i,2}^k - \beta\sigma(1 - \frac{\alpha L}{2})(2s_{\tau_k(i+1)} - 1) \\ (1 - \beta\mu + \alpha\beta\mu^2)y_i^k \end{bmatrix}.$$

Aggregating the SEG-RR updates over an entire epoch ($i = 0, \ldots, n-1$) results in

$$x_{0,1}^{k+1} = (1 - \beta L + \alpha\beta L^2)^n x_{0,1}^k,$$

$$x_{0,2}^{k+1} = \left(1 - \frac{\beta L}{2} + \frac{\alpha\beta L^2}{4}\right)^n x_{0,2}^k - \beta\sigma\left(1 - \frac{\alpha L}{2}\right) \underbrace{\sum_{i=1}^{n} (2s_{\tau_k(i)} - 1)\left(1 - \frac{\beta L}{2} + \frac{\alpha\beta L^2}{4}\right)^{n-i}}_{=: \Phi},$$

$$y_0^{k+1} = (1 - \beta\mu + \alpha\beta\mu^2)^n y_0^k.$$

We will now square both sides of these equations above and take expectations over $\tau_k$. In doing so, there is a useful identity:

$$\mathbb{E}[\Phi] = \sum_{i=1}^{n} \mathbb{E}[2s_{\tau_k(i)} - 1]\left(1 - \frac{\beta L}{2} + \frac{\alpha\beta L^2}{4}\right)^{n-i} = 0.$$

Also, it is worth mentioning that $\tau_k$ is independent of $\boldsymbol{z}_0^k = (x_{0,1}^k, x_{0,2}^k, y_0^k)$. Using these facts, we can arrange the terms to obtain

$$(x_{0,1}^{k+1})^2 = (1 - \beta L + \alpha\beta L^2)^{2n}(x_{0,1}^k)^2,$$

(129)

$$\mathbb{E}[(x_{0,2}^{k+1})^2] = \left(1 - \frac{\beta L}{2} + \frac{\alpha\beta L^2}{4}\right)^{2n} \mathbb{E}[(x_{0,2}^k)^2] + \beta^2\sigma^2\left(1 - \frac{\alpha L}{2}\right)^2 \mathbb{E}[\Phi^2],$$

(130)

$$(y_0^{k+1})^2 = (1 - \beta\mu + \alpha\beta\mu^2)^{2n}(y_0^k)^2.$$

(131)

Based on these three per-epoch update equations above, we now divide the choices of SEG-RR stepsizes $\alpha, \beta > 0$ into the following four cases and handle them separately:

1. $\alpha > \frac{1}{L}$, in which case we show that SEG-RR makes $(x_{0,1}^{k+1})^2 > (x_{0,1}^k)^2$ hold deterministically, so that if we initialize at $x_{0,1}^0 = \frac{\sigma}{\sqrt{L\mu}}$ then we have

$$\mathbb{E}\left[\left\|z_0^K\right\|^2\right] \geq (x_{0,1}^K)^2 > (x_{0,1}^0)^2 = \frac{\sigma^2}{L\mu}.$$

2. $\alpha \leq \frac{1}{L}$ and $\beta \leq \frac{1}{\mu n K}$, in which case we show that SEG-RR initialized at $y_0^0 = \frac{\sigma}{\sqrt{L\mu}}$ suffers

$$\mathbb{E}\left[\left\|z_0^K\right\|^2\right] = \Omega\left(\frac{\sigma^2}{L\mu}\right),$$

3. $\alpha \leq \frac{1}{L}$ and $\frac{1}{\mu n K} < \beta < \frac{1}{nL}$, in which case we show that SEG-RR initialized at $x_{0,2}^0 = 0$ suffers

$$\mathbb{E}\left[\left\|z_0^K\right\|^2\right] = \Omega\left(\frac{L\sigma^2}{\mu^3 n K^3}\right),$$

4. $\alpha \leq \frac{1}{L}, \beta > \frac{1}{\mu n K}$, and $\beta \geq \frac{1}{nL}$ in which case we show that SEG-RR initialized at $x_{0,2}^0 = 0$ suffers

$$\mathbb{E}\left[\left\|z_0^K\right\|^2\right] = \Omega\left(\frac{\sigma^2}{L\mu n K}\right).$$

Notice that the third case $\frac{1}{\mu n K} < \beta < \frac{1}{nL}$ only makes sense when $K > L/\mu$; otherwise, the third case just disappears. Hence, for the "large epoch" regime where $K > L/\mu$, the third case achieves the minimum error possible, so it holds that

$$\mathbb{E}\left[\left\|z_0^K\right\|^2\right] = \Omega\left(\frac{L\sigma^2}{\mu^3 n K^3}\right).$$

For the "small epoch" regime ($K \leq L/\mu$), the third case does not exist and the fourth case achieves the minimum, so

$$\mathbb{E}\left[\left\|z_0^K\right\|^2\right] = \Omega\left(\frac{\sigma^2}{L\mu n K}\right).$$

Combining the two cases yields the desired lower bound in the theorem statement. It now remains to carry out the case analysis.

**Case 1: $\alpha > \frac{1}{L}$.**   For this case, we use (129) to prove divergence. Notice from $\alpha > \frac{1}{L}$ that

$$1 - \beta L + \alpha\beta L^2 = 1 + \beta L(\alpha L - 1) > 1,$$

regardless of $\beta > 0$. Hence, from (129), we get

$$\mathbb{E}\left[\left\|z_0^K\right\|^2\right] \geq (x_{0,1}^K)^2 > (x_{0,1}^0)^2.$$

If we initialize at $x_{0,1}^0 = \frac{\sigma}{\sqrt{L\mu}}$, then this proves

$$\mathbb{E}\left[\left\|z_0^K\right\|^2\right] \geq \frac{\sigma^2}{L\mu}.$$

**Case 2: $\alpha \leq \frac{1}{L}$ and $\beta \leq \frac{1}{\mu n K}$.**   For this case, we employ (131) to show that the "contraction rate" is too small to make enough "progress." Notice from our stepsizes that

$$1 - \beta\mu + \alpha\beta\mu^2 \geq 1 - \beta\mu \geq 1 - \frac{1}{nK} \geq 0.$$

Applying this inequality to (131), we have

$$(y_0^{k+1})^2 \geq \left(1 - \frac{1}{nK}\right)^{2n} (y_0^k)^2,$$

which in turn means that the progress over $K$ epoch is bounded from below by

$$(y_0^K)^2 \geq \left(1 - \frac{1}{nK}\right)^{2nK} (y_0^0)^2 \geq \frac{(y_0^0)^2}{16},$$

where we used our assumption that $n \geq 2$ and $K \geq 1$. Hence, if our initialization was given as $y_0^0 = \frac{\sigma}{\sqrt{L\mu}}$, then this proves

$$\mathbb{E}\left[\|\mathbf{z}_0^K\|^2\right] \geq (y_0^K)^2 \geq \frac{(y_0^0)^2}{16} = \Omega\left(\frac{\sigma^2}{L\mu}\right).$$

**Case 3:** $\alpha \leq \frac{1}{L}$ **and** $\frac{1}{\mu nK} < \beta < \frac{1}{nL}$. For stepsizes in this interval, we use (130) to derive the desired bound. Here, it is important to characterize a lower bound on the quantity

$$\mathbb{E}[\Phi^2] := \mathbb{E}\left[\left(\sum_{i=1}^{n} (2s_{\tau_k(i)} - 1)\left(1 - \frac{\beta L}{2} + \frac{\alpha\beta L^2}{4}\right)^{n-i}\right)^2\right].$$

To this end, we can use a lemma from Safran and Shamir [49], stated below:

**Lemma H.9** (Lemma 1 of Safran and Shamir [49]). *Let* $\pi_1, \ldots, \pi_n$ *(for even $n$) be a random permutation of* $(1, 1, \ldots, 1, -1, -1, \ldots, -1)$ *where both $1$ and $-1$ appear exactly $n/2$ times. Then there is a numerical constant $c > 0$ such that for any $\nu > 0$,*

$$\mathbb{E}\left[\left(\sum_{i=1}^{n} \pi_i(1-\nu)^{n-i}\right)^2\right] \geq c \cdot \min\left\{1 + \frac{1}{\nu}, n^3\nu^2\right\}.$$

One can notice that Lemma H.9 is directly applicable to $\mathbb{E}[\Phi^2]$, with $\nu \leftarrow \frac{\beta L}{2} - \frac{\alpha\beta L^2}{4}$. Since

$$\nu = \frac{\beta L}{2} - \frac{\alpha\beta L^2}{4} \leq \frac{\beta L}{2} \leq \frac{1}{2n},$$

we have $n^3\nu^2 \leq \frac{1}{8\nu}$, thereby

$$\min\left\{1 + \frac{1}{\nu}, n^3\nu^2\right\} \geq \min\left\{\frac{1}{\nu}, n^3\nu^2\right\} = n^3\nu^2.$$

Therefore, Lemma H.9 gives

$$\mathbb{E}[\Phi^2] \geq cn^3\left(\frac{\beta L}{2} - \frac{\alpha\beta L^2}{4}\right)^2 = \frac{c\beta^2 n^3 L^2}{4}\left(1 - \frac{\alpha L}{2}\right)^2 \geq \frac{c\beta^2 n^3 L^2}{16}, \tag{132}$$

where the last inequality used $\alpha \leq \frac{1}{L}$. Applying (132) to (130) and also using $(1 - \frac{\alpha L}{2})^2 \geq \frac{1}{4}$,

$$\mathbb{E}[(x_{0,2}^{k+1})^2] \geq \left(1 - \frac{\beta L}{2} + \frac{\alpha\beta L^2}{4}\right)^{2n} \mathbb{E}[(x_{0,2}^k)^2] + \frac{c\beta^4 n^3 L^2 \sigma^2}{64}.$$

Unrolling the inequality for $k = 0, \ldots, K-1$ gives

$$\mathbb{E}\left[(x_{0,2}^K)^2\right] \geq \left(1 - \frac{\beta L}{2} + \frac{\alpha\beta L^2}{4}\right)^{2nK} (x_{0,2}^0)^2 + \frac{c\beta^4 n^3 L^2 \sigma^2}{64} \sum_{j=0}^{K-1}\left(1 - \frac{\beta L}{2} + \frac{\alpha\beta L^2}{4}\right)^{2nj}$$

$$= \left(1 - \frac{\beta L}{2} + \frac{\alpha\beta L^2}{4}\right)^{2nK} (x_{0,2}^0)^2 + \frac{c\beta^4 n^3 L^2 \sigma^2}{64} \cdot \frac{1 - \left(1 - \frac{\beta L}{2} + \frac{\alpha\beta L^2}{4}\right)^{2nK}}{1 - \left(1 - \frac{\beta L}{2} + \frac{\alpha\beta L^2}{4}\right)^{2n}}.$$

Now note that our initialization $x_{0,2}^0$ can be set to zero, which eliminates the need to think about the first term in the RHS. It is now left to bound the second term. First, by the stepsize range $\alpha \leq \frac{1}{L}$, $\beta > \frac{1}{\mu nK}$ and our assumption $L/\mu \geq 2$, we have

$$\left(1 - \frac{\beta L}{2} + \frac{\alpha\beta L^2}{4}\right)^{2nK} \leq \left(1 - \frac{\beta L}{4}\right)^{2nK} \leq \left(1 - \frac{L}{4\mu nK}\right)^{2nK} \leq e^{-\frac{L}{2\mu}} \leq e^{-1}.$$

Next, by Bernoulli's inequality

$$\left(1 - \frac{\beta L}{2} + \frac{\alpha\beta L^2}{4}\right)^{2n} \geq \left(1 - \frac{\beta L}{2}\right)^{2n} \geq 1 - \beta n L > 0.$$

Plugging in the two inequalities to above, we obtain

$$\mathbb{E}\left[(x_{0,2}^K)^2\right] \geq \frac{c\beta^4 n^3 L^2 \sigma^2}{64} \cdot \frac{1 - \left(1 - \frac{\beta L}{2} + \frac{\alpha\beta L^2}{4}\right)^{2nK}}{1 - \left(1 - \frac{\beta L}{2} + \frac{\alpha\beta L^2}{4}\right)^{2n}}$$

$$\geq \frac{c\beta^4 n^3 L^2 \sigma^2}{64} \cdot \frac{1 - e^{-1}}{1 - (1 - \beta n L)} = c'\beta^3 n^2 L\sigma^2$$

for a numerical constant $c' > 0$. Plugging in the lower bound $\beta > \frac{1}{\mu n K}$ yields

$$\mathbb{E}\left[\left\|z_0^K\right\|^2\right] \geq \mathbb{E}\left[(x_{0,2}^K)^2\right] = \Omega\left(\frac{L\sigma^2}{\mu^3 n K^3}\right).$$

**Case 4:** $\alpha \leq \frac{1}{L}$, $\beta > \frac{1}{\mu n K}$, **and** $\beta \geq \frac{1}{nL}$. We again use (130). By noticing that the initialization $x_{0,2}^0 = 0$, we can unroll (130) for $k = 0, \ldots, K-1$ to get

$$\mathbb{E}\left[(x_{0,2}^K)^2\right] \geq \frac{\beta^2 \sigma^2}{4}\mathbb{E}[\Phi^2] \sum_{j=0}^{K-1} \left(1 - \frac{\beta L}{2} + \frac{\alpha\beta L^2}{4}\right)^{2nj} \geq \frac{\beta^2 \sigma^2}{4}\mathbb{E}[\Phi^2], \tag{133}$$

where the last inequality holds regardless of $\beta$ because each summand with $j \geq 1$ is nonnegative. We then invoke Lemma H.9 to lower bound $\mathbb{E}[\Phi^2]$, again with $\nu \leftarrow \frac{\beta L}{2} - \frac{\alpha\beta L^2}{4}$. Since

$$\nu = \frac{\beta L}{2} - \frac{\alpha\beta L^2}{4} \geq \frac{\beta L}{4} \geq \frac{1}{4n},$$

we have $n^3\nu^2 \geq \frac{1}{64\nu}$, thereby

$$\min\left\{1 + \frac{1}{\nu}, n^3\nu^2\right\} \geq \min\left\{\frac{1}{\nu}, n^3\nu^2\right\} \geq \frac{1}{64\nu}.$$

Therefore, Lemma H.9 gives

$$\mathbb{E}[\Phi^2] \geq \frac{c}{64\nu} = \frac{c}{32\beta L} \cdot \frac{1}{1 - \frac{\alpha L}{2}} \geq \frac{c}{32\beta L}. \tag{134}$$

Combining (134) with (133) gives

$$\mathbb{E}\left[(x_{0,2}^K)^2\right] \geq \frac{c\beta\sigma^2}{128L} = \Omega\left(\frac{\sigma^2}{L\mu n K}\right),$$

where the last step used $\beta > \frac{1}{\mu n K}$. This finishes the case analysis, hence the proof of Theorem H.6.

# I   Additional Experiments

To evaluate our algorithm SEG-FFA as well as other baseline algorithms, we conduct numerical experiments on monotone and strongly monotone problems. Specifically, as we have mentioned in Section 6, we consider random quadratic problems of the form

$$\min_{\boldsymbol{x}\in\mathbb{R}^{d_x}} \max_{\boldsymbol{y}\in\mathbb{R}^{d_y}} \frac{1}{n}\sum_{i=1}^{n} \begin{bmatrix}\boldsymbol{x}\\\boldsymbol{y}\end{bmatrix}^\top \begin{bmatrix}\boldsymbol{A}_i & \boldsymbol{B}_i\\\boldsymbol{B}_i^\top & -\boldsymbol{C}_i\end{bmatrix} \begin{bmatrix}\boldsymbol{x}\\\boldsymbol{y}\end{bmatrix} - \boldsymbol{t}_i^\top \begin{bmatrix}\boldsymbol{x}\\\boldsymbol{y}\end{bmatrix}.$$

We choose $d_x = d_y = 20$ and $n = 40$ for all the experiments. Numerical computations are done using NumPy [24] and SciPy [52], and the plots are drawn using Matplotlib [26].

### I.1 Problem Constructions for Experiments in Section 6

For an experiment for the monotone case, the random components are sampled as follows. We choose $\boldsymbol{B}_i$ so that each element is an i.i.d. sample from a uniform distribution over the interval $[0, 1]$, and $\boldsymbol{t}_i$ so that each element is an i.i.d. sample from a standard normal distribution. We chose $\boldsymbol{A}_i$ to be diagonal matrices in the following procedure: for each $j = 1, \ldots, 20$ we randomly chose a subset $\mathcal{I}_j$ of $\frac{n}{2} = 20$ indices from $[n] = \{1, \ldots, 40\}$, and set the $(j, j)$-entry of $\boldsymbol{A}_i$ to be

$$(\boldsymbol{A}_i)_{j,j} = \begin{cases} 2 & \text{if } i \in \mathcal{I}_j \\ -2 & \text{otherwise} \end{cases}.$$

We repeat the exact same procedure for $\boldsymbol{C}_i$ as well. Notice that $\sum_{i=1}^n \boldsymbol{A}_i = \sum_{i=1}^n \boldsymbol{C}_i = \boldsymbol{0}$ by design. Hence, each of the component functions will be a nonconvex-nonconcave quadratic function in general, but the objective function itself becomes a convex-concave function.

For the experiment in the strongly monotone case, we sample $\boldsymbol{B}_i$ and $\boldsymbol{t}_i$ in the same way as in the monotone case, but we use different choices of $\boldsymbol{A}_i$ and $\boldsymbol{C}_i$ to ensure the objective function to be strongly-convex-strongly-concave. In particular, for each $i = 1, \ldots, n$, we sample $\boldsymbol{A}_i$ by computing $\boldsymbol{A}_i = \boldsymbol{Q}_i \boldsymbol{D}_i \boldsymbol{Q}_i^\top$, where $\boldsymbol{D}_i$ is a random diagonal matrix whose diagonal entries are i.i.d. samples from a uniform distribution over the interval $[\frac{1}{2}, 1]$, and $\boldsymbol{Q}_i$ is a random orthogonal matrix obtained by computing a *QR* decomposition of a $20 \times 20$ random matrix whose elements are i.i.d. samples from a standard normal distribution. We sample $\boldsymbol{C}_i$ by the exact same method.

### I.2 Monotone Case & Ablation Study on the Anchoring Step

In Section 6, we compared the empirical performance of various SEGs, namely SEG-FFA, SEG-FF, SEG-RR, and SEG-US. Here, as an ablation study on the anchoring technique, we additionally compare *SEG-RRA* and *SEG-USA*, which are each SEG-RR and SEG-US with an additional anchoring step, respectively. For these two methods, we take the anchoring step after every $n$ iterations. We ran those methods on the same 5 random instances used in Section 6. For both SEG-RRA and SEG-USA, we ran the method with two different stepsize choices, namely $\alpha_k = \beta_k = \eta_k$ (inspired by the stepsize used in deterministic EG) and $\alpha_k = \beta_k/2 = \eta_k/2$ (the stepsize used for SEG-FFA) where we again set $\eta_k = \eta_0/(1+k/10)^{0.34}$ with $\eta_0 = \min\{0.01, \frac{1}{L}\}$.

The results are plotted in Figure 2. As SEG-RRA and SEG-USA are designed to take one pass per epoch, for those methods, we compute the ratio $\frac{\|\boldsymbol{F}\boldsymbol{z}_0^t\|^2}{\|\boldsymbol{F}\boldsymbol{z}_0^0\|^2}$ where $t$ denotes the number of passes, and plot the geometric mean over the 5 runs.

From the performance of SEG-RRA with $\alpha_k = \beta_k$ and the two variants of SEG-USA, it is possible to observe that adding the anchoring step does improve the performance of the method up to a certain level, but it alone does not fully resolve the nonconvergence issue. On the other hand, quite interestingly, SEG-RRA with $\alpha_k = \beta_k/2$ shows a hint of convergence. While its performance is slightly worse compared to SEG-FFA, it is nonetheless still notable as it is the only other method from SEG-FFA that seems to be capable of converging to an optimum.

We conjecture that this intriguing performance of SEG-RRA with $\alpha_k = \beta_k/2$ is because it achieves an *"expected"* second order matching to the (deterministic) EG. Indeed, following the notations of Proposition D.1, one can deduce from Proposition D.1 that using SEG-RRA with $\alpha = \beta/2$ will result in an epoch-level update of

$$\boldsymbol{z}^\sharp = \boldsymbol{z}_0 - \frac{\beta}{2}\sum_{j=0}^{n-1} \boldsymbol{T}_j \boldsymbol{z}_0 + \frac{\beta^2}{4}\sum_{j=0}^{n-1} D\boldsymbol{T}_j(\boldsymbol{z}_0)\boldsymbol{T}_j \boldsymbol{z}_0 + \frac{\beta^2}{2}\sum_{0 \le i < j \le n-1} D\boldsymbol{T}_j(\boldsymbol{z}_0)\boldsymbol{T}_i \boldsymbol{z}_0 + \frac{\boldsymbol{\epsilon}_n}{2} \quad (135)$$

with $\boldsymbol{\epsilon}_n = o(\beta^2)$. Here, notice that $(\boldsymbol{T}_0, \boldsymbol{T}_1, \ldots, \boldsymbol{T}_{n-1}) = (\boldsymbol{F}_{\tau(1)}, \boldsymbol{F}_{\tau(2)}, \ldots, \boldsymbol{F}_{\tau(n)})$ for some randomly chosen permutation $\tau \in \mathcal{S}_n$. Now, observe that for any two distinct $i, j \in [n]$, there are exactly $\frac{n!}{2}$ permutations in $\mathcal{S}_n$ such that $i$ comes before $j$ in the sequence $\tau(1), \tau(2), \ldots, \tau(n)$, and also exactly $\frac{n!}{2}$ permutations such that $j$ comes before $i$. Thus, in taking the expectation over the

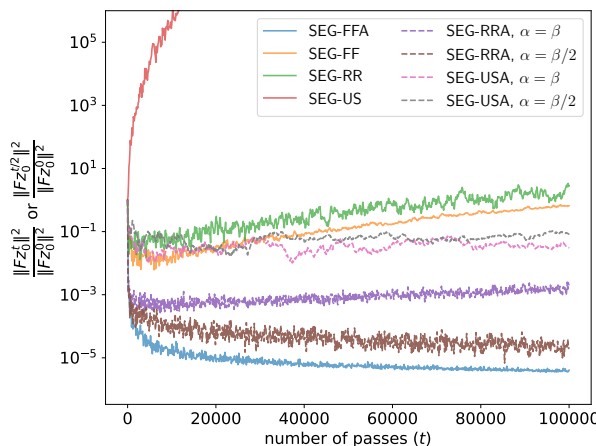

Figure 2: Experimental results in the monotone example, comparing the performance of SEG-RRA and SEG-USA with the results displayed in Figure 1. Because SEG-FFA and SEG-FF use two passes per epoch, for those two methods, we plot $\|\boldsymbol{F}\boldsymbol{z}_0^{t/2}\|^2/\|\boldsymbol{F}\boldsymbol{z}_0^0\|^2$.

randomness of choosing the permutation $\tau$, we get

$$
\mathbb{E}_\tau \left[ \sum_{0 \le i < j \le n-1} D\boldsymbol{T}_j(\boldsymbol{z}_0)\boldsymbol{T}_i\boldsymbol{z}_0 \right] = \mathbb{E}_\tau \left[ \sum_{1 \le i < j \le n} D\boldsymbol{F}_{\tau(j)}(\boldsymbol{z}_0)\boldsymbol{F}_{\tau(i)}\boldsymbol{z}_0 \right]
$$

$$
= \frac{1}{n!} \sum_{\tau \in \mathcal{S}_n} \sum_{1 \le i < j \le n} D\boldsymbol{F}_{\tau(j)}(\boldsymbol{z}_0)\boldsymbol{F}_{\tau(i)}\boldsymbol{z}_0
$$

$$
= \frac{1}{2} \sum_{i \ne j} D\boldsymbol{F}_j(\boldsymbol{z}_0)\boldsymbol{F}_i\boldsymbol{z}_0,
$$

where in getting the third line we have used the previously made observation that for any fixed $i$ and $j$ with $i \ne j$, the term $D\boldsymbol{F}_j(\boldsymbol{z}_0)\boldsymbol{F}_i\boldsymbol{z}_0$ appears exactly $\frac{n!}{2}$ times in the sum on the second line. Hence, taking the expectation with respect to the random permutation on (135) we get

$$
\mathbb{E}_\tau \left[ \boldsymbol{z}^\sharp \right] = \boldsymbol{z}_0 - \frac{n\beta}{2}\boldsymbol{F}\boldsymbol{z}_0 + \frac{\beta^2}{4}\sum_{j=1}^n D\boldsymbol{F}_j(\boldsymbol{z}_0)\boldsymbol{F}_j\boldsymbol{z}_0 + \frac{\beta^2}{4}\sum_{i \ne j} D\boldsymbol{F}_j(\boldsymbol{z}_0)\boldsymbol{F}_i\boldsymbol{z}_0 + \frac{1}{2}\mathbb{E}_\tau\left[\boldsymbol{\epsilon}_n\right]
$$

$$
= \boldsymbol{z}_0 - \frac{n\beta}{2}\boldsymbol{F}\boldsymbol{z}_0 + \frac{\beta^2}{4}\sum_{j=1}^n\sum_{i=1}^n D\boldsymbol{F}_j(\boldsymbol{z}_0)\boldsymbol{F}_i\boldsymbol{z}_0 + \frac{1}{2}\mathbb{E}_\tau\left[\boldsymbol{\epsilon}_n\right]
$$

$$
= \boldsymbol{z}_0 - \frac{n\beta}{2}\boldsymbol{F}\boldsymbol{z}_0 + \frac{n^2\beta^2}{4}D\boldsymbol{F}(\boldsymbol{z}_0)\boldsymbol{F}\boldsymbol{z}_0 + \frac{1}{2}\mathbb{E}_\tau\left[\boldsymbol{\epsilon}_n\right].
$$

Comparing this to (7) when $\eta_1 = \eta_2 = {}^{n\beta}/_2$, we indeed see that the update rule of SEG-RRA with $\alpha = {}^\beta/_2$ achieves a second-order matching *on expectation* to the (deterministic) EG update with stepsize ${}^{n\beta}/_2$.

We also conjecture that the relatively worse performance of SEG-RRA with $\alpha = {}^\beta/_2$ compared to SEG-FFA is because the error over an epoch is $O(\eta^3)$ only on expectation, and thus the actual error occurring in each epoch can be larger than $O(\eta^3)$. Unfortunately, our convergence analysis on SEG-FFA relies on the error over an epoch being $O(\eta^3)$ *deterministically* (*cf.* Proposition 5.3), hence cannot be directly applied to SEG-RRA with $\alpha = {}^\beta/_2$. We leave the search for a theoretical explanation on this alluring performance of SEG-RRA with $\alpha = {}^\beta/_2$ as a stimulating direction for future work.

### I.3 Monotone Case: Comparison with Hsieh et al. [25]

Let us also compare the performance of SEG-FFA with the *independent-sample* double stepsize SEG (*DSEG*) by Hsieh et al. [25]. Writing in terms of the finite-sum structure, the update rule of DSEG

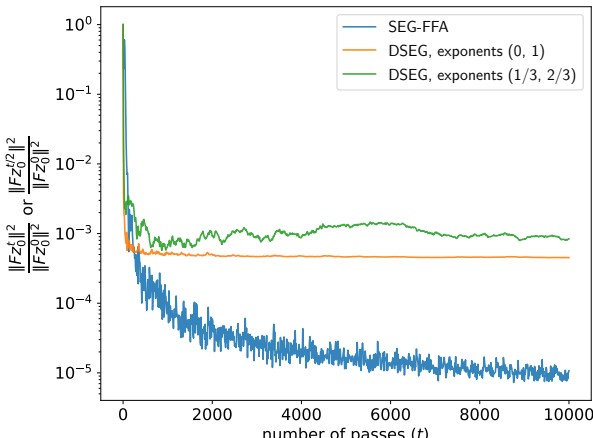

Figure 3: Experimental results in the monotone example, comparing SEG-FFA and the methods proposed by Hsieh et al. [25]. By the same reason as in Figure 2, we plot $\|Fz_0^{t/2}\|^2/\|Fz_0^0\|^2$ for SEG-FFA only.

can be written as

$$w^k \leftarrow z^k - \eta_{1,k} F_{i(1,k)} z^k$$
$$z^{k+1} \leftarrow z^k - \eta_{2,k} F_{i(2,k)} w^k$$

where $i(1,k)$ and $i(2,k)$ are random indices that are independently drawn from $[n]$ for each $k$. The stepsizes are chosen in the form of $\eta_{1,k} = \Theta(1/k^{r_1})$ and $\eta_{2,k} = \Theta(1/k^{r_2})$, where setting $r_1 \leq r_2$ is the key point of DSEG. Two choices of the exponent pair $(r_1, r_2)$ proposed in [25] are $(1/3, 2/3)$ for general monotone problems and $(0, 1)$ exclusively for the case when $F$ is affine.

We again use the same component functions as in the previous experiment. The setup for running SEG-FFA are kept the same. For DSEG, we use the default choices suggested by Hsieh et al. [25], namely $\eta_{1,k} = \gamma_0/(k+19)^{r_1}$ and $\eta_{2,k} = \eta_0/(k+19)^{r_2}$, where $(\gamma_0, \eta_0) = (1, 0.1)$ for the bilinear case with $(r_1, r_2) = (0, 1)$ and $(\gamma_0, \eta_0) = (0.1, 0.05)$ for the general case with $(r_1, r_2) = (1/3, 2/3)$.

The results are displayed in Figure 3, where the details on how the plots are drawn are the same as Figure 2. Here we can clearly see that SEG-FFA outperforms both versions of DSEG.

### I.4 Strongly Monotone Case Again, with Various Stepsizes

We also ran the experiment on strongly monotone problems described in Section 6, but with changing the stepsizes. We tested six different values of $\eta_k$; we have tested with $\eta_k = a \times 10^b$ where $a \in \{1, 2, 5\}$ and $b \in \{-4, -3\}$. Notice that the case $\eta_k = 10^{-3}$ is exactly the experiment conducted in Section 6.

The results are plotted in Figure 4. The overall details are the same as described in Section 6, as the only difference is the stepsize choice. We can observe that, while the initial speed of convergence may not be the fastest depending on the stepsize, SEG-FFA is always the method that eventually finds the point with the smallest gradient. In other words, as predicted by our theoretical analyses, the supremacy of SEG-FFA is in general not affected by the choice of the stepsize, as long as the chosen stepsize is reasonably small.

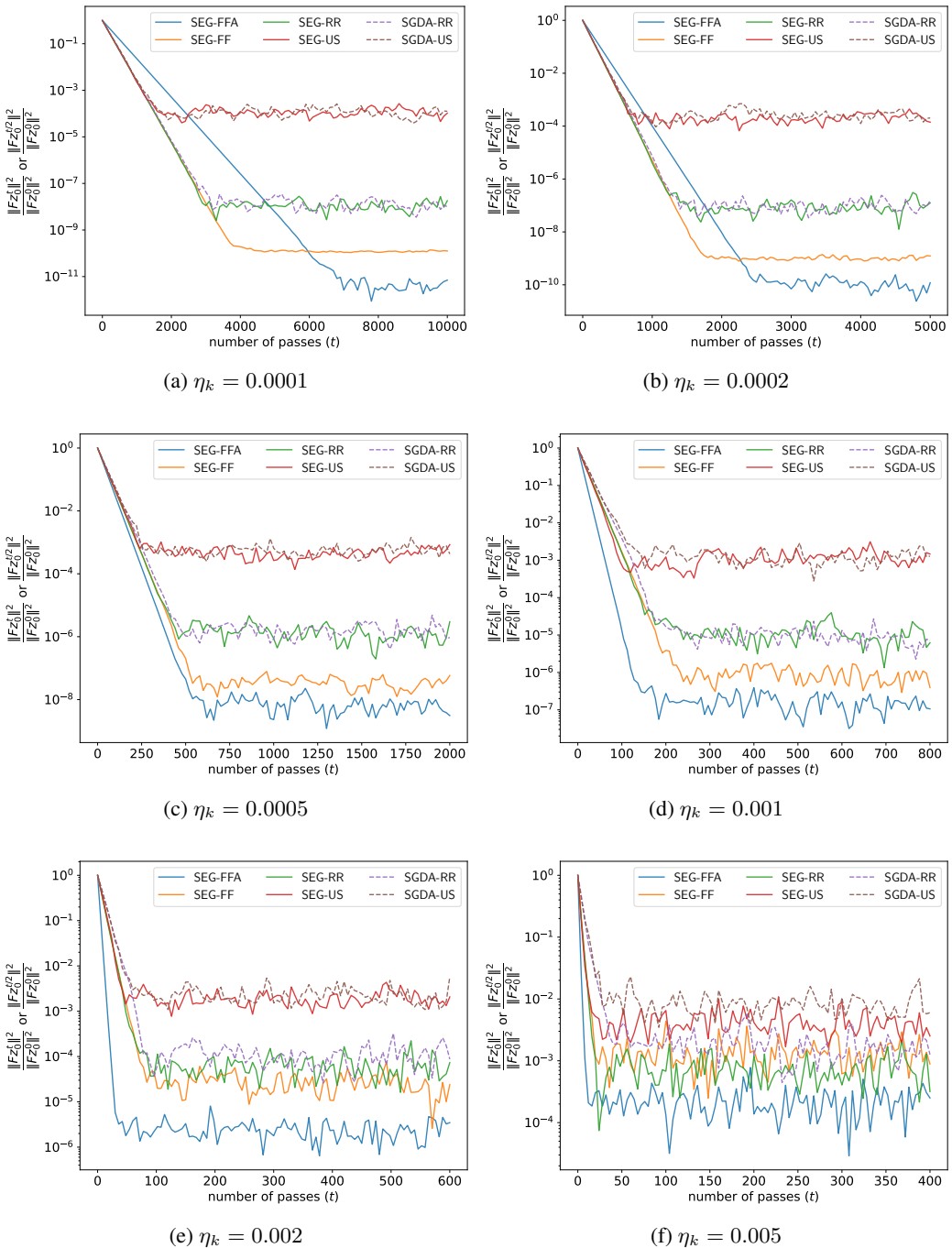

Figure 4: Experimental results on the strongly monotone problems with different stepsizes. Notice that Figure 4d is exactly the plot that is included in Section 6. The only difference between the experiments conducted is the choice of the stepsize.

