# OpenReview forum: "Stochastic Extragradient with Flip-Flop Shuffling & Anchoring: Provable Improvements"
_NeurIPS.cc/2024/Conference — NeurIPS 2024 poster_

### Official Review · Reviewer_k7BS · 2024-07-09

**Soundness:** 3
**Presentation:** 2
**Contribution:** 3
**Rating:** 6
**Confidence:** 4

**Summary:**

The paper studies unconstrained (strongly)-monotone finite sum minimization problems. They introduce a new scheme called SEG-FFA which:

i) for a given epoch runs stochastic extragradient (SEG) with possibly two different stepsizes
ii) uses flip-flop shuffling per epoch
iii) uses the average of the epoch initialization and the last iterate as the initialization for the next epoch

With $K$ being the number of epochs and $n$ being the number of finite sum elements, they show:

- For the monotone case a $\tilde{\mathcal O}(1/K^{1/3})$ rate for the squared norm of the operator.
- For the $\mu$-strongly-monotone case a $\mathcal O(\exp{(-\mu K^\epsilon)\Vert z^0_0 - z^\star \lVert^2} + \tfrac{1}{nK^{4-5\epsilon}})$ rate where $\epsilon \in (0,2/3)$.

**Strengths:**

The paper reads very well in terms of laying out what problem is being addressed and the current landscape of the literature (including the extended discussion in the appendix). The reasoning behind their algorithmic construction is also very clear.

**Weaknesses:**

My main concern is with the strength of the results:

- Regarding the monotone case (Thm. 5.4): The argument seems to be that one _full epoch_ of SEG-FFA can approximate one step of the deterministic EG accurately. In other words, the rate has no dependency on $n$. Why not instead simply run full batch EG which would get the much faster $\mathcal O(1/K)$ rate? (e.g. using gradient accumulation to make it memory efficient)
- Regarding the strongly-monotone case (Thm. 5.5): There is a tradeoff in the rate. If we want fast rate for the stochastic term $\mathcal O(\tfrac{1}{nK^{4-5\epsilon}})$ we need to take $\varepsilon \rightarrow 0$ in which case the linear rate $\mathcal O(\exp{(-\mu K^\epsilon)\Vert z^0_0 - z^\star \lVert^2})=\mathcal O(\exp{(-\mu)\Vert z^0_0 - z^\star \lVert^2})$ cannot be made small. Even if we ask to just match the lowerbound $\mathcal O(\tfrac{1}{nK^{3}})$ on SGDA-RR and SEG-RR we run into trouble.  Pick $\epsilon=1/5$ to get $\mathcal O(\exp{(-\mu K^{1/5})\Vert z^0_0 - z^\star \lVert^2} + \tfrac{1}{nK^{3}})$. If we plot the two terms (ignoring the constants) we will see that the linear rate dominates the polynomial term for even very large $K$ (100M+). In other words, it is unfair to compare only the polynomial terms in this case, and the benefit of SEG-FFA over e.g.  SEG-RR is much less clear when the linear term is taken into account.

Other concerns:

- Regarding Thm. 4.1: It seems surprising that it is possible to show divergence for e.g. SEG-US (with two stepsizes) since this is the scheme that [24] shows converges almost surely. How is it possible for these two results to coexist?
- Currently the rates do not state dependencies on noise parameters. It is good practice to include $\rho$ and $\sigma$ in the rates as we should expect different dependencies.
- Table 1 currently does not include the dependency on the initialization in the rate for SEG-FFA, which can be misleading. I suggest also including the rate $\mathcal O(\exp{(-\mu^2 K)\Vert z^0_0 - z^\star \lVert^2} + \tfrac{1}{nK^2})$ of e.g. SEG-RR [17, Thm. 2.1] to make the smaller dependency on $K$ in the linear rate apparent.

Minor:

- SEG-RR/SEG-FF is already introduced using two stepsizes, so it is slightly confusing when EG+ is introduced in (7) and contrasted with EG as being a version with two stepsizes. You might also want to shortly comment that strictly speaking the method you consider is not an instance of [16] as you consider a _larger_ second stepsize as oppose to a smaller second stepsize.

Related work:

- Apart from mention KM iterations it might be worth referencing the Lookahead algorithm for minimax https://arxiv.org/pdf/2006.14567 (LA–ExtraGrad, for which also two different stepsizes are considered, is essential equivalent to SEG-FFA but without flipflip)

**Questions:**

See weaknesses.

**Limitations:**

The paper lacks appropriate discussion on the rates obtained in the main results of Thm. 5.4-5.5 (see weaknesses).

---

> ### Author Rebuttal · Authors · 2024-08-07
>
> We appreciate the effort made by the reviewer in inspecting our manuscript.
>
> 1. **Monotone Case: Why Not Use Full Batch? (W1):**
>
>      This point, raised by the reviewer, is valid. However, in practice, shuffling-based stochastic methods are prevalent; it is not an exaggeration to say that they are now the *de facto* standard. While shuffling-based stochastic minimization is now relatively well understood, there has been limited progress on studies that theoretically support using shuffling-based stochastic minimax optimization methods.  In particular, it was unknown whether shuffling is beneficial in the monotone setting. The main purpose of our study is to enhance our understanding of this already widely used sampling scheme.
>
>     Just to add our two cents: in minimization problems, recall that SGD is asymptotically slower than GD,  as SGD only converges sublinearly on strongly convex problems while GD enjoys linear convergence. In spite of that, SGD and its variants are used everywhere in practice,  whereas full-batch GD appears rarely. We believe the machine learning community has reached a consensus that using SGD has benefits beyond textbook convergence rates, such as better generalization properties and implicit regularization; see for example `[A]` and `[B]`, and we refer to `[C, Chapter 9]` for an in-depth discussion. It is natural to expect similar benefits in minimax problems if stochastic methods are used, but to reach that state, we first have to understand the convergence properties of stochastic minimax methods. Thus, we believe our work is an important step forward in this direction.
>
> 2. **Tradeoff in the Convergence Rate for Strongly Monotone Problems (W2):**
>
>     We have realized that the convergence rate in the original submission was derived suboptimally regrading $\varepsilon$, and a slight modification in the final few steps of the proof makes the exponent of the exponentially decaying term to depend on $K$ instead of $K^\varepsilon$. Please refer to the general comments for the details.
>
> 3. **Theorem 4.1 vs. Hsieh et al. (W3):**
>
>     Hsieh et al. [24] use independent-sample SEG, while we consider same-sample SEG. Thus, the two results can coexist. Our focus lies on the same-sample strategy as it combines more naturally with shuffling-based schemes.  Please refer to the paragraph on lines 57–65, and the footnote in the same page.
>
> 4. **Dependencies on Noise Parameters (W4):**
>
>      Please refer to the general comments.
>
> 5. **On How Rates are Demonstrated in Table 1 (W5):**
>
>     Considering that an improved rate (regarding the exponentially decaying term) has now been obtained, we now could safely say that omitting the exponentially decaying term does not give a decisive advantage to us.  In fact, as we can see from Table 1 in [17] as an example, it is customary to hide the logarithmic dependencies when one summarizes the convergence rates into a table, while deferring the exact formulae to the theorem statements. We hope that the reviewer will agree that, given the improved exponential term, omitting the exponential terms in the table is no longer misleading.
>
> 6. **On Using Two Stepsizes (W6):**
>
>     We are indeed using the term EG in a broad sense, allowing it to use two stepsizes. Hence we introduced the update rule of EG using two stepsizes from the beginning, in (2). However, while we admittedly have not made this point clear enough, we do not assume any explicit constraints on $\alpha_k$ and $\beta_k$, unlike [16] which only considers when $\alpha_k \geq \beta_k$.
>
>     Also, while the inner iteration of SEG-FFA does use $\alpha = \beta/2$, thanks to the anchoring step, the overall update of the epoch sums up to the "standard" EG as introduced by Korpelevich [27]), modulo a small noise. Hence, our method is not *completely* different from [16].
>
>     We understand that these might cause slight confusions. We will try to find a way to present this clearer in our revision.
>
> 7. **Additional Related Work (W7):**
>
>     Thank you for notifying us about the related work.  We agree with you on that there is an interesting resemblance between SEG-FFA and the mentioned Lookahead methods, and we will add a comment on it in our revision.
>
> We hope that our response resolves your concerns. We would greatly appreciate it if you could consider re-evaluating our paper. Thank you.
>
> > [A] On Large-Batch Training for Deep Learning, Keskar et al. (2017)
> >
> > [B] On the Generalization Benefit of Noise in Stochastic Gradient Descent, Smith et al. (2020)
> >
> > [C] Understanding Deep Learning, S.J.D. Prince (2023)

---

> > ### Comment · Reviewer_k7BS · 2024-08-09
> >
> > I thank the authors for the rebuttal, which addresses my concern regarding the strongly monotone case, where there now seems to be a significant improvement in terms of the rate. Trusting that the derivation is correct and that the constant $b$ does not have strange dependencies (like e.g. on $\Vert z^0-z^* \Vert^2$), I have increased my score.
> >
> > I still consider the results in the monotone case somewhat unsatisfying, since the rate $\mathcal O(1/K^{1/3})$ is worse than the $\mathcal O(1/K)$ rate of EG. In other words, there seem to be no reason for running SEG-FFA if we know that only monotonicity holds:
> >
> > - For this reason I would downplay the monotone case by instead focusing on SEG-FFA provably benefiting the strongly monotone, while providing some kind of "fallback" guarantee in the monotone case.
> > - Since we are in the finite sum case it is possible to run deterministic methods (e.g. EG) even with same memory footprint by simple accumulating gradients. I suggest comparing against EG in Table 1.
> >
> > > "there is an interesting resemblance between SEG-FFA and the mentioned Lookahead methods"
> >
> > I would argue that its more than a resemblance, since Lookahead has also been connected with the Krasnosel’skii-Mann iteration (see e.g. https://arxiv.org/pdf/2310.13459). Compared with Lookahead with EG+ as the base optimizer the main differences seems to be:
> >
> > - SEG-FFA picks a smaller extrapolation stepsize (so that the inner loop is no longer strictly the EG+ scheme of [16] – this is why I think it is important to differentiate SEG-FFA from EG+ regarding the inner loop. Your modification is crucial!)
> > - SEG-FFA uses flip-flop
> >
> > Both are important modification, so I am not trying to argue against the contributions, but I think making the connection precise might be valuable. Especially considering that both Lookahead for minimax and the original Lookahead algorithm for minimization (https://arxiv.org/pdf/1907.08610) introduces the scheme in order to reduce variance.

---

> > > ### Author Response · Authors · 2024-08-13
> > >
> > > We appreciate your thoughtful reevaluation of our work.
> > >
> > > We agree that the strongly monotone case results now deserve more emphasis than in the original manuscript given the improvement, and that the current convergence rate in the (star-)monotone case is not ideal. While we fully respect your perspective, we also hope you understand our assertion that the monotone case result is worth some highlighting, as it effectively demonstrates our second-order matching framework and offers new theoretical explanations on how the SEG variants behave in the more practical setting where stochastic gradients are used throughout the optimization procedure. In the revision, we will
> > > adjust our tone, taking into account what deserves more attention and what remains relatively limited.
> > >
> > > Regarding the connection between our work and the Lookahead method, we thoroughly agree with you; we condensed our viewpoint into the phrase “interesting resemblance” because of the character limit. Your additional suggestions will help us make a more detailed comparison in our revision.
> > >
> > > Please feel free to add any further comments or questions.

---

### Official Review · Reviewer_hmmd · 2024-07-11

**Soundness:** 3
**Presentation:** 3
**Contribution:** 3
**Rating:** 7
**Confidence:** 3

**Summary:**

In this paper the authors study stochastic extragradient methods for solving unconstrained minimax convex-concave problems with a finite sum structure. In particular, various shuffling schemes (random reshuffling without replacement, flip-flop, uniform sampling with replacement) are investigated and it was shown that stochastic extragradient with them can lead to divergent behavior when $f$ is merely convex-concave. The authors proceed to propose stochastic extragradient with flip-flop + anchoring which successfully converges in convex-concave problems with rate $O(1/K^{1/3})$ and rate $O(1/nK^{4-5\epsilon})$ for $\epsilon < 2/3$ when $f$ is strongly-convex-strongly-concave. This is supplemented with a lower bound on random-reshuffling based stochastic gradient descent-ascent and stochastic extragradient in the same setup, demonstrating advantage of the proposed algorithm. The analysis is based on controlling the degree of approximation to a known convergent method. Some numerical experiments are presented to illustrate the practical performance.

**Strengths:**

The results in this paper removed several limitations from the previous work and shed light on algorithm design for the appropriate sampling scheme on stochastic min-max problem. I believe the contributions are of great interest for the community and the resulting proposal is conceptually simple, easy to implement with minimum modification from known algorithms. The exposition is clear with nice explanation of the design principle (second-order matching to a known deterministic convergent method) and the relevant literature is thoroughly surveyed and compared against. The technical claims are sound with supporting numerical evidence.

**Weaknesses:**

I've listed a few questions in the section below but I don't think there are major weakness with the paper.

Minor comments:

- In the second set of equations in (2), I'd recommend changing the subscript in nabla to u and v instead, or some other notation to avoid confusion.

- In the definition of Assumption 3.3, it might make more sense to call each $F_i$ $L$-lipschitz (corresponding to (i)) and each $F_i$ $M$-smooth (corresponding to (ii)) - this would possibly make things more consistent with convex optimization literature. In Line 178, should it be $f$ being $L$-smooth or $F$?

**Questions:**

- It seems like this gap of $1/nK^3$ for SEG-RR vs. $1/nK^{4-5\epsilon}$ for SEG-FFA only exists in a very narrow range of $\epsilon$. Is $\epsilon$ tied to other parts of the results? If not, I'd recommend just optimizing over $\epsilon$ and state the final complexity instead.

- Is there any special meaning behind the $1/2$ in (12)? Would other convex combination of the iterates work as well? Or possibly with other anchor points or even more than $1$ point help for higher-order matching?

- In Theorem 5.4, can one afford a constant (and larger) stepsize if instead of asking guarantee in $\min_k ...$ we do tail averaging?

- The parameter $\rho$ or $\sigma$ from Assumption 3.4 don't make an appearance in Theorem 5.4 and 5.5?

- What's the difficulty of generalizing this framework from same sample to independent sample extragradient methods?

**Limitations:**

Yes it's adequately addressed.

---

> ### Author Rebuttal · Authors · 2024-08-07
>
> We appreciate the reviewer for the positive feedback and thoughtful comments.
>
> 1. **On the Comments on the Notations Used (W1, W2):**
>
>     The reviewer has made valid points, and let us share our thoughts about the comments made by the reviewer. The second set of equations in (2) may cause a bit of confusion, as concerned by the reviewer. However, considering the problem formulation in Eq. $(1)$, the subscripts $x$ and $y$ are to represent which argument we are taking the derivative with respect to. In contrast, $u$ and $v$ in Eq. $(2)$ are actual points we evaluate the gradients at. This notation is consistently used throughout the paper.
>
>     Regarding the second suggestion, we realized that saying that $F_i$ is $L$-Lipschitz and $M$-smooth indeed sounds more natural, coherent to the existing literature. We will make necessary changes in the revision.
>
>     In line 178, as the first equation in Asmp. 3.3 asserts that $F$ is $L$-Lipschitz, we would have $f$ being $L$-smooth, as in the original submission.
>
> 2. **Existence of $\varepsilon$ in the Convergence Rate (Q1):**
>
>     Please refer to the general comments.
>
> 3. **Other Possible Variants of SEG-FFA (Q2):**
>
>     The weights $1/2$ in the convex combination $(12)$ are judiciously chosen so that SEG-FFA achieves second-order matching, under the choice of $\alpha_k = \eta_k /2 $ and $\beta_k = \eta_k$. Our Prop. D.1 demonstrates what happens when we choose other convex combinations. It is indeed possible to choose $\alpha_k$, $\beta_k$, and the weight $\theta$ in the convex combination as long as Eq. (29) achieves an error of $\mathcal{O}(\eta^3)$ à la Prop. 5.3.
>
> 4. **On the Idea of Tail Averaging (Q3):**
>
>     Based on our current understanding, the benefits of the averaging technique (in minimization problems) mostly comes from applying Jensen's inequality. However, in minimax problems where a possibly nonconvex function $||F(\cdot)||$ is used as an optimality measure, we are not sure how averaging could be applied. We would also like to remark that this inability to apply Jensen's inequality to $||F(\cdot)||$ was overlooked in [17], leading to an invalid convergence result.
>
> 5. **Appearence of the Noise Parameters $\rho$ and $\sigma$ (Q4):**
>
>     Please refer to the general comments.
>
> 6. **Extending to Independent Sampling (Q5):**
>
>     As we work in the context of random reshuffling schemes, the correct update rule for the independent-sample setting is less clear than in the same-sample setting. The main challenge in generalizing our framework to independent-sample SEGs thus lies in interpreting it in terms of random reshuffling, rather than in technical difficulties. Nonetheless, should we aim to devise an independent-sample method, the simplest way would be to use two independently chosen permutations, one for the extrapolation step and the other for the update step. For such a method, Eq. (8) reads $w_i^k = z_i^k - \alpha T_i^k z_i^k$,  $z_{i+1}^k = z_i^k - \beta T_{\pi(i)}^k w_i^k$ for some permutation $\pi:[n]\to[n]$. Accordingly, the right hand side of Eq. (11) then becomes ${\alpha\beta} \sum_{j=0}^{N-1} DT_{\pi(j)}^k (z_0^k) T_j^k z_0^k + {\beta^2} \sum_{ i < j }  DT_{\pi(j)}^k (z_0^k) T_i^k z_0^k$. It is not difficult to check that the exact same choices of $\alpha$, $\beta$, and $\theta$ used in the same-sample SEG-FFA will also achieve second-order matching in the independent-sampling case. As our convergence analyses are valid for generic methods that achieve the second-order matching property (recall the statements of Theorems F.5 and G.4), we expect that it is possible to derive analogous results for the independent-sample variant of SEG-FFA, but there remain details to be checked of course.

---

> > ### Comment · Reviewer_hmmd · 2024-08-09
> >
> > Thanks for the explanation. I intend to keep my rating.

---

> > > ### Author Response · Authors · 2024-08-13
> > >
> > > Thank you for your time and effort in reviewing our work. If you have anything you would like to discuss further, please feel free to leave additional comments.

---

### Official Review · Reviewer_jRnf · 2024-07-12

**Soundness:** 3
**Presentation:** 3
**Contribution:** 3
**Rating:** 6
**Confidence:** 4

**Summary:**

This paper proposes a new algorithm, SEG-FFA, which converges for the convex-concave minimax problem, while existing algorithms like SEG-RR and SEG-FF diverge for the same class of problem. Moreover, the authors show that SEG-FFA can better approximate EG in comparison to SEG-RR and SEG-FF.

**Strengths:**

- It proposes a new stochastic algorithm, SEG-FFA, which simultaneously incorporates flip-flop shuffling and anchoring to prove convergence. To the best of my knowledge, this is the first work to successfully combine these two techniques.
- Usual Halpern iterations take a convex combination with the initial iterate, while SEG-FFA takes a convex combination with the initial iterate of the epoch. This idea is interesting and certainly paves the way for further research.
- I find section 5.1 very interesting, where the authors explain how well different algorithms can approximate EG.

**Weaknesses:**

- Lipschitz Hessian in Assumption 3.3 is too restrictive, and the authors agree. I don't know if practical ML applications satisfy such assumptions.
- In each epoch, SEG-FFA and SEG-FF make $4n$ oracle calls compared to $2n$ oracle calls made by SEG-RR and SEG-US and $n$ oracle calls by SGDA-US and SGDA-RR. Therefore, the comparison made in the second plot of Figure 1 is unfair. I recommend having the number of oracle calls on the $x$-axis to compare these algorithms fairly.
- In Theorem 5.4, authors propose a decreasing step size of the order $\mathcal{O} \left( 1/ k^{1/3} \log k \right)$ while in plot 1 of the experiment, they use a different step size. Why not use the step size proposed in the Theorem?

**Questions:**

- What is $\eta_k$ in Theorem 5.4? The step sizes of SEG-FFA are $\alpha_k$ and $\beta_k$ in Eq 5.
- Is it possible to relax Assumption 3.3?
- Multiple papers show the convergence of SEG and Stochastic Past Extragradient [1] method for monotone problems with increasing batch sizes. Moreover, a work [2] proposes BCSEG (bias-corrected SEG), which proves convergence without increasing batch sizes. I recommend comparing these works with your Theorem 5.4 in terms of the number of oracle calls to achieve a given accuracy $\epsilon$.


[1] "Single-Call Stochastic Extragradient Methods for Structured Non-monotone Variational Inequalities: Improved Analysis under Weaker Conditions."

[2] "Solving stochastic weak Minty variational inequalities without increasing batch size."

---

> ### Author Rebuttal · Authors · 2024-08-07
>
> We appreciate the effort made by the reviewer in inspecting our manuscript.
>
> 1. **On the Lipschitz Hessian Assumption (W1):**
>     As both our manuscript and the reviewer have mentioned, the Hessian Lipschitz assumption is a somewhat unusual one, not widely assumed in the literature. Still, it is not too difficult to find machine learning problems that satisfy the Hessian Lipschitz condition. As a basic example, consider the logistic loss function $\ell(w;a) = \log(1+e^{aw})$. It is not difficult to verify that $\ell'''(w;a) = \frac{a^3 e^{aw} \left(1 - e^{aw}\right)}{\left(1 + e^{aw}\right)^3} = \frac{a^3}{1+e^{aw}} \cdot \frac{e^{aw}}{1+e^{aw}} \cdot \tanh(-\frac{1}{2}aw)$ and thus $|\ell'''(w;a)| \leq |a|^3$. Now, when given a dataset $(x_i, y_i), i=1, \dots, n$, recall the typical form of the loss function $\frac{1}{n} \sum_{i=1}^n \ell(w;-y_ix_i)$. Since $|\ell'''(w;-y_ix_i)|$ is upper bounded by a constant $\max_{i=1,\ldots,n} |y_i x_i|^3$ for all $i$, the Hessian Lipschitzness follows.
>
> 2. **Relaxations of Assumption 3.3 (Q2):**
>     While we unfortunately do not have a definite answer to this question, let us at least share our intuitions on this assumption. The reason we impose the Hessian Lipschitzness assumption originates from the analysis of the flip-flop sampling scheme (*cf.* [41]). In short, the desideratum is that the change in the Hessian of $f$ remains controllable throughout an epoch, so that after aggregating the updates we get a good enough approximation of a deterministic EG step. If such control is possible with other assumptions then one would be able to replace Assumption 3.3 with it.
>
>       We would also like to make a final remark that as we discussed in lines 199–202, the lower bound results that are obtained under our set of assumptions also serve as valid lower bounds for weaker settings that are implied by ours, so having the Lipschitz Hessian assumption is not a weakness in this scope.
>
> 3. **Comparing Methods on the Same Plot (W2):**
>     We would like to remark that, in the plot(s), we actually do count in the fact that SEG-FF and SEG-FFA make twice as many oracle calls than other methods.
>         That is why for those two methods we plot the values of $\|\|Fz_0^{t/2}\|\|$ instead of $\|\|Fz_0^{t}\|\|$; please refer to the captions below the plots.
>
> 4. **Choice of Stepsizes in the Experiments (W3):**
>     We thank the reviewer for bringing up this point.  We realized that the rationale behind our choice of the $\mathcal{O}(1/k^{0.34})$ stepsize schedule  has not been thoroughly discussed in the submission, so please allow us to elaborate further.
>
>     Strictly speaking, to make use of the convergence result of Thm 5.4—or more precisely, Thm. G.4—one should choose $\eta_0$ so that it satisfies both equations $(121)$ and $(122)$. However, if one closely examines the proof of Thm. G.4., it is not difficult to realize that choosing the stepsizes so that $\eta_k = \Omega({1}/{k^q})$ for $\frac{1}{3} < q < \frac{1}{2}$ while $\eta_k \leq \frac{\eta_0 \sqrt[3]{2} \log 2}{(k+2)^{1/3} \log (k+2) }$, we will still achieve a convergence to a stationary point, but at a cost of having a slightly slower rate of convergence $\mathcal{O} ({1}/{K^{1-2q}} )$. But then, since the decay of $1/k^q$ is faster than that of $1/(k^{1/3}\log k)$, simply taking $\eta_k = {\eta_0^*}/{k^e}$ for a suitably small $\eta_0^*$ will suffice, as there will exist some $K_0$ such that $k \geq K_0$ implies the inequality $\eta_k \leq \frac{\eta_0 \sqrt[3]{2} \log 2}{(k+2)^{1/3} \log (k+2) }$, and we may simply ignore the first $K_0$ iterates to get a convergence guarantee.
>
>     We will add a more formal version of the discussion above in our revision.
>
> 5. **Notation in Thm. 5.4 (Q1):**
>     Please refer to the general comments.
>
> 6. **Additional Comparisons with the Existing Works (Q3):**
>     Despite everything, if we were to compare the number of gradient oracle calls according to their face values, SEG-FFA would in fact require $\tilde{\Omega}(1/\epsilon^3)$ calls as the rate of convergence is $\tilde{\mathcal{O}} (1/K^{1/3})$, whereas, for example SPEG would require $\Omega(1/\epsilon^2)$ calls as claimed in p. 34 in their paper.
>
>      However, we would like to raise the point of whether it is fair to compare our SEG-FFA in its current form to methods that allow increasing batch sizes. SEG-FFA allows a constant batch size (of 1), which is common in practice, by coping with noise induced by gradient variance. Conversely, increasing batch size is not only rarely used in practice but also essentially sidesteps the complexities of dealing with gradient variance, as a batch size of $b$  reduces variance by a factor of $b$. Please recall Thm. H.4, where we showed that methods known to converge with increasing batch sizes may no longer converge if the batch size is kept constant.
>
>     Thus, if we really were to compare SEG-FFA to such methods, it shall be allowed for SEG-FFA to have increasing batch sizes as well. But this approach not only reduces the RHS of Assumption 3.4 by the size of the batch but also reduces the number of passes in an epoch—which is $n$ for single-sample-in-a-batch SEG-FFA—according to how batches are formed. Consequently, the size of the noise term $\|\|r^k\|\|$ will be reduced, and SEG-FFA would enjoy a much better convergence rate.  We believe that convergence analyses incorporating this change should be conducted as future work.
>
>     On a different note, as we have listed in Table 2, the work on BC-SEG+ ([39] in our paper) assumes uniformly bounded gradients. This, as discussed in lines 190–196, is too strong to be a realistic assumption, so we have not included a thorough review of it in the current submission. But yet, we agree that supplementing additional comparisons with [39], and the suggested related work by Choudhury et al., will make how our paper is positioned within the existing literature clearer. We will modify Section B.1 accordingly.

---

> > ### Comment · Reviewer_jRnf · 2024-08-09
> >
> > Thank you for the reply. Please add the above details in the updated version.
> >
> > I will raise my score to 6.

---

> > > ### Author Response · Authors · 2024-08-13
> > >
> > > We are pleased that your concerns have been resolved. We will ensure that the additional details discussed in the comments are well incorporated into the revision.
> > >
> > > If you have any further questions or comments, we would be happy to address them.

---

### Official Review · Reviewer_19M7 · 2024-07-12

**Soundness:** 3
**Presentation:** 3
**Contribution:** 3
**Rating:** 6
**Confidence:** 2

**Summary:**

The paper studies various same-sample SEG algorithms under different shuffling schemes, including SEG-US, SEG-RR and SEG-FF. The three algorithms all can diverge when $f$ is convex-concave. Furthermore, the authors discuss the underlying cause for the nonconvergence of the three algorithms. Moreover, the authors propose a novel stochastic extragradient method named as SEG-FFA, which is SEG amended with flip-flop shuffling scheme and anchoring. The proposed algorithm enjoys improved convergence guarantees and the convergence rate is calculated under different conditions of $f$. Finally, the authors conduct numerical experiments to verify the convergence of SEG-FFA.

**Strengths:**

1. The paper proposes a novel algorithm SEG-FFA for minimax problems, which achieves a better convergence rate when $f$ is strongly-convex-strongly-concave. Furthermore, the algorithm enjoys convergence when $f$ is convex-concave while other baseline algorithms diverge under this setting.
2. The paper provides comprehensive proof for divergence of other baseline algorithms when $F$ is merely monotone.
3. The proposed algorithm SEG-FFA can match extragradient up to second-order terms and get an error of $\mathcal{O}(\eta)$ where $\eta$ is the stepsize. As a result, SEG-FFA achieves convergence in monotone problems.
4. Sufficient numerical experiments verify the convergence result and superiority of SEG-FFA.

**Weaknesses:**

1. Due to the presence of anchoring step, the initial point seems to be more important, which implies that the algorithm is more dependent on the selection of the initial points.
2. How to select the parameter $\epsilon$ concerning convergence in Table 1 and Theorem 5.5 for practical applications and specific problems.
3. Table 1 does not provide the lower bound of SEG-FFA under convex-concave setting. Is there such a lower bound on the iterative complexity for SEG-FFA?

**Questions:**

1. Section 4 explains the reasons why simple shuffling is not enough for convergence for monotone problems. However,  why the added anchoring step can make the algorithm SEG-FFA converge is not sufficiently demonstrated in the proof, especially compared with SEG-FF.
2. What is the difference between the stepsizes $\alpha_k, \beta_k$ of SEG-FF and $\eta_k$ of SEG-FFA.
3. In Figure 4(a) and 4(b), the convergence result of SEG-FFA is better than other baseline algorithms. However, the convergence rate appears slower.
4. There are few typos in the main text and appendix.

**Limitations:**

1. The length of the paper is excessive, especially the appendix section.
2. The experiments could consider demonstrating the performance of SEG-FFA on more large-scale problems.

---

> ### Author Rebuttal · Authors · 2024-08-07
>
> We appreciate the reviewer for the constructive feedback and thoughtful comments.
>
> 1. **Importance of Initial Point due to the Anchoring Step (W1):**
>
>     For any optimization method, its behavior is more or less influenced by the choice of the initial point, and our SEG-FFA is not an exception. However, as we have shown (in Prop. 5.3) that the total update of SEG-FFA made over an epoch is equal to an update made by a deterministic EG up to a small $\mathcal{O}(\eta^3)$ noise, we think the dependency of SEG-FFA on the initial point is as minimal as that of the deterministic EG.
>
> 2. **On $\varepsilon$ in the Convergence Rates (W2):**
>
>     Please refer to the general comments.
>
> 3. **Lower Bounds on SEG-FFA (W3):**
>
>     Unfortunately, at the moment we are not aware of the lower bound complexities for SEG-FFA. To the best of our knowledge, our work is the first to introduce the idea of flip-flop + anchoring, and has not been investigated even in the context of minimization problems. A search for an explicit lower bound would also be an interesting direction of future work.
>
> 4. **Why SEG-FFA is better than SEG-FF (Q1):**
>
>     The explanations on how SEG-FFA outperforms SEG-FF are detailed thoroughly in Section 5 through the lens of Taylor expansion matching, with a summary in lines 257–261, rather than in the proofs. To summarize, SEG-FFA can achieve a second-order matching to the deterministic EG, leaving an error as small as $\mathcal{O}(\eta^3)$. Whereas, SEG-FF is at best only a first-order matching method, since an attempt to make it a second-order matching method leads to it approximating a nonconvergent method. Please refer to to Section 5.1.2, and the related Appendices D and E, for the details.
>
> 5. **Difference in the Notations for Stepsizes across the Methods (Q2):**
>
>     Please refer to the general comments.
>
> 6. **Interpretations on Fig. 4 (Q3):**
>
>     The reviewer has made a correct observation. However, as discussed in lines 1519–1521, these results do not counter our theoretical analyses. Indeed, the convergence results in the strongly monotone setting are established under a fixed choice of the time horizon $K$ (see Thm. 5.5). More precisely, the results for strongly monotone $F$ are interested on how small $\|\| F z \|\|$ can be eventually made after running for $K$ epochs. So, for the initial few steps, the *speed* of convergence may be slightly suboptimal.
>
> 7. **Typos (Q4):**
>
>     Thank you for notifying us about the typos. We will try our best to fix them. We would also greatly appreciate it if you could point out some of them to help us correcting them.
>
> 8. **Paper is Too Long (L1):**
>
>     It is true that the paper is longer than average, but we hope that the reviewer understands the length of the paper as an unavoidable side effect of encompassing multiple rigorous proofs with technical details.
>
> 9. **Additional Experiments on Large-Scale Problems (L2):**
>
>     Due to the nature of our paper focusing more on the theoretical results, we admittedly did not consider much about additional experiments. Yet, we thank the reviewer's suggestion, and will contemplate on which additional experiments will be beneficial to our paper.

---

> > ### Comment · Reviewer_19M7 · 2024-08-13
> >
> > I thank the authors for providing detailed replies to all my concerns.  The authors' discussions on the practical algorithms and theoretical investigation are thoughtful.

---

> > > ### Author Response · Authors · 2024-08-13
> > >
> > > We are glad that our explanations have cleared up your concerns. If there are further comments or questions, we would be glad to continue our discussion.

---

### Author Rebuttal · Authors · 2024-08-07

We thank all the reviewers for their constructive feedback. Here, we would like to discuss some important
issues commonly raised by the reviewers.

1. **On the Convergence Rate in the Strongly Monotone Setting ([k7BS] W2, [hmmd] Q1, [19M7] W2):**

    One of the common questions raised by the reviewers was about the parameter $\varepsilon$ in the convergence rate derived in the strongly monotone setting. Admittedly, in our original submission, we were more or less satisfied with getting a rate that is provably faster than the $\Omega(1/nK^3)$ lower bound of SEG-RR, hence the rate we derived remained having an additional obscure parameter $\varepsilon$. However, reading the comments from the reviewers, and following the suggestion of one of the reviewers, we revisited our analyses to see if we can optimize the rate with respect to $\varepsilon$. Surprisingly, with a slight modification in the proofs, we were able to remove the dependency on $\varepsilon$ and derive a rate of $\mathbb{E} \|\| z_0^K - z^*\|\|^2 \leq \exp \left( - \frac{b \mu^2 K}{L^2}  \right)  \|\|z_0^{0} - z^*\|\|^2 + \mathcal{O}\left(\frac{\left(\log(n^{1/4} K)\right)^{4}}{ n  K^{4}} \right)$ for a constant $b$ that does not depend on $\mu$, $L$, $n$, and $K$.

    Since we cannot make a revision of our paper during the discussion phase, please allow us to briefly explain how our proofs can be modified to obtain a new rate of $\tilde{\mathcal{O}}(1/nK^4)$ below.  We are more than happy to take follow-up questions if anyone wishes for further verifications/clarifications.

    First we modify Lemma F.4, so that the number of epoch $K$ is no longer involved: more precisely, we show that using a constant stepsize (possibly independent of $K$) it holds that $\|\| F z^k\|\| \leq \|\| F z^0\|\| + V_1 / (\mu L)$ for all $k = 0, 1, \dots$. The trick is simple. In the current form of Lemma F.4 we show that $\|\|Fz^{k+1}\|\|$ is bouned by $\|\|Fz^k\|\|$ plus some error term; see Eq. (95). However, by choosing a smaller stepsize it is not difficult to improve this bound to $(1-\frac{\mu \eta n}{10}) \|\|Fz^k\|\|$ plus some error term. This "exponential decay with an error" type of recurrence also arises in the analysis of SGD on stongly convex problems (see, *e.g.*, Thm. 5.8 in arXiv:2301.11235), so we can use standard techniques with mathematical induction to unroll this recurrence and get the claimed bound.

    With this modification, in Thm. F.5, we then have that the inequality right below line 1057 holds whenever $\eta$ is sufficiently small, regardless of the choice of $K$. Again unravelling this ``exponential decay with an error'' type recurrence, one gets $\mathbb{E} \|\| z^K - z^* \|\|^2 \leq \left(1 - \frac{1}{2} \mu \eta n \right)^K \|\| z^{0} - z^* \|\|^2 + \frac{4+2\mu \eta n}{\mu^2} \cdot \eta^{2a-2} n^{2a-3} V_2$ , for any $K \geq 0$. Compare this with the first line of Eq. (101), which was obtained by a slightly looser unravelling of the same recurrence. However, unlike Eq. (101) where $\eta$ was already constrained to be $\mathcal{O}(1/K^{1-\varepsilon})$ due to (the unmodified) Lemma F.4, we have no dependency of $\eta$ on $K$ yet. So we can choose $\eta$ now, to have the best dependency on $K$ that minimizes the bound. We set $\eta$ to be $\mathcal{O}(\log(K) / K)$, inspired by the convergence analyses made in p. 26 of [17]. It is then easily able to show the $\tilde{\mathcal{O}}(1/nK^4)$ rate claimed above.



2. **Dependence of Convergence Rates on the Noise Parameters $\rho$ and $\sigma$ ([k7BS] Q4, [hmmd] Q4):**

    As mentioned by the reviewers, it is indeed good practice to write down the dependencies on the problem-dependent constants as explicitly as possible. However, the dependencies of the convergence rates on $\rho$ and $\sigma$ are quite involved (see, *e.g.*, Theorems E.9 and E.13), so we decided to highlight the concise versions of the rates in the main text, while deferring the exact details to the appendices. We hope the reviewers generously understand this as our effort to find the sweet spot between readability and rigor.



3. **What is $\eta_k$ in Theorem 5.4? ([jRnf] Q1, [19M7] Q2):**

    We admit that the relationship between the step sizes $\alpha_k,\beta_k$ and $\eta_k$ for the SEG-FFA  was not clearly stated. Following Prop. 5.3, regarding equations (5) and (6) in the case of SEG-FFA, we selected step sizes $\alpha_k = \beta_k/2$ and $\beta_k = \eta_k$ for some sequence $\eta_k$, $k \geq 0$. This was implicitly assumed in Thm. 5.4 (and only written explicitly in Alg. 4). We will clarify this by adding that we choose $\alpha_k = \beta_k/2$ and $\beta_k = \eta_k$ in the statement of Thm. 5.4.

---

### Decision · Program_Chairs · 2024-09-25

**Decision:**

Accept (poster)

**Comment:**

This paper introduces a new algorithm, SEG-FFA, which successfully converges for the convex-concave minimax problem, whereas existing algorithms such as SEG-RR and SEG-FF fail to converge for this class of problems. Additionally, the authors establish both upper and lower bounds in the strongly convex-strongly concave setting, demonstrating that SEG-FFA achieves a provably faster convergence rate compared to other shuffling-based methods.

All reviewers agree that the paper is clearly written and presents a novel approach of significant interest. Some questions were raised regarding assumptions about Hessian Lipschitzness, hyperparameter settings, and comparisons with EG. These concerns have been satisfactorily addressed in the authors' responses. Therefore, I recommend acceptance.